# Bayesian Optimization of Robustness Measures under Input Uncertainty: A Randomized Gaussian Process Upper Confidence Bound Approach

**Yu Inatsu** *inatsu.yu@nitech.ac.jp*
*Department of Computer Science*
*Nagoya Institute of Technology*

**Reviewed on OpenReview:** *https://openreview.net/forum?id=FDzojiLSia*

## Abstract

Bayesian optimization based on the Gaussian process upper confidence bound (GP-UCB) offers a theoretical guarantee for optimizing black-box functions. In practice, however, black-box functions often involve input uncertainty. To handle such cases, GP-UCB can be extended to optimize evaluation criteria known as robustness measures. However, GP-UCB-based methods for robustness measures require a trade-off parameter, $\beta$, which, as in the original GP-UCB, must be set sufficiently large to ensure theoretical validity. In this study, we propose randomized robustness measure GP-UCB (RRGP-UCB), a novel method that samples $\beta$ from a chi-squared-based probability distribution. This approach eliminates the need to explicitly specify $\beta$. Notably, the expected value of $\beta$ under this distribution is not excessively large. Furthermore, we show that RRGP-UCB provides tight bounds on the expected regret between the optimal and estimated solutions. Numerical experiments demonstrate the effectiveness of the proposed method.

## 1 Introduction

In this study, we address the optimization problem of robustness measures for black-box functions under input uncertainty. In various practical applications, particularly in engineering, black-box functions with high evaluation costs are frequently used. In practice, these functions often exhibit input uncertainty. Let $f(\boldsymbol{x}, \boldsymbol{w})$ be a black-box function, where $\boldsymbol{x} \in \mathcal{X}$ and $\boldsymbol{w} \in \Omega$ are input variables referred to as design variables and environmental variables, respectively. The design variable $\boldsymbol{x}$ is completely controllable, whereas the environmental variable $\boldsymbol{w}$ is uncontrollable and follows a certain probability distribution. In practical applications, identifying the optimal design variables for black-box functions that include stochastic environmental variables requires the use of measures that depend solely on the design variables while accounting for the influences of environmental uncertainty. Robustness measures are evaluation criteria defined only in terms of the design variables, effectively removing influence of environmental uncertainty. Examples of such robustness measures include the expectation measure $\mathbb{E}_{\boldsymbol{w}}[f(\boldsymbol{x}, \boldsymbol{w})]$, which takes the expected value over the distribution of the environmental variables, and the worst-case measure $\inf_{\boldsymbol{w} \in \Omega} f(\boldsymbol{x}, \boldsymbol{w})$, which considers the worst-case scenario. In this paper, we consider the following optimization problem for a given robustness measure $F(\boldsymbol{x})$:

$$\arg \max_{\boldsymbol{x} \in \mathcal{X}} F(\boldsymbol{x}).$$

Bayesian optimization (BO) (Shahriari et al., 2015), based on Gaussian processes (GPs) (Williams & Rasmussen, 2006), is a powerful approach for optimizing black-box functions. Numerous BO methods have been developed for optimizing black-box functions without input uncertainty. In contrast, applying standard GP-based BO methods to the optimization of robustness measures under input uncertainty is not straightforward. The main difficulty lies in the fact that even if the black-box function $f$ follows a GP, the

resulting robustness measure $F$ generally does not follow a GP. However, recent studies have proposed BO methods specialized for specific robustness measures by utilizing the GP assumption for $f$, without requiring distributional information about $F$ (Iwazaki et al., 2021b; Nguyen et al., 2021b;a; Kirschner et al., 2020). In addition, methods have been proposed for optimizing general robustness measures (Cakmak et al., 2020), as well as for multi-objective robust optimization using BO (Inatsu et al., 2024a).

However, a theoretical evaluation of the performance of BO is vital. Regret, defined as the difference between the solution obtained by an optimization algorithm and the true optimal solution, is commonly used to evaluate the performance of such algorithms. In particular, within the standard BO framework without input uncertainty, the Gaussian process upper confidence bound (GP-UCB) algorithm (Srinivas et al., 2010) is a prominent example of a BO method with theoretical performance guarantees. GP-UCB has been shown to achieve sublinear regret with high probability by appropriately tuning the trade-off parameter $\beta_t$, which is specified by the user. It is highly scalable, and numerous extensions have been proposed, including multi-objective BO, multi-fidelity BO, high-dimensional BO, parallel BO, multi-stage BO, and BO of robustness measures (Zuluaga et al., 2016; Kandasamy et al., 2016; 2017; 2015; Rolland et al., 2018; Contal et al., 2013; Kusakawa et al., 2022; Iwazaki et al., 2021b; Nguyen et al., 2021b;a; Kirschner et al., 2020; Inatsu et al., 2024a). These extended GP-UCB-based methods also provide theoretical guarantees for regret-like performance metrics.

However, to ensure theoretical validity, the trade-off parameter $\beta_t$ in GP-UCB and its variants must increase on the order of $\log t$ with iteration $t$. This results in overly conservative behavior in practice. Such conservatism can significantly impair practical performance (Takeno et al., 2023). To solve this problem, the improved randomized GP-UCB (IRGP-UCB) (Takeno et al., 2023) was proposed, which replaces the deterministic setting of $\beta_t$ with a random sample drawn from a two-parameter exponential distribution. IRGP-UCB avoids the need to increase $\beta_t$ by $\log t$, thereby mitigating the conservativeness of the theoretically recommended values in GP-UCB. Furthermore, the cumulative regret of BO using IRGP-UCB has been shown to remain sublinear in expectation and achieves a tighter bound than that of the original GP-UCB. Additionally, an optimization method was introduced within the level-set estimation framework that applies a similar sampling-based technique to replace the trade-off parameter in UCB-based methods (Inatsu et al., 2024b). Thus, IRGP-UCB not only resolves the limitations of the original GP-UCB but also shows promise for generalization across a variety of settings, similar to its predecessor. In this study, we propose a new BO method for robustness measures by extending the randomized GP-UCB-based method used in IRGP-UCB.

## 1.1 Related Work

BO is a powerful tool for optimizing black-box functions with high evaluation costs. It typically comprises three main steps: constructing a surrogate model, selecting the next evaluation point, and evaluating the function. GP or the kernel ridge regression model (Williams & Rasmussen, 2006) is commonly used as the surrogate model. The next evaluation point is determined by optimizing a utility function known as an acquisition function. Research in BO has focused on the design of new acquisition functions. For standard black-box optimization problems, widely used acquisition functions include expected improvement (Močkus, 1975), Thompson sampling (Thompson, 1933), entropy search (Hernández-Lobato et al., 2014), knowledge gradient (Wu & Frazier, 2016), and GP-UCB (Srinivas et al., 2010). Many of these acquisition functions have been extended to accommodate various BO settings, such as multi-objective optimization, constrained optimization, high-dimensional problems, and optimization involving robustness measures.

When optimizing a black-box function in the presence of input uncertainty, such as that introduced by environmental variables, robustness measures too have to be optimized. These are defined solely in terms of the design variables and reflect the uncertainty associated with the environmental variables. Representative robustness measures include the expectation measure (Beland & Nair, 2017), worst-case measure (Bogunovic et al., 2018), probability threshold robustness (PTR) measure (Iwazaki et al., 2021a), value-at-risk (Nguyen et al., 2021b), conditional value-at-risk (Nguyen et al., 2021a), mean-variance measure (Iwazaki et al., 2021b), and distributionally robust measure (Kirschner et al., 2020; Tay et al., 2022; 2024). Corresponding acquisition functions for the BO of robustness measures include GP-UCB-based methods (Bogunovic et al., 2018; Iwazaki et al., 2021a; Nguyen et al., 2021b;a; Iwazaki et al., 2021b; Kirschner et al., 2020), knowledge gradient-based methods (Cakmak et al., 2020), Thompson sampling-based methods (Iwazaki et al., 2021a; Tay et al.,

2024), and approximation-based methods (Tay et al., 2022). In particular, GP-UCB-based optimization methods for robustness measures provide theoretical guarantees with respect to regret. Furthermore, Inatsu et al. (2024a) proposed the bounding box-based multi-objective BO (BBBMOBO) method—a theoretically guaranteed GP-UCB-based optimization method for multi-objective robust BO involving multiple general robustness measures. If we consider the special case in which only a single robustness measure is involved, the method proposed in Inatsu et al. (2024a) provides a theoretically guaranteed optimization method based on GP-UCB for general robustness measures. However, in GP-UCB-based optimization methods for robustness measures, establishing theoretical guarantees requires the trade-off parameter $\beta_t$ to increase with the iteration index $t$, resulting in a conservative setting that adversely affects practical performance. On the other hand, Tay et al. (2024) propose a BO method for a robustness measure defined as the weighted sum of the distributionally robust expectation measures and their right derivatives, using a method based on Thompson sampling. Furthermore, they provide theoretical bounds for Bayesian regret, which has a slightly different definition from the regret considered in this study. However, while Thompson sampling does not require trade-off parameters like GP-UCB in terms of algorithm design, it is necessary to use GP-UCB, which requires the trade-off parameter $\beta_t$ to increase with iteration $t$, when performing theoretical analysis. As a result, this influence is also included in the theoretical bounds they derived.

Two studies closely related to the present work are Inatsu et al. (2024a) and Takeno et al. (2023). The former addresses BO for Pareto optimization with multiple robustness measures using a GP-UCB-based framework. As a special case, it also considers optimization for a single robustness measure and provides high-probability regret bounds for general robustness measures. The latter study, Takeno et al. (2023), introduces a randomized approach to GP-UCB in which the trade-off parameter $\beta_t$ is sampled from a two-parameter exponential distribution. This approach avoids the need to increase $\beta_t$ logarithmically and achieves a tighter regret bound than standard GP-UCB under certain conditions. However, the regret bounds achieved by Takeno et al. (2023) depend heavily on the problem setting, the specific definition of regret, and the choice of the sampling distribution for $\beta_t$. As pointed out by Inatsu et al. (2024b), these factors must be carefully tailored to the target problem to replicate the theoretical guarantees in different contexts. Therefore, a direct substitution of the trade-off parameter in the method of Inatsu et al. (2024a) with a random sample from a two-parameter exponential distribution will not suffice to obtain a tighter regret bound. To the best of our knowledge, no research has been conducted on BO methods based on GP-UCB for general robustness measures that achieves a tighter regret bound without requiring the growth of $\beta_t$.

## 1.2 Contribution

In this study, we propose randomized robustness measure GP-UCB (RRGP-UCB), a new algorithm for efficiently optimizing robustness measures of black-box functions. RRGP-UCB modifies the BBBMOBO framework in Inatsu et al. (2024a) by introducing random sampling for the trade-off parameter and selecting points with high uncertainty between the optimistic and average-based maxima. This enables a tighter theoretical analysis of regret for solutions based on the surrogate model's average prediction. Table 1 summarizes the correspondence between IRGP-UCB, BBBMOBO, and RRGP-UCB, while Table 2 presents theoretical bounds on cumulative regret for representative robustness measures and existing methods. In addition, Figure 1 shows the behavior of $\beta_t$ in the proposed method and the relationship of $\beta_t$ with respect to the increase in iteration $t$. The main contributions of this study are as follows:

- RRGP-UCB introduces a randomized trade-off parameter $\beta_t$ for GP-UCB in robustness measure optimization. This randomization, along with certain modifications, eliminates the need to explicitly specify the parameter $\beta_t$ or to increase it on the order of $\log t$. As a result, it avoids the problem of overly conservative behavior.

- RRGP-UCB applies to general robustness measures. We theoretically show that the expected cumulative regret is sublinear for many robustness measures, including the expectation measure.

- RRGP-UCB is extended to various robustness optimization settings: controllable environmental variables (simulator settings), uncontrollable settings, finite input spaces, and continuous input spaces.

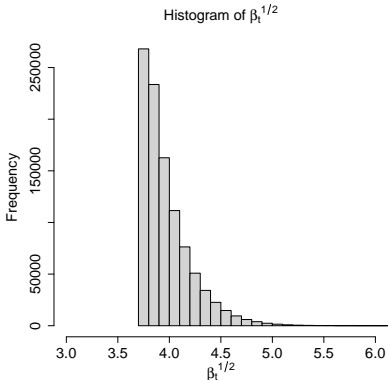
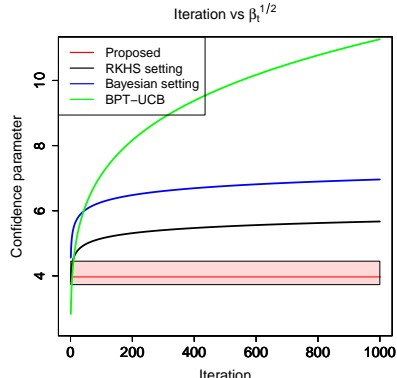

Figure 1: Comparison of $\beta_t$ in the proposed method and existing methods when $|\mathcal{X} \times \Omega| = 1000$. The figure on the left shows the histogram of $\beta_t^{1/2}$ in the proposed method with $1,000,000$ samplings. The figure on the right shows $\beta_t$ at the theoretically recommended value in the proposed method, the setting where the black-box function is assumed to be an element of a reproducing kernel Hilbert space (RKHS setting) (Bogunovic et al., 2018; Nguyen et al., 2021b; Kirschner et al., 2020; Inatsu et al., 2024a), the setting where the black-box function is assumed to be a sample path from a GP (Bayesian setting) (Nguyen et al., 2021a), and BO method for PTR measure (BPT-UCB) (Iwazaki et al., 2021a). The pink area in the figure on the right represents the 95% confidence interval, and for the RKHS setting, Bayesian setting, and BPT-UCB, $1 + \sqrt{2(\log(t) + 1 + \log(1/0.05))}$, $\sqrt{2\log(|\mathcal{X} \times \Omega|\pi^2 t^2/(6 \times 0.05))}$, $\left(|\mathcal{X} \times \Omega|\pi^2 t^2/(6 \times 0.05)\right)^{1/10}$ were used as $\beta_t^{1/2}$, respectively.

Table 1: Theoretical guarantee of regret in IRGP-UCB, BBBMOBO, and RRGP-UCB (Proposed).

| Method | Confidence parameter $\beta_t$ | Next point to be evaluated | Regret |
|---|---|---|---|
| IRGP-UCB (Takeno et al., 2023) | $\beta_t \sim 2\log(|\mathcal{X}|/2) + \chi_2^2$ | $\boldsymbol{x}_t = \arg\max_{\boldsymbol{x} \in \mathcal{X}} \text{ucb}_{t-1}^{(f)}(\boldsymbol{x})$ | $r_t = \max_{\boldsymbol{x} \in \mathcal{X}} f(\boldsymbol{x}) - f(\boldsymbol{x}_t)$ |
| BBBMOBO (Inatsu et al., 2024a) | $\beta_t = \left(B + \sqrt{2(\gamma_t + \log(1/\delta))}\right)^2$ | $\boldsymbol{x}_t = \arg\max_{\boldsymbol{x} \in \mathcal{X}} \left(\text{ucb}_{t-1}^{(F)}(\boldsymbol{x}) - \text{lcb}_{t-1}^{(F)}(\check{\boldsymbol{x}}_t)\right)_+$ | $r_t = \max_{\boldsymbol{x} \in \mathcal{X}} F(\boldsymbol{x}) - F(\check{\boldsymbol{x}}_t)$ |
| Proposed | $\beta_t \sim 2\log(|\mathcal{X} \times \Omega|) + \chi_2^2$ | $\boldsymbol{x}_t = \arg\max_{\boldsymbol{x} \in \{\tilde{\boldsymbol{x}}_t, \hat{\boldsymbol{x}}_t\}} \left(\text{ucb}_{t-1}^{(F)}(\boldsymbol{x}) - \text{lcb}_{t-1}^{(F)}(\boldsymbol{x})\right)$ | $r_t = \max_{\boldsymbol{x} \in \mathcal{X}} F(\boldsymbol{x}) - F(\hat{\boldsymbol{x}}_t)$ |

$\chi_2^2$: Chi-squared distribution with two degrees of freedom
$\gamma_t$: Maximum information gain, $(\cdot)_+ \equiv \max\{\cdot, 0\}$
$\check{\boldsymbol{x}}_t = \arg\max_{\boldsymbol{x} \in \mathcal{X}} \text{lcb}_{t-1}^{(F)}(\boldsymbol{x})$, $\hat{\boldsymbol{x}}_t = \arg\max_{\boldsymbol{x} \in \mathcal{X}} \rho(\mu_{t-1}(\boldsymbol{x}, \boldsymbol{w}))$, $\tilde{\boldsymbol{x}}_t = \arg\max_{\boldsymbol{x} \in \mathcal{X}} \left(\text{ucb}_{t-1}^{(F)}(\boldsymbol{x}) - \text{lcb}_{t-1}^{(F)}(\check{\boldsymbol{x}}_t)\right)_+$

- Experimental results on both synthetic and real-world datasets show that RRGP-UCB achieves performance comparable to or better than existing methods.

## 2 Preliminary

**Problem Setup**  Let $f : \mathcal{X} \times \Omega \to \mathbb{R}$ be an expensive-to-evaluate black-box function, where $\mathcal{X}$ and $\Omega$ are finite sets[1]. For each $(\boldsymbol{x}_t, \boldsymbol{w}_t) \in \mathcal{X} \times \Omega$, we observe $f(\boldsymbol{x}_t, \boldsymbol{w}_t)$ with noise $\varepsilon_t$ as follows: $y_t = f(\boldsymbol{x}_t, \boldsymbol{w}_t) + \varepsilon_t$, where $\varepsilon_1, \ldots, \varepsilon_t$ are mutually independent and follow some distribution with zero mean. Let $\boldsymbol{x} \in \mathcal{X}$ be a design variable and $\boldsymbol{w} \in \Omega$ an environmental variable, where $\boldsymbol{w}$ is uncontrollable and follows a distribution $P^*$. In black-box optimization involving environmental variables, two types of settings are considered: simulator settings (Cakmak et al., 2020; Nguyen et al., 2021b; Beland & Nair, 2017; Iwazaki et al., 2021a) and uncontrollable settings (Kirschner et al., 2020; Inatsu et al., 2024a; 2022; Iwazaki et al., 2021b). In the simulator setting, the value of $\boldsymbol{w}$ can be arbitrarily selected during optimization, whereas in the uncontrollable setting, $\boldsymbol{w}$ cannot be controlled even during optimization. In the main text, we focus on the simulator setting; the uncontrollable setting is discussed in Appendix B. Let $\vartheta(\boldsymbol{w})$ be a function of $\boldsymbol{w}$, and let $\rho(\cdot)$ be a mapping from the function $\vartheta(\cdot)$ to the real numbers. For any function $\varphi(\boldsymbol{x}, \boldsymbol{w})$ defined on $\mathcal{X} \times \Omega$,

---

[1]The case where $\mathcal{X}$ and $\Omega$ are continuous is discussed in Appendix A.

Table 2: Theoretical bounds on cumulative regret $R_T$ for existing and proposed methods for expectation (EXP), value-at-risk (VaR), conditional VaR (CVaR), and distributionally robust expectation (DREXP) measures.

| Method | EXP | VaR | CVaR | DREXP |
|---|---|---|---|---|
| DRBO (Kirschner et al., 2020) | $R_T \leq_\delta \sqrt{T\beta_T\gamma_T}$ | - | - | $R_T \leq_\delta \sqrt{T\beta_T\gamma_T}$ |
| V-UCB (Nguyen et al., 2021b) | - | $R_T \leq_\delta \sqrt{T\beta_T\gamma_T}$ | - | - |
| CV-UCB (Nguyen et al., 2021a) | - | - | $R_T \leq_\delta \sqrt{T\beta_T\gamma_T}$ | - |
| BBBMOBO (Inatsu et al., 2024a) | $R_T \leq_\delta \sqrt{T\beta_T\gamma_T}$ | $R_T \leq_\delta \sqrt{T\beta_T\gamma_T}$ | $R_T \leq_\delta \sqrt{T\beta_T\gamma_T}$ | $R_T \leq_\delta \sqrt{T\beta_T\gamma_T}$ |
| Proposed | $\mathbb{E}[R_T] \leq \sqrt{T\gamma_T}$ | $\mathbb{E}[R_T] \leq \sqrt{T\gamma_T}$ | $\mathbb{E}[R_T] \leq \sqrt{T\gamma_T}$ | $\mathbb{E}[R_T] \leq \sqrt{T\gamma_T}$ |

Definition of cumulative regret $R_T$ is not necessarily the same for each method
$$R_T \leq_\delta a \Leftrightarrow \mathbb{P}(R_T \leq a) \geq 1 - \delta$$

when $\boldsymbol{x}$ is fixed, $\varphi(\boldsymbol{x}, \boldsymbol{w})$ is a function of $\boldsymbol{w}$ and is denoted by $\varphi(\boldsymbol{x}, \cdot)$. In particular, when $\boldsymbol{x}$ is fixed, we define $\rho(f(\boldsymbol{x}, \cdot)) \equiv F(\boldsymbol{x})$ as a robustness measure of $f$ in $\boldsymbol{x}$. Representative robustness measures include: the expectation measure $F_1(\boldsymbol{x}) = \mathbb{E}[f(\boldsymbol{x}, \boldsymbol{w})]$, the worst-case measure $F_2(\boldsymbol{x}) = \inf_{\boldsymbol{w} \in \Omega} f(\boldsymbol{x}, \boldsymbol{w})$, the best-case measure $F_3(\boldsymbol{x}) = \sup_{\boldsymbol{w} \in \Omega} f(\boldsymbol{x}, \boldsymbol{w})$, the $\alpha$-value-at-risk measure $F_4(\boldsymbol{x}; \alpha) = \inf\{b \in \mathbb{R} \mid \alpha \leq \mathbb{P}(f(\boldsymbol{x}, \boldsymbol{w}) \leq b)\}$, the $\alpha$-conditional value-at-risk measure $F_5(\boldsymbol{x}; \alpha) = \mathbb{E}[f(\boldsymbol{x}, \boldsymbol{w}) | f(\boldsymbol{x}, \boldsymbol{w}) \leq F_4(\boldsymbol{x}; \alpha)]$, and the mean absolute deviation measure $F_6(\boldsymbol{x}) = \mathbb{E}[|f(\boldsymbol{x}, \boldsymbol{w}) - F_1(\boldsymbol{x})|]$, where the expectation or probability is taken with respect to $\boldsymbol{w}$. Our goal is to identify the following $\boldsymbol{x}^*$ using as few function evaluations as possible:

$$\boldsymbol{x}^* = \arg\max_{\boldsymbol{x} \in \mathcal{X}} F(\boldsymbol{x}).$$

We emphasize that while the optimization target is $F(\boldsymbol{x})$, we cannot directly observe $F(\boldsymbol{x})$; instead, only the noisy evaluations of $f(\boldsymbol{x}, \boldsymbol{w})$ are available.

**Regularity Assumption** We introduce regularity assumptions for the function $f$. Let $k : (\mathcal{X}, \Omega) \times (\mathcal{X}, \Omega) \to \mathbb{R}$ be a positive-definite kernel such that $k((\boldsymbol{x}, \boldsymbol{w}), (\boldsymbol{x}, \boldsymbol{w})) \leq 1$ for all $(\boldsymbol{x}, \boldsymbol{w}) \in \mathcal{X} \times \Omega$. Assume that $f$ is a sample path from a GP $\mathcal{GP}(0, k((\boldsymbol{x}, \boldsymbol{w}), (\boldsymbol{x}', \boldsymbol{w}')))$ with zero mean and kernel function $k(\cdot, \cdot)$. We further assume that the noise terms $\varepsilon_t$ are independently drawn from a normal distribution with mean zero and variance $\sigma_{\text{noise}}^2$, and that $f, \varepsilon_1, \ldots, \varepsilon_t$ are mutually independent.

**Gaussian Process** In this study, we predict $F$ based on a surrogate model of the black-box function $f$. We assume that the prior distribution of $f$ is a GP $\mathcal{GP}(0, k((\boldsymbol{x}, \boldsymbol{w}), (\boldsymbol{x}', \boldsymbol{w}')))$. Given the dataset $\{(\boldsymbol{x}_j, \boldsymbol{w}_j, y_j)\}_{j=1}^t$, the posterior distribution of $f$ remains a GP. The posterior mean $\mu_t(\boldsymbol{x}, \boldsymbol{w})$ and posterior variance $\sigma_t^2(\boldsymbol{x}, \boldsymbol{w})$ are given by standard results from the GP regression (Williams & Rasmussen, 2006):

$$\mu_t(\boldsymbol{x}, \boldsymbol{w}) = \boldsymbol{k}_t(\boldsymbol{x}, \boldsymbol{w})^\top (\boldsymbol{K}_t + \sigma_{\text{noise}}^2 \boldsymbol{I}_t)^{-1} \boldsymbol{y}_t,$$
$$\sigma_t^2(\boldsymbol{x}, \boldsymbol{w}) = k((\boldsymbol{x}, \boldsymbol{w}), (\boldsymbol{x}, \boldsymbol{w})) - \boldsymbol{k}_t(\boldsymbol{x}, \boldsymbol{w})^\top (\boldsymbol{K}_t + \sigma_{\text{noise}}^2 \boldsymbol{I}_t)^{-1} \boldsymbol{k}_t(\boldsymbol{x}, \boldsymbol{w}),$$

where $\boldsymbol{k}_t(\boldsymbol{x}, \boldsymbol{w})$ is the $t$-dimensional vector whose $j$-th element is $k((\boldsymbol{x}, \boldsymbol{w}), (\boldsymbol{x}_j, \boldsymbol{w}_j))$, $\boldsymbol{y}_t = (y_1, \ldots, y_t)^\top$, $\boldsymbol{I}_t$ is the $t \times t$ identity matrix, and $\boldsymbol{K}_t$ is the $t \times t$ kernel matrix with the $(j, k)$-th element $k((\boldsymbol{x}_j, \boldsymbol{w}_j), (\boldsymbol{x}_k, \boldsymbol{w}_k))$.

**Notations** We summarize particularly important notations used in the main text in Table 3.

## 3 Proposed Method

In this section, we propose a BO method to efficiently identify $\boldsymbol{x}^*$. First, in Section 3.1, we construct credible intervals for $F(\boldsymbol{x})$ based on credible intervals for $f(\boldsymbol{x}, \boldsymbol{w})$. Next, in Section 3.2, we present a method for estimating the optimal solution. In Section 3.3, we describe a method for selecting $\boldsymbol{x}_t$ and $\boldsymbol{w}_t$. The pseudo-code of the proposed algorithm is provided in Algorithm 1.

Table 3: Notations used in the main text and their meanings.

| Notation | Meaning |
|---|---|
| $\boldsymbol{x}$ | Design variable |
| $\boldsymbol{w}$ | Environmental variable |
| $\mathcal{X}$ | Set of design variables |
| $\Omega$ | Set of environmental variables |
| $f(\boldsymbol{x}, \boldsymbol{w})$ | Black-box function defined on $\mathcal{X} \times \Omega$ |
| $\varepsilon_t$ | Observetion noise following a normal distribution |
| $\sigma_{\text{noise}}^2$ | Variance of the noise ditribution |
| $\varphi(\boldsymbol{x}, \boldsymbol{w})$ | Arbitrary function defined on $\mathcal{X} \times \Omega$ |
| $\varphi(\boldsymbol{x}, \cdot)$ | In $\varphi(\boldsymbol{x}, \boldsymbol{w})$, a function with respect to $\boldsymbol{w}$ when $\boldsymbol{x}$ is fixed |
| $\rho(\cdot)$ | Robustness measure, a mapping from functions with respect to $\boldsymbol{w}$ to real numbers |
| $F(\boldsymbol{x})$ | Target robustness measure defined by $F(\boldsymbol{x}) = \rho(f(\boldsymbol{x}, \cdot))$ |
| $\boldsymbol{x}^*$ | Optimal solution defined by $\arg\max_{\boldsymbol{x} \in \mathcal{X}} F(\boldsymbol{x})$ |
| $k(\cdot, \cdot)$ | Kernel function defined on $(\mathcal{X} \times \Omega) \times (\mathcal{X} \times \Omega)$ |
| $\mu_t(\boldsymbol{x}, \boldsymbol{w})$ | Posterior mean based on GP for $f(\boldsymbol{x}, \boldsymbol{w})$ |
| $\sigma_t^2(\boldsymbol{x}, \boldsymbol{w})$ | Posterior variance based on GP for $f(\boldsymbol{x}, \boldsymbol{w})$ |
| $u_t(\boldsymbol{x}, \boldsymbol{w})$ | Upper confidence bound for $f(\boldsymbol{x}, \boldsymbol{w})$ |
| $l_t(\boldsymbol{x}, \boldsymbol{w})$ | Lower confidence bound for $f(\boldsymbol{x}, \boldsymbol{w})$ |
| $\beta_t$ | Parameter for adjusting the confidence bound width |
| $\text{ucb}_t(\boldsymbol{x})$ | Upper confidence bound for $F(\boldsymbol{x})$ |
| $\text{lcb}_t(\boldsymbol{x})$ | Lower confidence bound for $F(\boldsymbol{x})$ |
| $\hat{\boldsymbol{x}}_t$ | Estimated solution for the optimal solution defined by $\hat{\boldsymbol{x}}_t = \arg\max_{\boldsymbol{x} \in \mathcal{X}} \rho(\mu_{t-1}(\boldsymbol{x}, \cdot))$ |
| $(\cdot)_+$ | Operator defined by $(a)_+ = \max\{a, 0\}$ |
| $\tilde{\boldsymbol{x}}_t$ | Optimistic maximum solution defined by $\tilde{\boldsymbol{x}}_t = \arg\max_{\boldsymbol{x} \in \mathcal{X}}(\text{ucb}_{t-1}(\boldsymbol{x}) - \max_{\boldsymbol{x} \in \mathcal{X}} \text{lcb}_{t-1}(\boldsymbol{x}))_+$ |
| $\boldsymbol{x}_t$ | Next design variable to be evaluated |
| $\xi_t$ | Random variable following the chi-squared distribution with two digrees of freedom |
| $\boldsymbol{w}_t$ | Next environmental variable to be evaluated |
| $r_t$ | Instantaneous regret regarding the optimal solution and estimated solution |
| $R_t$ | Cumulative instantaneous regret regarding the optimal solution and estimated solution |
| $\gamma_t$ | Maximum information gain |
| $\mathbb{E}_{t-1}[\cdot]$ | Conditional expectation given the dataset $D_{t-1} = \{(\boldsymbol{x}_1, \boldsymbol{w}_1, \varepsilon_1, \beta_1), \ldots, (\boldsymbol{x}_{t-1}, \boldsymbol{w}_{t-1}, \varepsilon_{t-1}, \beta_{t-1})\}$ |
| $\hat{t}$ | Optimal index given by $\hat{t} = \arg\max_{1 \le i \le t} \mathbb{E}_{t-1}[F(\hat{\boldsymbol{x}}_i)]$ |

---

**Algorithm 1** RRGP-UCB for robustness measures.

---

**Input:** GP prior $\mathcal{GP}(0, k)$
  **for** $t = 1, 2, \ldots$ **do**
    Generate $\xi_t$ from chi-squared distribution with two degrees of freedom
    Compute $\beta_t = 2\log(|\mathcal{X} \times \Omega|) + \xi_t$
    Compute $Q_{t-1}(\boldsymbol{x}, \boldsymbol{w})$ for each $(\boldsymbol{x}, \boldsymbol{w}) \in \mathcal{X} \times \Omega$
    Compute $Q_{t-1}(\boldsymbol{x})$ for each $\boldsymbol{x} \in \mathcal{X}$
    Estimate $\hat{\boldsymbol{x}}_t$ by $\hat{\boldsymbol{x}}_t = \arg\max_{\boldsymbol{x} \in \mathcal{X}} \rho(\mu_{t-1}(\boldsymbol{x}, \cdot))$
    Select next evaluation point $\boldsymbol{x}_t$ by equation 4
    Select next evaluation point $\boldsymbol{w}_t$ by equation 5
    Observe $y_t = f(\boldsymbol{x}_t, \boldsymbol{w}_t) + \varepsilon_t$ at point $(\boldsymbol{x}_t, \boldsymbol{w}_t)$
    Update GP by adding observed data
  **end for**

---

### 3.1 Credible Interval for Robustness Measures

For each $(\boldsymbol{x}, \boldsymbol{w}) \in \mathcal{X} \times \Omega$ and $t \geq 1$, let $Q_{t-1}(\boldsymbol{x}, \boldsymbol{w}) = [l_{t-1}(\boldsymbol{x}, \boldsymbol{w}), u_{t-1}(\boldsymbol{x}, \boldsymbol{w})]$ denote a credible interval for $f(\boldsymbol{x}, \boldsymbol{w})$, where $l_{t-1}(\boldsymbol{x}, \boldsymbol{w})$ and $u_{t-1}(\boldsymbol{x}, \boldsymbol{w})$ are given by

$$l_{t-1}(\boldsymbol{x}, \boldsymbol{w}) = \mu_{t-1}(\boldsymbol{x}, \boldsymbol{w}) - \beta_t^{1/2} \sigma_{t-1}(\boldsymbol{x}, \boldsymbol{w}), \ u_{t-1}(\boldsymbol{x}, \boldsymbol{w}) = \mu_{t-1}(\boldsymbol{x}, \boldsymbol{w}) + \beta_t^{1/2} \sigma_{t-1}(\boldsymbol{x}, \boldsymbol{w}).$$

Here, $\beta_t \geq 0$ is a user-defined trade-off parameter. Due to the properties of GPs, the posterior distribution of $f(\boldsymbol{x}, \boldsymbol{w})$ after observing data is a normal distribution with mean $\mu_{t-1}(\boldsymbol{x}, \boldsymbol{w})$ and variance $\sigma_{t-1}^2(\boldsymbol{x}, \boldsymbol{w})$. Therefore, by selecting an appropriate value of $\beta_t$ [2], the interval $Q_{t-1}(\boldsymbol{x}, \boldsymbol{w})$ contains $f(\boldsymbol{x}, \boldsymbol{w})$ with high probability. Next, for each $\boldsymbol{x}$, we define $G_{t-1}(\boldsymbol{x})$, the set of functions over $\boldsymbol{w}$, as follows:

$$G_{t-1}(\boldsymbol{x}) = \{g(\boldsymbol{x}, \cdot) \mid \text{for any } \boldsymbol{w} \in \Omega, g(\boldsymbol{x}, \boldsymbol{w}) \in Q_{t-1}(\boldsymbol{x}, \boldsymbol{w})\}.$$

If $Q_{t-1}(\boldsymbol{x}, \boldsymbol{w})$ is a high-probability credible interval for $f(\boldsymbol{x}, \boldsymbol{w})$ for all $\boldsymbol{w} \in \Omega$, then the function $f(\boldsymbol{x}, \cdot)$ lies in $G_{t-1}(\boldsymbol{x})$ with high probability. Therefore, the following inequality holds with high probability:

$$\inf_{g(\boldsymbol{x}, \cdot) \in G_{t-1}(\boldsymbol{x})} \rho(g(\boldsymbol{x}, \cdot)) \leq \rho(f(\boldsymbol{x}, \cdot)) = F(\boldsymbol{x}) \leq \sup_{g(\boldsymbol{x}, \cdot) \in G_{t-1}(\boldsymbol{x})} \rho(g(\boldsymbol{x}, \cdot)). \tag{1}$$

We can thus construct a high-probability credible interval for $F(\boldsymbol{x})$ using the left- and right-hand sides of equation 1. However, computing the exact bounds in equation 1 is generally intractable. To address this, we introduce the lower bound $\mathrm{lcb}_{t-1}(\boldsymbol{x})$ and upper bound $\mathrm{ucb}_{t-1}(\boldsymbol{x})$, which satisfy:

$$\mathrm{lcb}_{t-1}(\boldsymbol{x}) \leq \inf_{g(\boldsymbol{x}, \cdot) \in G_{t-1}(\boldsymbol{x})} \rho(g(\boldsymbol{x}, \cdot)), \quad \sup_{g(\boldsymbol{x}, \cdot) \in G_{t-1}(\boldsymbol{x})} \rho(g(\boldsymbol{x}, \cdot)) \leq \mathrm{ucb}_{t-1}(\boldsymbol{x}). \tag{2}$$

Inatsu et al. (2024a) showed that, for commonly used robustness measures, including expectation, the bounds $\mathrm{lcb}_{t-1}(\boldsymbol{x})$ and $\mathrm{ucb}_{t-1}(\boldsymbol{x})$ can be analytically calculated using $l_{t-1}(\boldsymbol{x}, \boldsymbol{w})$ and $u_{t-1}(\boldsymbol{x}, \boldsymbol{w})$ [3]. Inatsu et al. (2024a) differs from this study in that it deals with multi-objective optimization for robustness measures under input uncertainty and adopts the assumption that black-box functions are in RKHS. On the other hand, it shares commonalities with this study in that it adopts a GP as a surrogate model and calculates upper and lower bounds for robustness measures based on equation 2. Therefore, the computational methods for $\mathrm{lcb}_{t-1}(\boldsymbol{x})$ and $\mathrm{ucb}_{t-1}(\boldsymbol{x})$ presented in Inatsu et al. (2024a) for various robustness measures are applicable to the setting of this study. Representative robustness measures and their corresponding $\mathrm{lcb}_{t-1}(\boldsymbol{x})$ and $\mathrm{ucb}_{t-1}(\boldsymbol{x})$ are summarized in Table 4. On the other hand, even for robustness measures not listed in Table 4, it is possible to approximate $\mathrm{lcb}_{t-1}(\boldsymbol{x})$ and $\mathrm{ucb}_{t-1}(\boldsymbol{x})$ by sampling. First, we generate $S$ sample paths $f^{(1)}(\boldsymbol{x}, \boldsymbol{w}), \ldots, f^{(S)}(\boldsymbol{x}, \boldsymbol{w})$ from the posterior distribution of the function $f(\boldsymbol{x}, \boldsymbol{w})$ and then estimating $\mathrm{lcb}_{t-1}(\boldsymbol{x})$ and $\mathrm{ucb}_{t-1}(\boldsymbol{x})$ as follows:

$$\mathrm{lcb}_{t-1}(\boldsymbol{x}) = \min_{1 \leq s \leq S} \rho(f^{(s)}(\boldsymbol{x}, \cdot)), \ \mathrm{ucb}_{t-1}(\boldsymbol{x}) = \max_{1 \leq s \leq S} \rho(f^{(s)}(\boldsymbol{x}, \cdot)).$$

Using these bounds, we define the credible interval $Q_{t-1}(\boldsymbol{x}) = [\mathrm{lcb}_{t-1}(\boldsymbol{x}), \mathrm{ucb}_{t-1}(\boldsymbol{x})]$ for $F(\boldsymbol{x})$.

### 3.2 Estimation of Optimal Solution

We present a method for estimating the optimal solution $\boldsymbol{x}^*$ at each iteration $t \geq 1$. Recall that in this setting, the value of the objective function $F(\boldsymbol{x})$ is unobservable; we can only access noisy observations of $f(\boldsymbol{x}, \boldsymbol{w})$. As a result, directly estimating $\boldsymbol{x}^*$ from the observed data is not possible. To address this, we define the estimated solution $\hat{\boldsymbol{x}}_t$ based on the estimate of $F(\boldsymbol{x})$ calculated from the posterior mean of $f$ as follows:

$$\hat{\boldsymbol{x}}_t = \arg\max_{\boldsymbol{x} \in \mathcal{X}} \rho(\mu_{t-1}(\boldsymbol{x}, \cdot)). \tag{3}$$

---

[2] For example, if $\beta_t^{1/2} = 1.96$, then $f(\boldsymbol{x}, \boldsymbol{w}) \in Q_{t-1}(\boldsymbol{x}, \boldsymbol{w})$ holds with probability 0.95.

[3] In Tables 3 and 4 in Inatsu et al. (2024a), the terms "risk measure" and "Bayes risk" are used instead of "robustness measure" and "expectation measure," respectively.

Table 4: Values of $\mathrm{lcb}_t(\boldsymbol{x})$ and $\mathrm{ucb}_t(\boldsymbol{x})$ for commonly used robustness measures (Table 3 in Inatsu et al. (2024a)).

| Robustness measure | Definition | $\mathrm{lcb}_t(\boldsymbol{x})$ | $\mathrm{ucb}_t(\boldsymbol{x})$ |
|---|---|---|---|
| Expectation | $\mathbb{E}[f_{\boldsymbol{x},\boldsymbol{w}}]$ | $\mathbb{E}[l_{t,\boldsymbol{x},\boldsymbol{w}}]$ | $\mathbb{E}[u_{t,\boldsymbol{x},\boldsymbol{w}}]$ |
| Worst-case | $\inf_{\boldsymbol{w}\in\Omega} f_{\boldsymbol{x},\boldsymbol{w}}$ | $\inf_{\boldsymbol{w}\in\Omega} l_{t,\boldsymbol{x},\boldsymbol{w}}$ | $\inf_{\boldsymbol{w}\in\Omega} u_{t,\boldsymbol{x},\boldsymbol{w}}$ |
| Best-case | $\sup_{\boldsymbol{w}\in\Omega} f_{\boldsymbol{x},\boldsymbol{w}}$ | $\sup_{\boldsymbol{w}\in\Omega} l_{t,\boldsymbol{x},\boldsymbol{w}}$ | $\sup_{\boldsymbol{w}\in\Omega} u_{t,\boldsymbol{x},\boldsymbol{w}}$ |
| $\alpha$-value-at-risk | $\inf\{b\in\mathbb{R}\mid\alpha\le\mathbb{P}(f_{\boldsymbol{x},\boldsymbol{w}}\le b)\}$ | $\inf\{b\in\mathbb{R}\mid\alpha\le\mathbb{P}(l_{t,\boldsymbol{x},\boldsymbol{w}}\le b)\}$ | $\inf\{b\in\mathbb{R}\mid\alpha\le\mathbb{P}(u_{t,\boldsymbol{x},\boldsymbol{w}}\le b)\}$ |
| $\alpha$-conditional value-at-risk | $\mathbb{E}[f_{\boldsymbol{x},\boldsymbol{w}}\mid f_{\boldsymbol{x},\boldsymbol{w}}\le v_f(\boldsymbol{x};\alpha)]$ | $\frac{1}{\alpha}\int_0^\alpha v_{l_t}(\boldsymbol{x};\alpha')\mathrm{d}\alpha'$ | $\frac{1}{\alpha}\int_0^\alpha v_{u_t}(\boldsymbol{x};\alpha')\mathrm{d}\alpha'$ |
| Mean absolute deviation | $\mathbb{E}[|f_{\boldsymbol{x},\boldsymbol{w}}-\mathbb{E}[f_{\boldsymbol{x},\boldsymbol{w}}]|]$ | $\mathbb{E}[\min\{|\check{l}_{t,\boldsymbol{x},\boldsymbol{w}}|,|\check{u}_{t,\boldsymbol{x},\boldsymbol{w}}|\}-\mathrm{STR}(\check{l}_{t,\boldsymbol{x},\boldsymbol{w}},\check{u}_{t,\boldsymbol{x},\boldsymbol{w}})]$ | $\mathbb{E}[\max\{|\check{l}_{t,\boldsymbol{x},\boldsymbol{w}}|,|\check{u}_{t,\boldsymbol{x},\boldsymbol{w}}|\}]$ |
| Standard deviation | $\sqrt{\mathbb{E}[|f_{\boldsymbol{x},\boldsymbol{w}}-\mathbb{E}[f_{\boldsymbol{x},\boldsymbol{w}}]|^2]}$ | $\sqrt{\mathbb{E}[\min\{|\check{l}_{t,\boldsymbol{x},\boldsymbol{w}}|^2,|\check{u}_{t,\boldsymbol{x},\boldsymbol{w}}|^2\}-\mathrm{STR}^2(\check{l}_{t,\boldsymbol{x},\boldsymbol{w}},\check{u}_{t,\boldsymbol{x},\boldsymbol{w}})]}$ | $\sqrt{\mathbb{E}[\max\{|\check{l}_{t,\boldsymbol{x},\boldsymbol{w}}|^2,|\check{u}_{t,\boldsymbol{x},\boldsymbol{w}}|^2\}]}$ |
| Variance | $\mathbb{E}[|f_{\boldsymbol{x},\boldsymbol{w}}-\mathbb{E}[f_{\boldsymbol{x},\boldsymbol{w}}]|^2]$ | $\mathbb{E}[\min\{|\check{l}_{t,\boldsymbol{x},\boldsymbol{w}}|^2,|\check{u}_{t,\boldsymbol{x},\boldsymbol{w}}|^2\}-\mathrm{STR}^2(\check{l}_{t,\boldsymbol{x},\boldsymbol{w}},\check{u}_{t,\boldsymbol{x},\boldsymbol{w}})]$ | $\mathbb{E}[\max\{|\check{l}_{t,\boldsymbol{x},\boldsymbol{w}}|^2,|\check{u}_{t,\boldsymbol{x},\boldsymbol{w}}|^2\}]$ |
| Distributionally robust | $\inf_{P\in\mathcal{A}} F(\boldsymbol{x};P)$ | $\inf_{P\in\mathcal{A}} \mathrm{lcb}_t(\boldsymbol{x};P)$ | $\inf_{P\in\mathcal{A}} \mathrm{ucb}_t(\boldsymbol{x};P)$ |
| Monotonic Lipschitz map | $\mathcal{M}(F(\boldsymbol{x}))$ | $\min\{\mathcal{M}(\mathrm{lcb}_t(\boldsymbol{x})),\mathcal{M}(\mathrm{ucb}_t(\boldsymbol{x}))\}$ | $\max\{\mathcal{M}(\mathrm{lcb}_t(\boldsymbol{x})),\mathcal{M}(\mathrm{ucb}_t(\boldsymbol{x}))\}$ |
| Weighted sum | $\alpha_1 F^{(m_1)}(\boldsymbol{x})+\alpha_2 F^{(m_2)}(\boldsymbol{x})$ | $\alpha_1 \mathrm{lcb}_t^{(m_1)}(\boldsymbol{x})+\alpha_2 \mathrm{lcb}_t^{(m_2)}(\boldsymbol{x})$ | $\alpha_1 \mathrm{ucb}_t^{(m_1)}(\boldsymbol{x})+\alpha_2 \mathrm{ucb}_t^{(m_2)}(\boldsymbol{x})$ |
| Probabilistic threshold | $\mathbb{P}(f_{\boldsymbol{x},\boldsymbol{w}}\ge\theta)$ | $\mathbb{P}(l_{t,\boldsymbol{x},\boldsymbol{w}}\ge\theta)$ | $\mathbb{P}(u_{t,\boldsymbol{x},\boldsymbol{w}}\ge\theta)$ |

$f_{\boldsymbol{x},\boldsymbol{w}}\equiv f(\boldsymbol{x},\boldsymbol{w}),\ l_{t,\boldsymbol{x},\boldsymbol{w}}\equiv l_t(\boldsymbol{x},\boldsymbol{w}),\ u_{t,\boldsymbol{x},\boldsymbol{w}}\equiv u_t(\boldsymbol{x},\boldsymbol{w}),\ v_f(\boldsymbol{x};\alpha)\equiv\inf\{b\in\mathbb{R}\mid\mathbb{P}(f_{\boldsymbol{x},\boldsymbol{w}}\le b)\ge\alpha\}$

$v_{l_t}(\boldsymbol{x};\alpha)\equiv\inf\{b\in\mathbb{R}\mid\mathbb{P}(l_{t,\boldsymbol{x},\boldsymbol{w}}\le b)\ge\alpha\},\ v_{u_t}(\boldsymbol{x};\alpha)\equiv\inf\{b\in\mathbb{R}\mid\mathbb{P}(u_{t,\boldsymbol{x},\boldsymbol{w}}\le b)\ge\alpha\},\ \alpha\in(0,1)$

$\check{l}_{t,\boldsymbol{x},\boldsymbol{w}}\equiv l_{t,\boldsymbol{x},\boldsymbol{w}}-\mathbb{E}[u_{t,\boldsymbol{x},\boldsymbol{w}}],\ \check{u}_{t,\boldsymbol{x},\boldsymbol{w}}\equiv u_{t,\boldsymbol{x},\boldsymbol{w}}-\mathbb{E}[l_{t,\boldsymbol{x},\boldsymbol{w}}]\ ,\ \mathrm{STR}(a,b)\equiv\max\{\min\{-a,b\},0\}$

$F(\boldsymbol{x};P)$: Robustness measure $F(\boldsymbol{x})$ defined based on the distribution $P$

$\mathrm{lcb}_t(\boldsymbol{x};P),\mathrm{ucb}_t(\boldsymbol{x};P)$: $\mathrm{lcb}_t(\boldsymbol{x})$ and $\mathrm{ucb}_t(\boldsymbol{x})$ for $F(\boldsymbol{x};P)$

$\mathrm{lcb}_t^{(m)}(\boldsymbol{x}),\mathrm{ucb}_t^{(m)}(\boldsymbol{x})$: $\mathrm{lcb}_t(\boldsymbol{x})$ and $\mathrm{ucb}_t(\boldsymbol{x})$ for $F^{(m)}(\boldsymbol{x})$

$\mathcal{M}(\cdot)$: Monotonic Lipschitz continuous map with Lipschitz constant $K$

$\alpha_1,\alpha_2\ge 0$

Expectation and probability are taken with respect to distribution of $\boldsymbol{w}$, and $\alpha$-value-at-risk is same meaning as $\alpha$-quantile

## 3.3 Acquisition Function

We introduce acquisition functions to determine the next evaluation point $(\boldsymbol{x}_t,\boldsymbol{w}_t)$. In this study, the estimated solution $\hat{\boldsymbol{x}}_t$ and the next evaluation point $\boldsymbol{x}_t$ are not necessarily identical. In previous studies on BO under robustness considerations and environmental variability (Kirschner et al., 2020; Inatsu et al., 2024a), $\boldsymbol{x}_t$ is typically chosen based on the upper bound of a credible interval for a robustness measure, while $\boldsymbol{w}_t$ is selected to maximize the posterior variance of $f(\boldsymbol{x}_t,\boldsymbol{w})$. We propose a modification to the method for selecting $\boldsymbol{x}_t$ and partially adopt the aforementioned approach. In particular, as $\boldsymbol{w}_t$ is selected based on the posterior variance of $f$, this eliminates the need for hyperparameter tuning in the acquisition function, unlike the credible interval for $F(\boldsymbol{x})$, which depends on a user-specified parameter $\beta_t$. To address this, we avoid fixing $\beta_t$ explicitly and instead treat it as a realization from a probability distribution. For example, Takeno et al. (2023) proposed IRGP-UCB within the standard BO framework, in which $\beta_t$ for GP-UCB is randomly sampled from a two-parameter exponential distribution. Similarly, Inatsu et al. (2024b) proposed the randomized straddle method for level-set estimation, where $\beta_t$ in the straddle acquisition function (Bryan et al., 2005) is drawn from a chi-squared distribution. These methods remove the need to manually specify hyperparameters and also yield tighter theoretical guarantees than conventional GP-UCB or straddle methods. However, these algorithms are designed for problems in which the target function $f$ itself is modeled as a GP, and their theoretical guarantees critically depend on $f$ following GPs. In contrast, the target function in our setting is $F(\boldsymbol{x})$, which generally does not follow a GP, even if $f$ does. Therefore, to derive a theoretically sound acquisition strategy, we modify the GP-UCB method to suit the context in which $\boldsymbol{x}_t$ is selected based on the upper bound of the credible interval for $F(\boldsymbol{x})$. Before introducing this method, we reformulate standard GP-UCB in the conventional BO setting without input uncertainty and highlight its essential structural properties. The following lemma characterizes GP-UCB in this simplified setting:

**Lemma 3.1.** For a black-box function $f(\boldsymbol{x})$ modeled as a GP, let $\mu_{t-1}(\boldsymbol{x})$ denote the posterior mean, $\sigma_{t-1}^2(\boldsymbol{x})$ the posterior variance, and $\beta_t\ge 0$ a user-defined parameter. Define $\mathrm{u}_{t-1}(\boldsymbol{x})=\mu_{t-1}(\boldsymbol{x})+\beta_t^{1/2}\sigma_{t-1}(\boldsymbol{x})$ and $\mathrm{l}_{t-1}(\boldsymbol{x})=\mu_{t-1}(\boldsymbol{x})-\beta_t^{1/2}\sigma_{t-1}(\boldsymbol{x})$, and

$$\boldsymbol{x}_t^{(\mathrm{u})}=\arg\max_{\boldsymbol{x}\in\mathcal{X}}\mathrm{u}_{t-1}(\boldsymbol{x}),\ \tilde{\boldsymbol{x}}_t^{(f)}=\arg\max_{\boldsymbol{x}\in\mathcal{X}}(\mathrm{u}_{t-1}(\boldsymbol{x})-\max_{\boldsymbol{x}\in\mathcal{X}}\mathrm{l}_{t-1}(\boldsymbol{x}))_+,$$

$$\hat{\boldsymbol{x}}_t^{(f)}=\arg\max_{\boldsymbol{x}\in\mathcal{X}}\mu_{t-1}(\boldsymbol{x}),\ \boldsymbol{x}_t^{(f)}=\arg\max_{\boldsymbol{x}\in\{\tilde{\boldsymbol{x}}_t^{(f)},\hat{\boldsymbol{x}}_t^{(f)}\}}(\mathrm{u}_{t-1}(\boldsymbol{x})-\mathrm{l}_{t-1}(\boldsymbol{x}),$$

where $(a)_+$ denotes $a$ if $a > 0$, and otherwise is 0. Then, the equality $\mathrm{u}_{t-1}(\boldsymbol{x}_t^{(\mathrm{u})}) = \mathrm{u}_{t-1}(\boldsymbol{x}_t^{(f)}))$ holds.

The proof is provided in Appendix C. Using this lemma, we define the selection rule for $\boldsymbol{x}_t$ as follows.

**Definition 3.1** (Selection rule for $\boldsymbol{x}_t$). Let $\xi_1, \ldots, \xi_t$ be independent random variables drawn from the chi-squared distribution with two degrees of freedom, where $f, \varepsilon_1, \ldots, \varepsilon_t, \xi_1, \ldots, \xi_t$ are mutually independent. Define $\beta_t = 2\log(|\mathcal{X} \times \Omega|) + \xi_t$. Assume that $\mathrm{lcb}_{t-1}(\boldsymbol{x})$ and $\mathrm{ucb}_{t-1}(\boldsymbol{x})$ are lower and upper bounds that satisfy equation 2, and let $\hat{\boldsymbol{x}}_t$ be defined by equation 3. Then, $\boldsymbol{x}_t$ is selected as follows:

$$\boldsymbol{x}_t = \underset{\boldsymbol{x} \in \{\tilde{\boldsymbol{x}}_t, \hat{\boldsymbol{x}}_t\}}{\arg\max} (\mathrm{ucb}_{t-1}(\boldsymbol{x}) - \mathrm{lcb}_{t-1}(\boldsymbol{x})), \tag{4}$$

where $\tilde{\boldsymbol{x}}_t = \arg\max_{\boldsymbol{x} \in \mathcal{X}}(\mathrm{ucb}_{t-1}(\boldsymbol{x}) - \max_{\boldsymbol{x} \in \mathcal{X}} \mathrm{lcb}_{t-1}(\boldsymbol{x}))_+$.

Here, Lemma 3.1 provides a different perspective on Definition 3.1. The acquisition function in Definition 3.1 appears, at first glance, to be unrelated to the usual GP-UCB. However, the usual GP-UCB can be expressed in the same form as Lemma 3.1, and since this equivalent expression is similar to Definition 3.1, the proposed acquisition function can also be interpreted as an acquisition function based on GP-UCB. Finally, based on $\boldsymbol{x}_t$ selected by equation 4, we determine $\boldsymbol{w}_t$ as follows.

**Definition 3.2** (Selection rule for $\boldsymbol{w}_t$). The next environmental variable $\boldsymbol{w}_t$ is selected as follows:

$$\boldsymbol{w}_t = \underset{\boldsymbol{w} \in \Omega}{\arg\max} \, \sigma_{t-1}^2(\boldsymbol{x}_t, \boldsymbol{w}), \tag{5}$$

where $\boldsymbol{x}_t$ is given by equation 4.

## 4 Theoretical Analysis

In this section, we provide theoretical guarantees on the expected regret of the proposed algorithm. Detailed proofs are given in Appendix C. To evaluate the quality of the estimated solution, we define the instantaneous regret $r_t$ and cumulative regret $R_t$ as follows:

$$r_t = F(\boldsymbol{x}^*) - F(\hat{\boldsymbol{x}}_t), \ R_t = \sum_{i=1}^t \{F(\boldsymbol{x}^*) - F(\hat{\boldsymbol{x}}_i)\} = \sum_{i=1}^t r_i.$$

In addition, to derive theoretical guarantees for the proposed method, we introduce the concept of maximum information gain $\gamma_t$. This quantity is widely used in the theoretical analysis of GP-based BO and level-set estimation (Srinivas et al., 2010; Bogunovic et al., 2016; Gotovos et al., 2013), and is expressed as follows:

$$\gamma_t = \frac{1}{2} \sup_{\{(\boldsymbol{x}_{(1)}, \boldsymbol{w}_{(1)}), \ldots, (\boldsymbol{x}_{(t)}, \boldsymbol{w}_{(t)})\} \subset \mathcal{X} \times \Omega} \log\det(\boldsymbol{I}_t + \sigma_{\mathrm{noise}}^{-2} \tilde{\boldsymbol{K}}_t),$$

where $(\boldsymbol{x}_{(1)}, \boldsymbol{w}_{(1)}), \ldots, (\boldsymbol{x}_{(t)}, \boldsymbol{w}_{(t)})$ are arbitrary elements of $\mathcal{X} \times \Omega$, $\tilde{\boldsymbol{K}}_t$ is a $t \times t$ kernel matrix with the $(i, j)$-th entry given by $k((\boldsymbol{x}_{(i)}, \boldsymbol{w}_{(i)}), (\boldsymbol{x}_{(j)}, \boldsymbol{w}_{(j)}))$. For commonly used kernels, such as the linear, Gaussian, and Matérn kernels, $\gamma_t$ is known to grow sublinearly under mild conditions (see, e.g., Theorem 5 in Srinivas et al. (2010)). Let $h(a) : [0, \infty) \to [0, \infty)$ be a non-decreasing, concave function satisfying $h(0) = 0$, and denote by $\mathcal{H}$ the set of all $h(a)$. We then define a class of functions $q(a) : [0, \infty) \to [0, \infty)$, denoted by $\mathcal{Q}$, as follows:

$$\mathcal{Q} = \left\{ q(a) = \sum_{i=1}^n \zeta_i h_i \left( \sum_{j=1}^{s_i} \lambda_{ij} a^{\nu_{ij}} \right) \mid n, s_i \in \mathbb{N}, \zeta_i, \lambda_{ij}, \geq 0, \nu_{ij} > 0, h_i(\cdot) \in \mathcal{H} \right\}.$$

For $\mathcal{Q}$, we impose the following assumption on $\mathrm{ucb}_{t-1}(\boldsymbol{x}_t)$, $\mathrm{lcb}_{t-1}(\boldsymbol{x}_t)$, and the width term $2\beta_t^{1/2}\sigma_{t-1}(\boldsymbol{x}_t, \boldsymbol{w}_t)$.

**Assumption 4.1.** There exists a function $q(x) \in \mathcal{Q}$ such that for any $t \geq 1, \boldsymbol{x}_t, \beta_t$, and $\sigma_{t-1}(\boldsymbol{x}_t, \boldsymbol{w})$, the following inequality holds:

$$\mathrm{ucb}_{t-1}(\boldsymbol{x}_t) - \mathrm{lcb}_{t-1}(\boldsymbol{x}_t) \leq q(2\beta_t^{1/2} \max_{\boldsymbol{w} \in \Omega} \sigma_{t-1}(\boldsymbol{x}_t, \boldsymbol{w})). \tag{6}$$

Table 5: Specific forms of $q(a)$ for commonly used robustness measures (modified version of Table 4 in Inatsu et al. (2024a)).

| Robustness measure | Definition | $q(a)$ |
|---|---|---|
| Expectation | $\mathbb{E}[f_{\boldsymbol{x},\boldsymbol{w}}]$ | $a$ |
| Worst-case | $\inf_{\boldsymbol{w}\in\Omega} f_{\boldsymbol{x},\boldsymbol{w}}$ | $a$ |
| Best-case | $\sup_{\boldsymbol{w}\in\Omega} f_{\boldsymbol{x},\boldsymbol{w}}$ | $a$ |
| $\alpha$-value-at-risk | $\inf\{b\in\mathbb{R} \mid \alpha \leq \mathbb{P}(f_{\boldsymbol{x},\boldsymbol{w}} \leq b)\}$ | $a$ |
| $\alpha$-conditional value-at-risk | $\mathbb{E}[f_{\boldsymbol{x},\boldsymbol{w}} \mid f_{\boldsymbol{x},\boldsymbol{w}} \leq v_f(\boldsymbol{x};\alpha)]$ | $a$ |
| Mean absolute deviation | $\mathbb{E}[\lvert f_{\boldsymbol{x},\boldsymbol{w}} - \mathbb{E}[f_{\boldsymbol{x},\boldsymbol{w}}]\rvert]$ | $2a$ |
| Distributionally robust | $\inf_{P\in\mathcal{A}} F(\boldsymbol{x};P)$ | $q(a;F)$ |
| Monotonic Lipschitz map | $\mathcal{M}(F^{(m)}(\boldsymbol{x}))$ | $Kq^{(m)}(a)$ |
| Weighted sum | $\alpha_1 F^{(m_1)}(\boldsymbol{x}) + \alpha_2 F^{(m_2)}(\boldsymbol{x})$ | $\alpha_1 q^{(m_1)}(a) + \alpha_2 q^{(m_2)}(a)$ |

$f_{\boldsymbol{x},\boldsymbol{w}} \equiv f(\boldsymbol{x},\boldsymbol{w}),\ v_f(\boldsymbol{x};\alpha) \equiv \inf\{b\in\mathbb{R} \mid \mathbb{P}(f_{\boldsymbol{x},\boldsymbol{w}} \leq b) \geq \alpha\},\ \alpha \in (0,1)$
$F(\boldsymbol{x};P)$: Robustness measure $F(\boldsymbol{x})$ defined based on distribution $P$
$q(a;F)$: function $q(a)$ for $F(\boldsymbol{x})$ satisfying Assumption 4.1
$\mathcal{M}(\cdot)$: Monotonic Lipschitz continuous map with Lipschitz constant $K$
$q^{(\cdot)}(a)$: function $q(a)$ for $F^{(\cdot)}(\boldsymbol{x})$ satisfying Assumption 4.1
$\alpha_1, \alpha_2 \geq 0$
Expectation and probability are taken with respect to distribution of $\boldsymbol{w}$,
and $\alpha$-value-at-risk is same meaning as $\alpha$-quantile

In this study, $q(a)$ plays the same role as $q(a)$ in Inequality (3) of Inatsu et al. (2024a), but while their $q(a)$ does not have any assumptions regarding concave functions, ours does. The reason for this is that, unlike Inatsu et al. (2024a), we are subjecting the expected value of (cumulative) regret to theoretical analysis. This analysis requires an inequality evaluation of the expected value, which they did not need to perform. In particular, we need to use Jensen's inequality, which requires us to impose additional assumptions regarding concave functions. Similarly to equation 2, it also shares commonalities with Inatsu et al. (2024a) in terms of requiring $q(a)$ satisfying equation 6. They provide $q(a)$ for various robustness measures, including the expectation measure. In this study, we impose stronger assumptions on $q(a)$ than those in their; however, since the $q(a)$ provided by Inatsu et al. (2024a) satisfies Assumption 4.1 and $q(a) \in \mathcal{Q}$, their results can be applied. Representative robustness measures and their corresponding $q(a)$ functions are summarized in Table 5.[4] Then, the following theorem holds.

**Theorem 4.1.** Assume that equation 2, the regularity assumption, and Assumption 4.1 hold. Suppose that $\xi_1, \ldots, \xi_t$ are independent random variables following the chi-squared distribution with two degrees of freedom, and that $f, \varepsilon_1, \ldots, \varepsilon_t, \xi_1, \ldots, \xi_t$ are mutually independent. Define $\beta_t = 2\log(|\mathcal{X} \times \Omega|) + \xi_t$. Let $q(a) = \sum_{i=1}^{n} \zeta_i h_i\left(\sum_{j=1}^{s_i} \lambda_{ij} a^{\nu_{ij}}\right)$ be a function satisfying Assumption 4.1. Then, if Algorithm 1 is performed, the following inequality holds:

$$\mathbb{E}[R_t] \leq 2t \sum_{i=1}^{n} \zeta_i h_i\left(\frac{1}{t}\sum_{j=1}^{s_i} 2^{\nu_{ij}} \lambda_{ij} \left(tC_{2,\nu_{ij}}\right)^{1-\nu'_{ij}/2} \left(C_1\gamma_t\right)^{\nu'_{ij}/2}\right),$$

where $\nu'_{ij} = \min\{\nu_{ij}, 1\}, C_1 = \frac{2}{\log(1+\sigma_{\text{noise}}^{-2})}, C_{2,\nu_{ij}} = \mathbb{E}[\beta_t^{\nu_{ij}/(2-\nu'_{ij})}]$. The expectation is taken over all sources of randomness, including $f, \varepsilon_1, \ldots, \varepsilon_t, \beta_1, \ldots, \beta_t$.

If $\gamma_t$ is a sublinear function, the following convergence holds:

$$\lim_{t\to\infty} \frac{1}{t}\sum_{j=1}^{s_i} 2^{\nu_{ij}} \lambda_{ij} \left(tC_{2,\nu_{ij}}\right)^{1-\nu'_{ij}/2} \left(C_1\gamma_t\right)^{\nu'_{ij}/2} = 0. \tag{7}$$

---

[4]Inatsu et al. (2024a) derives $q(a)$ for measures such as the standard deviation and variance, but these rely on constants dependent on the norm in the RKHS, which differ from our setting and are thus excluded. The probabilistic threshold is also omitted as no explicit form of $q(a)$ is provided.

Then, if each $h_i(a)$ is continuous at 0, it follows from Theorem 4.1 that the $\mathbb{E}[R_t]/t$ satisfies

$$\lim_{t \to \infty} \frac{\mathbb{E}[R_t]}{t} = 0. \tag{8}$$

As a special case, according to Table 4 in Inatsu et al. (2024a), $q(a)$ corresponding to the expectation measure satisfies $q(a) = a$; i.e., $n = \zeta_i = s_i = \lambda_{ij} = \nu_{ij} = 1$ and $h_1(a) = a$. In this case, $\mathbb{E}[R_t]$ satisfies

$$\mathbb{E}[R_t] \le 4\sqrt{tC_{2,1}C_1\gamma_t} = 4\sqrt{C_1(2\log(|\mathcal{X} \times \Omega|) + 2)t\gamma_t}.$$

Similarly, $q(a)$ corresponding to the worst-case, best-case, $\alpha$-value-at-risk, $\alpha$-conditional value-at-risk, and mean absolute deviation measures satisfies $q(a) = 2a$ for the mean absolute deviation measure and $q(a) = a$ for the other measures. For these robustness measures, including the expectation measure, the following corollary holds:

**Corollary 4.1.** Under the assumptions of Theorem 4.1, the following inequality holds for the expectation, worst-case, best-case, $\alpha$-value-at-risk, $\alpha$-conditional value-at-risk, and mean absolute deviation measures:

$$\mathbb{E}[R_t] \le C\sqrt{tC_0\gamma_t},$$

where $C_0 = 2(2\log(|\mathcal{X} \times \Omega|) + 2)/\log(1 + \sigma_{\text{noise}}^{-2})$, and $C$ is 8 for the mean absolute deviation measure and 4 for the other measures.

While Theorem 4.1 and equation 8 provide a guarantee on the expected value of the cumulative regret, they do not directly provide a guarantee on the expected value of the instantaneous regret. Specifically, they do not answer the question regarding which estimated solution $\hat{\boldsymbol{x}}_i$, for $1 \le i \le t$, achieves the smallest value of $\mathbb{E}[r_i]$. To this end, we define the index $\hat{t}$ of the optimal estimated solution up to time $t$ as follows:

$$\hat{t} = \arg\min_{1 \le i \le t} \mathbb{E}_{t-1}[F(\boldsymbol{x}^*) - F(\hat{\boldsymbol{x}}_i)], \tag{9}$$

where $\mathbb{E}_{t-1}[\cdot]$ is the conditional expectation given the dataset $D_{t-1}$, defined as follows:

$$D_{t-1} = \{(\boldsymbol{x}_1, \boldsymbol{w}_1, \varepsilon_1, \beta_1), \dots, (\boldsymbol{x}_{t-1}, \boldsymbol{w}_{t-1}, \varepsilon_{t-1}, \beta_{t-1})\}$$

for $t \ge 2$, and $D_0 = \emptyset$. Then, the following theorem holds.

**Theorem 4.2.** Under the assumptions of Theorem 4.1, the following inequality holds:

$$\mathbb{E}[r_{\hat{t}}] \le \frac{\mathbb{E}[R_t]}{t} \le 2\sum_{i=1}^{n} \zeta_i h_i \left( \frac{1}{t} \sum_{j=1}^{s_i} 2^{\nu_{ij}} \lambda_{ij} \left(tC_{2,\nu_{ij}}\right)^{1-\nu'_{ij}/2} \left(C_1\gamma_t\right)^{\nu'_{ij}/2} \right),$$

where $\hat{t}$ is given by equation 9, and $h_i(\cdot)$ along with all coefficients are as defined in Theorem 4.1. In addition, for the expectation, worst-case, best-case, $\alpha$-value-at-risk, $\alpha$-conditional value-at-risk, and mean absolute deviation measures, the following bound holds:

$$\mathbb{E}[r_{\hat{t}}] \le \frac{\mathbb{E}[R_t]}{t} \le C\sqrt{\frac{C_0\gamma_t}{t}},$$

where $C$ and $C_0$ are given in Corollary 4.1.

From Theorem 4.2, if $\gamma_t$ is a sublinear function and $h_i(\cdot)$ is continuous at 0, then using $h_i(0) = 0$ and equation 7, $\mathbb{E}[r_{\hat{t}}]$ satisfies

$$\lim_{t \to \infty} \mathbb{E}[r_{\hat{t}}] = 0.$$

Therefore, the expected regret associated with index $\hat{t}$ converges to zero as $t \to \infty$. However, computing $\hat{t}$ requires solving equation 9, which is generally intractable analytically. Nevertheless, $\hat{t}$ can alternatively be written as follows:

$$\hat{t} = \arg\max_{1 \le i \le t} \mathbb{E}_{t-1}[F(\hat{\boldsymbol{x}}_i)].$$

Therefore, $\hat{t}$ can be defined without using the unknown $\boldsymbol{x}^*$. The reason for expressing $\hat{t}$ as in equation 9 is to simplify the evaluation of inequalities in theoretical analysis, such as equation 23. Furthermore, the posterior distribution of $f$ given $D_{t-1}$ follows GPs. By using this, we can generate sample paths $\hat{f}_1, \ldots, \hat{f}_M$ from this posterior. For each sample path $\hat{f}_j$, we evaluate $F(\boldsymbol{x})$ and use these $M$ evaluations to estimate $\mathbb{E}_{t-1}[F(\boldsymbol{x})]$. In this way, $\hat{t}$ can be approximated from the sampled paths. In contrast, for the expectation measure, the following theorem shows that one can use $t$ itself as a substitute for $\hat{t}$.

**Theorem 4.3.** Under the assumptions of Theorem 4.1, the following holds for the expectation measure:

$$\mathbb{E}[r_{\hat{t}}] = \mathbb{E}[r_t] \leq \frac{\mathbb{E}[R_t]}{t} \leq 4\sqrt{\frac{C_0 \gamma_t}{t}},$$

where $C_0$ is given in Corollary 4.1.

Theorem 4.3 states that when considering the expectation measure as a robustness measure, an upper bound on $\mathbb{E}[r_t]$ is given, while for other robustness measures, estimation of $\hat{t}$ is still required. If the number of samples $M$ generated from the posterior distribution is sufficiently large, the estimated $\hat{t}$ can be expected to be close to the true $\hat{t}$, but whether it is possible to accurately derive the true $\hat{t}$ without estimation remains an important direction for future work. Nevertheless, in the synthetic function experiments conducted in Section 5.1, $\mathbb{E}[r_t]$ is used as the evaluation metric instead of $\mathbb{E}[r_{\hat{t}}]$, but the behavior of the convergence of $\mathbb{E}[r_t]$ to 0 in the proposed method has been confirmed in all settings. Therefore, similarly to how an upper bound for $\mathbb{E}[r_t]$ is given by Theorem 4.3, it is expected that some kind of upper bound holds for $\mathbb{E}[r_t]$ in other robustness measures.

In this section, we have presented theoretical guarantees for the expected value of both the regret and the cumulative regret. However, we have not addressed high-probability bounds. Such bounds can nonetheless be readily derived via Markov's inequality. In fact, for a non-negative random variable $X$, Markov's inequality yields $\mathbb{P}(X > a) \leq \mathbb{E}[X]/a$. By setting $a = \delta^{-1}\mathbb{E}[X]$, we obtain that $X \leq \delta^{-1}\mathbb{E}[X]$ with probability at least $1 - \delta$. Hence, for any non-negative value $J$ satisfying $\mathbb{E}[X] \leq J$, we can conclude that $X \leq \delta^{-1}J$ with probability at least $1 - \delta$. Applying this argument to the regret and cumulative regret, which are both non-negative, and invoking Theorems 4.1 and 4.2, we obtain the following result:

**Theorem 4.4.** Let $\delta \in (0, 1)$. Under the assumptions of Theorem 4.1, for any $t \geq 1$, the following inequality holds with probability at least $1 - \delta$:

$$R_t \leq 2\delta^{-1}t \sum_{i=1}^{n} \zeta_i h_i \left( \frac{1}{t} \sum_{j=1}^{s_i} 2^{\nu_{ij}} \lambda_{ij} \left( tC_{2,\nu_{ij}} \right)^{1-\nu'_{ij}/2} \left( C_1 \gamma_t \right)^{\nu'_{ij}/2} \right),$$

where $h_i(\cdot)$ and all coefficients are as defined in Theorem 4.1. In addition, for the index $\hat{t}$ defined in equation 9, the following inequality holds with probability at least $1 - \delta$:

$$r_{\hat{t}} \leq 2\delta^{-1} \sum_{i=1}^{n} \zeta_i h_i \left( \frac{1}{t} \sum_{j=1}^{s_i} 2^{\nu_{ij}} \lambda_{ij} \left( tC_{2,\nu_{ij}} \right)^{1-\nu'_{ij}/2} \left( C_1 \gamma_t \right)^{\nu'_{ij}/2} \right).$$

Although Theorem 4.4 provides high-probability bounds, these bounds are not tight with respect to $\delta$. In fact, the right-hand sides of both inequalities depend on $\delta^{-1}$. In contrast, most high-probability bounds based on the GP-UCB framework—such as those in Srinivas et al. (2010)—involve a $\log(\delta^{-1})$ term instead of $\delta^{-1}$. Therefore, deriving tighter high-probability bounds with respect to $\delta$ than those in Theorem 4.4 remains an important direction for future work. Nevertheless, $\delta$ is a probability parameter specified in advance by the user and is a constant. Therefore, when considering the order with respect to $t$ by treating $\delta$ as a constant, the high-probability bound in the proposed method is tighter than that in GP-UCB-based methods such as Srinivas et al. (2010). For example, the standard BO on the input space $\mathcal{X}$ with no input uncertainty is equivalent to maximizing the expectation measure on $\mathcal{X} \times \Omega$ using $|\Omega| = 1$. From Theorem 1 of Srinivas et al. (2010), the order of the high-probability upper bound for the cumulative regret is $\sqrt{t\gamma_t \log t}$, whereas in the proposed method, it is $\sqrt{t\gamma_t}$ according to Corollary 4.1. Furthermore, when considering the

expectation measure as a robustness measure in the presence of input uncertainty, according to Theorem 1 and 2 in Inatsu et al. (2024a), at time $t$ the order of the high-probability bound for $r_{t'}$ is $\sqrt{t^{-1}\gamma_t^2}$, where $t' \leq t$. On the other hand, the high-probability bound for $r_t$ in our proposed method is $\sqrt{t^{-1}\gamma_t}$. However, as given in Table 1, the definitions of $r_t$ in Srinivas et al. (2010); Inatsu et al. (2024a) and this study differ slightly, so it is important to note that these high-probability bounds cannot be directly compared.

## 4.1 Extension to Other Settings

In this section, we provide some results for uncontrollable settings. Furthermore, we provide an overview of methods and results for extending the proposed method to continuous spaces, and explain issues involved. Details on the extension to continuous spaces and to general uncontrollable settings are given in Appendix A and B.

**Uncontrollable Setting when $\mathcal{X}$ and $\Omega$ are Finite** We consider the case of uncontrollable settings, i.e., $\boldsymbol{w}_1, \ldots, \boldsymbol{w}_t$ follow the distribution $P^*$ and cannot be controlled even during optimization. We assume that $\boldsymbol{w}_1, \ldots, \boldsymbol{w}_t$ are mutually independent. The difference between the proposed algorithm in this setting and Algorithm 1 is only in how $\boldsymbol{w}_t$ is obtained. Specifically, in this setting, the acquisition function equation 5 cannot be used, and $\boldsymbol{w}$ sampled from $P^*$ becomes $\boldsymbol{w}_t$ at time $t$. We give the pseudo-code for the case where $\mathcal{X}$ and $\Omega$ are finite in the uncontrollable setting in Algorithm 2. When Algorithm 2 is performed, similar theorems as in the simulator setting hold.

**Theorem 4.5.** Assume that the regularity assumption, Assumption 4.1 and equation 2 hold. Suppose that $\mathcal{X}$ and $\Omega$ are finite. Also suppose that $\boldsymbol{w}_1, \ldots, \boldsymbol{w}_t$ follow $P^*$, and $\xi_1, \ldots, \xi_t$ are random variables following the chi-squared distribution with two degrees of freedom, where $f, \varepsilon_1, \ldots, \varepsilon_t, \boldsymbol{w}_1, \ldots, \boldsymbol{w}_t, \xi_1, \ldots, \xi_t$ are mutually independent. Let $\Omega = \{\boldsymbol{w}^{(1)}, \ldots, \boldsymbol{w}^{(J)}\}$, $p_j = \mathbb{P}(\boldsymbol{w} = \boldsymbol{w}^{(j)})$ and $p_{min} = \min_{1 \leq j \leq J} p_j$, and assume that $p_{min} > 0$. Define $\beta_t = 2\log(|\mathcal{X} \times \Omega|) + \xi_t$. Let $q(a) = \sum_{i=1}^n \zeta_i h_i \left(\sum_{j=1}^{s_i} \lambda_{ij} a^{\nu_{ij}}\right)$ be a function satisfying Assumption 4.1. Then, under the uncontrollable setting, if Algorithm 2 is performed, the following holds:

$$\mathbb{E}[R_t] \leq 2t \sum_{i=1}^n \zeta_i h_i \left(\frac{1}{t}\sum_{j=1}^{s_i} 2^{\nu_{ij}} \lambda_{ij} \left(tC_{2,\nu_{ij}}\right)^{1-\nu'_{ij}/2} (C'_1 \gamma_t)^{\nu'_{ij}/2}\right),$$

where $\nu'_{ij} = \min\{\nu_{ij}, 1\}$, $C'_1 = \frac{2p_{min}^{-1}}{\log(1+\sigma_{\text{noise}}^{-2})}$, $C_{2,\nu_{ij}} = \mathbb{E}[\beta_t^{\nu_{ij}/(2-\nu'_{ij})}]$, and the expectation is taken over all sources of randomness, including $f, \varepsilon_1, \ldots, \varepsilon_t, \boldsymbol{w}_1, \ldots, \boldsymbol{w}_t, \beta_1, \ldots, \beta_t$. In addition, for the expectation, worst-case, best-case, $\alpha$-value-at-risk, $\alpha$-conditional value-at-risk, and mean absolute deviation measures, the following inequality holds:

$$\mathbb{E}[R_t] \leq C\sqrt{tC'_1(2\log(|\mathcal{X} \times \Omega|) + 2)\gamma_t},$$

where $C$ is 8 for the mean absolute deviation measure and 4 for the other measures.

**Theorem 4.6.** Under the assumptions of Theorem 4.5, the following holds:

$$\mathbb{E}[r_{\hat{t}}] \leq \frac{\mathbb{E}[R_t]}{t} \leq 2\sum_{i=1}^n \zeta_i h_i \left(\frac{1}{t}\sum_{j=1}^{s_i} 2^{\nu_{ij}} \lambda_{ij} \left(tC_{2,\nu_{ij}}\right)^{1-\nu'_{ij}/2} (C'_1 \gamma_t)^{\nu'_{ij}/2}\right),$$

where $\hat{t}$ is given by equation 9, and the function $h_i(\cdot)$ and all coefficients are as defined in Theorem 4.5. In addition, for the expectation, worst-case, best-case, $\alpha$-value-at-risk, $\alpha$-conditional value-at-risk, and mean absolute deviation measures, the following holds:

$$\mathbb{E}[r_{\hat{t}}] \leq \frac{\mathbb{E}[R_t]}{t} \leq C\sqrt{\frac{C'_1(2\log(|\mathcal{X} \times \Omega|) + 2)\gamma_t}{t}},$$

where $C$ is given in Theorem 4.5. Furthermore, in the expectation measure, the following holds:

$$\mathbb{E}[r_t] \leq \frac{\mathbb{E}[R_t]}{t} \leq 4\sqrt{\frac{C'_1(2\log(|\mathcal{X} \times \Omega|) + 2)\gamma_t}{t}}.$$

---

**Algorithm 2** RRGP-UCB for robustness measures in the uncontrollable setting when $\mathcal{X}$ and $\Omega$ are finite.

---

**Input:** GP prior $\mathcal{GP}(0, k)$
  **for** $t = 1, 2, \ldots$ **do**
    Generate $\xi_t$ from chi-squared distribution with two degrees of freedom
    Compute $\beta_t = 2 \log(|\mathcal{X} \times \Omega|) + \xi_t$
    Compute $Q_{t-1}(\boldsymbol{x}, \boldsymbol{w})$ for each $(\boldsymbol{x}, \boldsymbol{w}) \in \mathcal{X} \times \Omega$
    Compute $Q_{t-1}(\boldsymbol{x})$ for each $\boldsymbol{x} \in \mathcal{X}$
    Estimate $\hat{\boldsymbol{x}}_t$ by $\hat{\boldsymbol{x}}_t = \arg\max_{\boldsymbol{x} \in \mathcal{X}} \rho(\mu_{t-1}(\boldsymbol{x}, \cdot))$
    Select next evaluation point $\boldsymbol{x}_t$ by equation 4
    Generate $\boldsymbol{w}_t$ from $P^*$
    Observe $y_t = f(\boldsymbol{x}_t, \boldsymbol{w}_t) + \varepsilon_t$ at point $(\boldsymbol{x}_t, \boldsymbol{w}_t)$
    Update GP by adding observed data
  **end for**

---

Here, note that by using Theorems 4.5 and 4.6, and Markov's inequality, high-probability bounds similar to those in Theorem 4.4 can also be obtained in this setting.

**Extension to Continuous Spaces** In this section, we provide an overview of the extension of the proposed method to continuous spaces. In Algorithm 1 or 2, $\beta_t$ is defined using $|\mathcal{X} \times \Omega|$. Therefore, if $\mathcal{X}$ or $\Omega$ is a continuous space, this definition cannot be used directly. In this case, discretization of the continuous space is necessary, and to perform a theoretically valid discretization, additional assumptions on the differentiability of $f$, such as Assumption 2.1 in Takeno et al. (2023), are required. In fact, in Appendix A, the proposed method is extended to continuous spaces under similar assumptions. However, this approach has two issues. First, to propose a theoretically valid method, it is necessary to increase the number of discretized points according to time $t$, resulting in $\beta_t$ needing to be large according to time $t$, as in Theorem A.1. Thus, one of the advantages of the proposed method, namely that $\beta_t$ does not need to increase according to time $t$, is lost. Second, theoretically valid discretization requires constants such as $a_1$ and $b_1$ in Assumption A.1, and these are used to define $\beta_t$. However, although these constants $a_1$ and $b_1$ are determined by the kernel function, their actual values are difficult to calculate, resulting in the need for estimation or heuristic adjustment of these values. Therefore, the advantage of the proposed method, which eliminates the need for estimation or tuning of $\beta_t$, is also lost. Although an extension to continuous space is provided in Appendix A, the two issues mentioned above remain, and resolving these issues is an important direction for future work.

## 5 Numerical Experiments

In this section, we verify the performance of the proposed method using both synthetic benchmark functions and real-world data on carrier lifetime values of silicon ingots used in solar cells. In all experiments, a GP model with a zero mean function is used as the surrogate model. Further details regarding the experimental settings, as well as additional experiments not included in the main text, are described in Appendix D.

### 5.1 Synthetic Function

The input space $\mathcal{X} \times \Omega$ was defined as a subset of $[-M, M]^d \times [-M, M]^d \equiv [-M, M]^D$; there each coordinate was uniformly discretized into $s$ grid points. Three different configurations of $(M, D, s)$ were considered in the experiments: $(5, 2, 50)$, $(2.5, 4, 15)$, and $(2, 6, 7)$. The black-box function $f$ varied depending on the dimension $D$. When $D = 2$, $f$ was a sample path drawn from a GP, referred to as the 2D synthetic function. When $D = 4$, the black-box function was defined as $f(x_1, x_2, w_1, w_2) = f_H(x_1 + w_1, x_2 + 0.5w_2)$ (4D synthetic function), where $f_H(a, b)$ is Himmelblau's function with translation and scaling. For the case $D = 6$, $f$ was defined as the sum of four independent GP sample paths $f_1, \ldots, f_4$ (6D synthetic function), where the dependencies of each function were specified as follows: $f_1$ on $(x_1, x_2, x_3)$, $f_2$ on $(x_2, x_3, w_1)$, $f_3$ on $(x_3, w_1, w_2)$, and $f_4$ on $(w_1, w_2, w_3)$. In each setting, the kernel function $k(\cdot, \cdot)$ used in the GP surrogate model was defined as follows:

**(2D synthetic function):** $k(\boldsymbol{\theta}, \boldsymbol{\theta}') = \exp(\|\boldsymbol{\theta} - \boldsymbol{\theta}'\|_2^2/2)$.

**(4D synthetic function):** $k(\boldsymbol{\theta}, \boldsymbol{\theta}') = \exp(\|\boldsymbol{\theta} - \boldsymbol{\theta}'\|_2^2/10)$.

**(6D synthetic function):**

$$k(\boldsymbol{\theta}, \boldsymbol{\theta}') = 1.25\exp(\|\boldsymbol{\theta}_1 - \boldsymbol{\theta}'_1\|_2^2/1.75) + 0.75\exp(\|\boldsymbol{\theta}_2 - \boldsymbol{\theta}'_2\|_2^2/1.75) + \exp(\|\boldsymbol{\theta}_3 - \boldsymbol{\theta}'_3\|_2^2/2) + \exp(\|\boldsymbol{\theta}_4 - \boldsymbol{\theta}'_4\|_2^2/1.5),$$

where $\boldsymbol{\theta}_1 = (x_1, x_2, x_3)$, $\boldsymbol{\theta}_2 = (x_2, x_3, w_1)$, $\boldsymbol{\theta}_3 = (x_3, w_1, w_2)$ and $\boldsymbol{\theta}_4 = (w_1, w_2, w_3)$.

Among these settings, only the 2D synthetic function setting used a surrogate model that matched the true black-box function. The remaining two settings intentionally introduced a mismatch between the surrogate model and the black-box function. In addition, all experiments were conducted under the assumption that observations were corrupted by independent Gaussian noise with mean zero and variance $10^{-6}$. To evaluate robustness, three robustness measures were considered, using the probability mass function $p(\boldsymbol{w})$ of $\boldsymbol{w}$ defined for each setting:

**(EXP):** Expectation measure, $F(\boldsymbol{x}) = \mathbb{E}[f(\boldsymbol{x}, \boldsymbol{w})] \equiv F_{\exp}(\boldsymbol{x})$.

**(PTR):** Probability threshold robustness measure, $F(\boldsymbol{x}) = \mathbb{P}(f(\boldsymbol{x}, \boldsymbol{w}) \geq h)$.

**(EXP-MAE):** Weighted sum of expectation measure and mean absolute deviation,

$$F(\boldsymbol{x}) = F_{\exp}(\boldsymbol{x}) - \alpha\mathbb{E}[|f(\boldsymbol{x}, \boldsymbol{w}) - F_{\exp}(\boldsymbol{x})|].$$

In the 2D, 4D, and 6D synthetic function settings, the values of $(h, \alpha)$ were set to $(0.5, 1)$, $(0.18, 4)$, and $(2, 8)$, respectively. In this experiment, only the acquisition function was changed, and the evaluation metric was the regret $r_t = F(\boldsymbol{x}^*) - F(\hat{\boldsymbol{x}}_t)$. To compare performance, nine methods were evaluated, including the proposed method: random sampling (Random), uncertainty sampling (US), Bayesian quadrature (BQ) (Beland & Nair, 2017), BPT-UCB (Iwazaki et al., 2021a), BPT-UCB (fixed), BBBMOBO (Inatsu et al., 2024a), BBBMOBO (fixed), the proposed method (Proposed), and Proposed (fixed). The random method selected $(\boldsymbol{x}_t, \boldsymbol{w}_t)$ uniformly at random, while US selected $(\boldsymbol{x}_t, \boldsymbol{w}_t)$ by maximizing $\sigma_{t-1}^2(\boldsymbol{x}, \boldsymbol{w})$. For the remaining seven methods, $\boldsymbol{x}_t$ was selected according to each method's acquisition function. For all methods except BPT-UCB and BPT-UCB (fixed), $\boldsymbol{w}_t$ was selected using equation 5. The method for selecting $\boldsymbol{w}_t$ in BPT-UCB and BPT-UCB (fixed) is described in Appendix D. BQ and BPT-UCB were originally proposed for the EXP and PTR measures, respectively. Although BBBMOBO was designed for Pareto optimization over multiple robustness measures, it can be applied to a single robustness measure as well. The trade-off parameters used in BPT-UCB, BBBMOBO, and Proposed were set to theoretical values. In contrast, the methods marked with (fixed) used fixed values smaller than the theoretical ones. Note that in the EXP setting, BBBMOBO (fixed) and Proposed (fixed) are the same (see Appendix D for details). Under these settings, a single random initial point was selected, and the algorithms were run for 300 iterations. This process was repeated 100 times, and the average $r_t$ was calculated at each iteration. As shown in Figure 2, Random and US, which are not designed to maximize robustness measures, performed poorly in all settings. BQ, BPT-UCB, and BPT-UCB (fixed) were effective for EXP and PTR, but their performance for EXP-MAE in the 2D synthetic function was insufficient. This is because these methods are not tailored for EXP-MAE. For BBBMOBO and Proposed, as well as BBBMOBO (fixed) and Proposed (fixed), performance tended to be similar. This is because the only difference lies in whether $\boldsymbol{x}_t$ is set to $\tilde{\boldsymbol{x}}_t$ or selected via equation 4, aside from the trade-off parameters. For BBBMOBO (fixed) and Proposed (fixed), using smaller-than-theoretical trade-off parameters led to improved practical performance, achieving favorable results in many settings. However, in the 4D synthetic function under the EXP measure, regret was not fully reduced. One reason is that the surrogate model fails to correctly express the true black-box function. Furthermore, small trade-off parameters limit exploration, often resulting in convergence to local optima. In fact, Figure 5 of Srinivas et al. (2010) shows the results of BO (Mean Only) using only the posterior mean, and Figure 5 of Inatsu et al. (2024a) compares the differences in $\beta$ for the expectation measure, suggesting that in both cases, small $\beta$ tends to lead to local optima. In contrast, the Proposed method, by employing random trade-off parameters, occasionally explores more broadly. This increases the likelihood of escaping local solutions and,

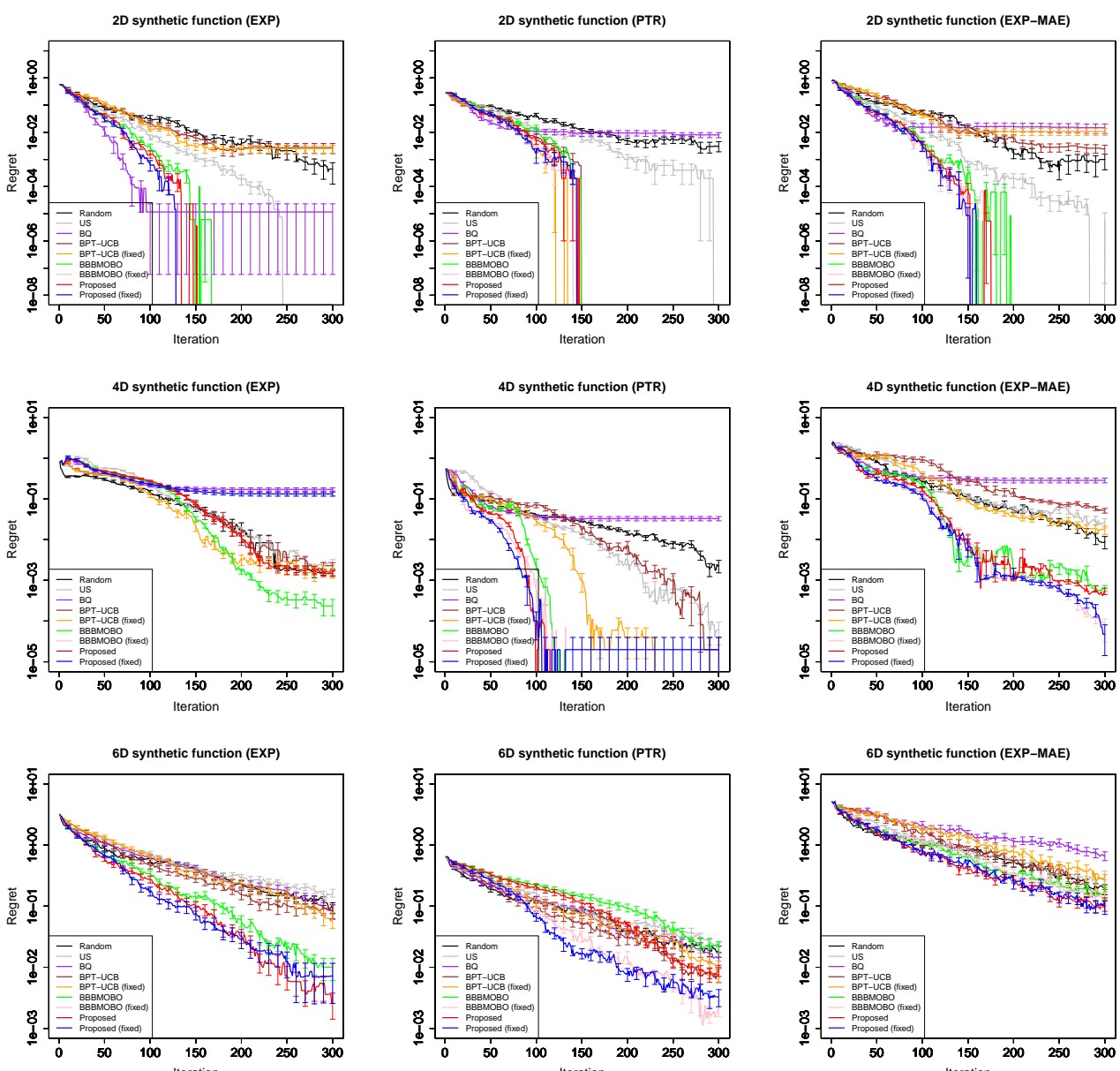

Figure 2: Average regret across 100 simulations for each method. Top, middle, and bottom rows correspond to 2D, 4D, and 6D synthetic function settings, respectively. Left, center, and right columns show results for EXP, PTR, and EXP-MAE, respectively. Since BBBMOBO (fixed) and Proposed (fixed) are the same in EXP, only Proposed (fixed) is displayed in the figures on the left column. Error bars represent twice the standard error.

consequently, improves performance. This demonstrates a key advantage of using randomly varying trade-off parameters beyond the theoretical guarantees. The trade-off parameters in BBBMOBO are on the order of $O(\log(t|\mathcal{X} \times \Omega|))$. Since they are often larger and more conservative than the $\beta_t$ values used in Proposed, Proposed generally outperformed BBBMOBO, except in the 4D synthetic function (EXP) setting. Overall, the Proposed method performed comparably to or better than the baseline methods across most settings.

## 5.2  Synthetic Function Experiments under Uncontrollable Settings

In this section, we changed the experiment in Section 5.1 to the uncontrollable setting, where $\boldsymbol{w}_t$ cannot be selected and is obtained randomly according to the probability mass function $p(\boldsymbol{w})$. The method for selecting $\boldsymbol{x}_t$ was the same as in Section 5.1. As in Section 5.1, Figure 3 shows that the proposed method outperforms the comparison methods in most settings. On the other hand, in the experiments in Section 5.1, the regret of Proposed (fixed) did not decrease in the 4D synthetic function (EXP), while in Figure 3, the regret decreased more than Figure 2. This is because $\boldsymbol{w}$ could not be selected, and as a result of random sampling, more exploration was performed, making it possible to avoid local solutions.

## 5.3  Carrier Lifetime Data

In this section, we conducted experiments using the carrier lifetime dataset (Kutsukake et al., 2015), which quantifies the performance of silicon ingots used in solar cells. The original dataset includes 6586 two-dimensional coordinates on the surface of a silicon ingot and the corresponding carrier lifetime values, denoted by $\mathrm{LT}(x_1, x_2)$ at each coordinate $(x_1, x_2)$. In this experiment, we focused on the subset $\tilde{\mathcal{X}} \equiv \{(2a+6, 2b+6) \mid 1 \leq a \leq 88, 1 \leq b \leq 72\}$, which includes 6336 of these points. The set of design variables $\mathcal{X}$ was defined as a subset of $\tilde{\mathcal{X}}$, specifically $\mathcal{X} = \{(22a - 4, 18b - 2) \mid 1 \leq a \leq 8, 1 \leq b \leq 8\}$. In addition, we defined $\Omega = \{(2a - 12, 2b - 10) \mid 1 \leq a \leq 11, 1 \leq b \leq 9\}$. This results in $|\mathcal{X}| = 64$, $|\Omega| = 99$, and $|\mathcal{X} \times \Omega| = 6336$, with the set $\mathcal{X} + \Omega \equiv \{\boldsymbol{x} + \boldsymbol{w} \mid \boldsymbol{x} \in \mathcal{X}, \boldsymbol{w} \in \Omega\}$ equal to $\tilde{\mathcal{X}}$. For each input $(x_1, x_2, w_1, w_2) \in \mathcal{X} \times \Omega$, the black-box function was defined as $f(x_1, x_2, w_1, w_2) = \mathrm{LT}(x_1 + w_1, x_2 + w_2)$. The kernel function used in the surrogate model was the Matérn 3/2 kernel, defined as follows:

$$k((x_1, x_2, w_1, w_2), (x_1', x_2', w_1', w_2')) = 4\left(1 + \frac{\sqrt{3}\|\boldsymbol{\theta} - \boldsymbol{\theta}'\|_2}{25}\right) \exp\left(-\frac{\sqrt{3}\|\boldsymbol{\theta} - \boldsymbol{\theta}'\|_2}{25}\right),$$

where $\boldsymbol{\theta} = (x_1 + w_1, x_2 + w_2)$ and $\boldsymbol{\theta}' = (x_1' + w_1', x_2' + w_2')$. The experiment was performed under the assumption of no observation noise. However, to ensure numerical stability when computing the inverse of the kernel matrix, a nominal noise variance of $\sigma_{\text{noise}}^2 = 10^{-6}$ was added. The same three robustness measures and nine methods as described in Section 5.1 were employed. The probability mass function was set to $p(\boldsymbol{w}) = 1/99$, and the parameters $(h, \alpha) = (2.9, 4)$. Under this setting, one initial point was selected at random, and each algorithm was run for 500 iterations. This procedure was repeated 100 times, and the average regret $r_t$ was computed for each iteration. As shown in Figure 4, the Proposed method demonstrated performance comparable to that of the baseline methods, even on the carrier lifetime dataset.

## 6  Conclusion

In this paper, we proposed a new method for BO of robustness measures for black-box functions with input uncertainty. The proposed method estimated the optimal solution using posterior means, sampled the parameters of GP-UCB from a probability distribution, and determined the next evaluation point based on the estimated solution, credible intervals, and posterior variance. In Section 4, we provided upper bounds on the expected regret and cumulative regret and showed that their orders for commonly used robustness measures, including the expectation measure, were $O(\sqrt{\gamma_t/t})$ and $O(\sqrt{t\gamma_t})$, respectively.

Compared to existing methods, the proposed method offered the following three advantages. First, unlike the methods in Beland & Nair (2017); Iwazaki et al. (2021a), which were tailored to specific robustness measures, our method was applicable more generally to any robustness measure satisfying the condition in equation 2. Second, in contrast to the method in Inatsu et al. (2024a), which was also not restricted to a particular robustness measure, our method did not require hyperparameters for the acquisition function. Third, we derived the order of the expected regret and cumulative regret defined in terms of the estimated solution based on the posterior mean. To the best of our knowledge, this study is the first to establish expected (cumulative) regret bounds for various robustness measures.

However, the proposed method also had certain limitations. Most significantly, while it randomly replaced the parameters of GP-UCB, a key feature of the method, this mechanism alone did not significantly improve

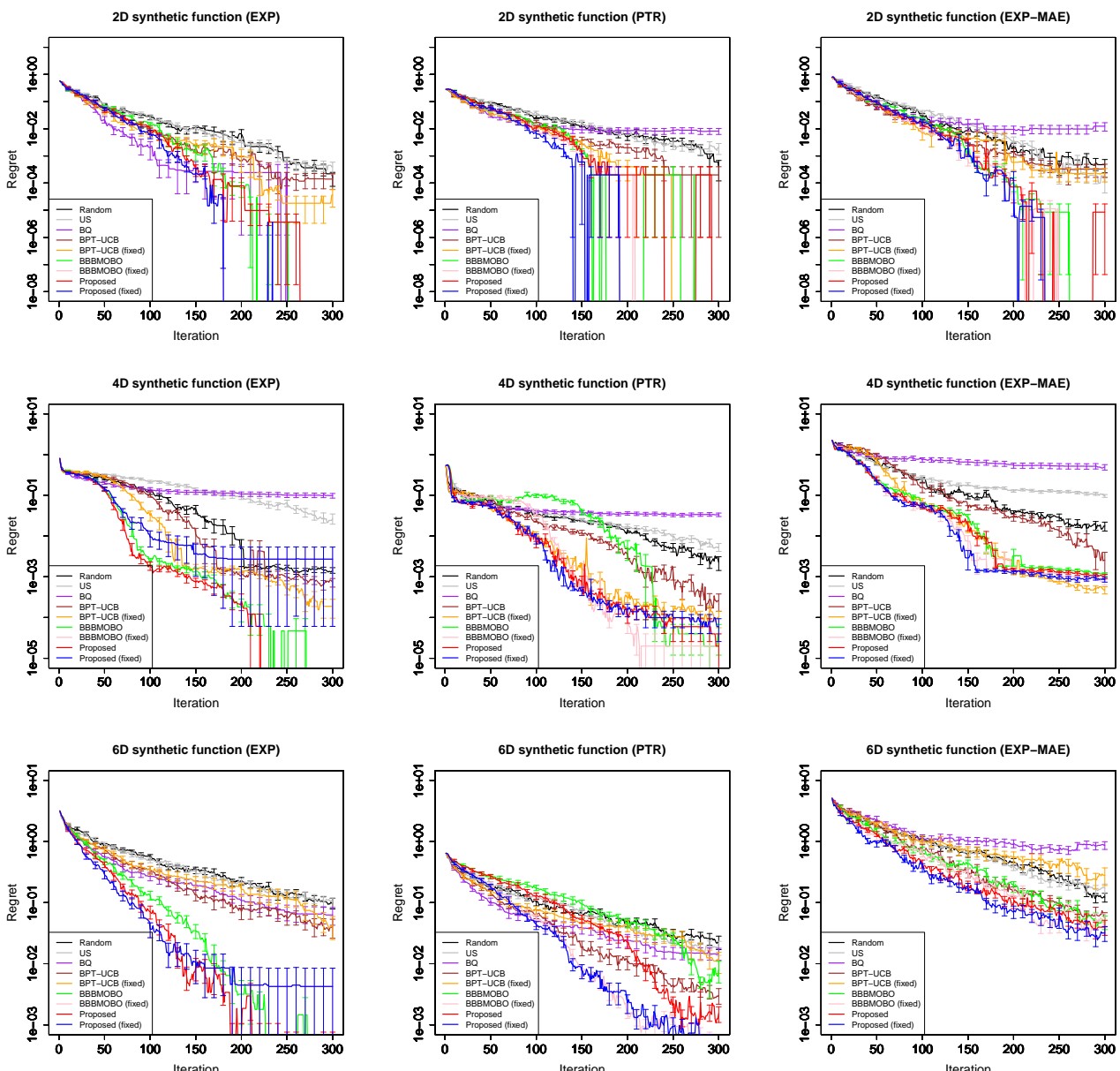

Figure 3: Average regrets across 100 simulations under the uncontrollable setting for each method. Top, middle, and bottom rows correspond to 2D, 4D, and 6D synthetic function settings, respectively. Left, center, and right columns show the results for EXP, PTR, and EXP-MAE, respectively. Since BBBMOBO (fixed) and Proposed (fixed) are the same in EXP, only Proposed (fixed) is displayed in the figures on the left column. Error bars represent twice the standard error.

practical performance. Additionally, although we derived a high-probability bound for the theoretical analysis, the appearance of $\delta^{-1}$ in the bound, which was not tight compared to typical GP-UCB-based bounds with respect to $\delta$, posed a limitation. Furthermore, although the proposed method can be extended to continuous spaces, it has the disadvantage of losing the advantages of the proposed method, namely, that $\beta_t$ does not need to be adjusted and that no increase in $t$ is required. Finally, it was generally difficult to calculate the best index $\hat{t}$ among the estimated optimal solutions $\hat{\boldsymbol{x}}_t$.

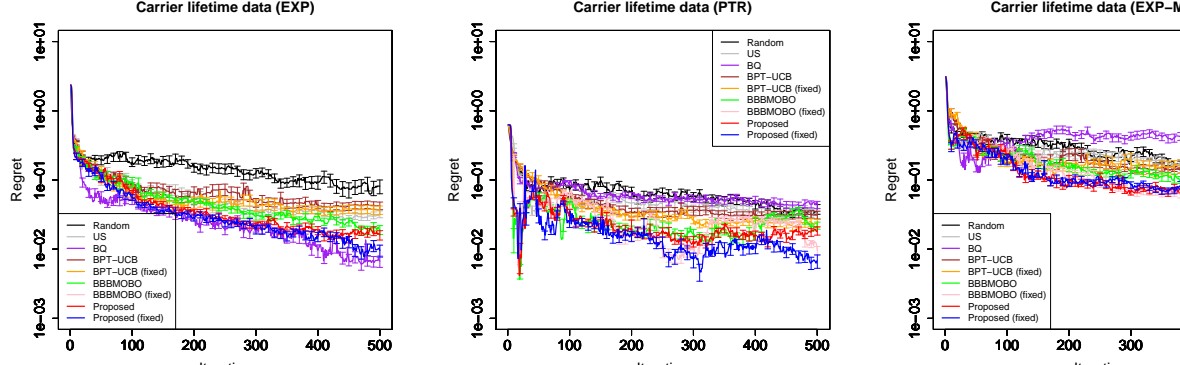

Figure 4: Average regrets on carrier lifetime dataset over 100 simulations for each method. Left, middle, and right columns correspond to results for EXP, PTR, and EXP-MAE, respectively. Since BBBMOBO (fixed) and Proposed (fixed) are the same in EXP, only Proposed (fixed) is displayed in the figures on the left column. Error bars represent twice standard error.

Addressing these disadvantages remains an important direction for future study.

## Acknowledgments

This work was supported by JSPS KAKENHI (JP20H00601,JP23K16943) and JST ACT-X (JPMJAX24C3).

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

---

**Algorithm 3** RRGP-UCB for robustness measures when $\mathcal{X}$ is continuous and $\Omega$ is finite.

---

**Input:** GP prior $\mathcal{GP}(0, \ k)$, $\{\kappa_t^{(1)}\}_{t \in \mathbb{N}}$, $1 \le \kappa_1^{(1)} \le \kappa_2^{(1)} \le \cdots$
  **for** $t = 1, 2, \ldots$ **do**
    Generate $\xi_t$ from chi-squared distribution with two degrees of freedom
    Compute $\beta_t = 2 \log \kappa_t^{(1)} + \xi_t$
    Compute $Q_{t-1}(\boldsymbol{x}, \boldsymbol{w})$ for each $(\boldsymbol{x}, \boldsymbol{w}) \in \mathcal{X} \times \Omega$
    Compute $Q_{t-1}(\boldsymbol{x})$ for each $\boldsymbol{x} \in \mathcal{X}$
    Estimate $\hat{\boldsymbol{x}}_t$ by $\hat{\boldsymbol{x}}_t = \arg\max_{\boldsymbol{x} \in \mathcal{X}} \rho(\mu_{t-1}(\boldsymbol{x}, \cdot))$
    Select next evaluation point $\boldsymbol{x}_t$ by equation 4
    Select next evaluation point $\boldsymbol{w}_t$ by equation 5
    Observe $y_t = f(\boldsymbol{x}_t, \boldsymbol{w}_t) + \varepsilon_t$ at point $(\boldsymbol{x}_t, \boldsymbol{w}_t)$
    Update GP by adding observed data
  **end for**

---

## Appendix

## A  Extension of the Proposed Method to Continuous Settings

In this section, we consider the case where $\mathcal{X}$ and $\Omega$ are not finite sets. We consider the following three cases separately: Only $\mathcal{X}$ is continuous, only $\Omega$ is continuous, and both $\mathcal{X}$ and $\Omega$ are continuous.

### A.1  Extension to Continuous Settings when $\mathcal{X}$ is Continuous and $\Omega$ is Finite

Let $\mathcal{X}$ be a continuous set, and let $\Omega$ be a finite set. Suppose that $\mathcal{X}$ is a compact and convex set with $\mathcal{X} \subset [0, r]^{d_1}$. In this setting, the only difference between Algorithm 1 and an extension of the proposed method is the distribution of $\beta_t$. Specifically, by using $\kappa_t^{(1)}$ with $1 \le \kappa_1^{(1)} \le \kappa_2^{(1)} \le \cdots$, we define $\beta_t = 2 \log \kappa_t^{(1)} + \xi_t$. Theoretically valid values of $\kappa_1^{(1)}, \ldots, \kappa_t^{(1)}$ and the theoretical analysis are given in Appendix A.1.1. The pseudo-code for the case when $\mathcal{X}$ is continuous and $\Omega$ is finite is provided in Algorithm 3.

### A.1.1  Theoretical Analysis in the Continuous Setting when $\mathcal{X}$ is Continuous and $\Omega$ is Finite

To derive the theoretical guarantee, we introduce additional two assumptions.

**Assumption A.1.** There exist $a_1, b_1 > 0$ such that

$$\mathbb{P}\left(\sup_{\boldsymbol{x} \in \mathcal{X}} \left| \frac{\partial f(\boldsymbol{x}, \boldsymbol{w})}{\partial x_j} \right| > L\right) \le a_1 \exp\left(-\left(\frac{L}{b_1}\right)^2\right) \quad \text{for } j \in \{1, \ldots, d_1\}, \boldsymbol{w} \in \Omega.$$

**Assumption A.2.** There exists a non-decreasing, concave function $q_1(a)$ such that $q_1(0) = 0$ and

$$|\rho(f(\boldsymbol{x}, \cdot)) - \rho(f(\boldsymbol{x}', \cdot))| \le q_1\left(\max_{\boldsymbol{w} \in \Omega} |f(\boldsymbol{x}, \boldsymbol{w}) - f(\boldsymbol{x}', \boldsymbol{w})|\right) \quad \text{for } \boldsymbol{x}, \boldsymbol{x}' \in \mathcal{X}.$$

For Assumption A.1, note that when $\boldsymbol{w}$ is fixed, $f(\boldsymbol{x}, \boldsymbol{w})$ is a GP on $\mathcal{X}$. In GP-based BOs, similar assumptions to Assumption A.1 are used in, for example, Srinivas et al. (2010); Takeno et al. (2023). Here, for the Bayes risk (expectation), worst-case, best-case, $\alpha$-value-at-risk, $\alpha$-conditional value-at-risk, and mean absolute deviation measures described in Table 4 in Inatsu et al. (2024a), we can use $q^{(m)}$ in the table as $q_1(a)$ in Assumption A.2. Details are described in Appendix C.5. Then, the following theorem holds.

**Theorem A.1.** Assume that the regularity assumption, Assumptions 4.1, A.1 and A.2, and equation 2 hold. Suppose that $\xi_1, \ldots, \xi_t$ are random variables following the chi-squared distribution with two degrees of freedom, where $f, \varepsilon_1, \ldots, \varepsilon_t, \xi_1, \ldots, \xi_t$ are mutually independent. Let $\kappa_t^{(1)} = (1 + \lceil b_1 d_1 r t^2 (\sqrt{\log(a_1 d_1 |\Omega|)} + \sqrt{\pi}/2) \rceil^{d_1}) |\Omega|$, and define $\beta_t = 2 \log \kappa_t^{(1)} + \xi_t$. Let $q_1(a)$ and $q(a) = \sum_{i=1}^{n} \zeta_i h_i \left( \sum_{j=1}^{s_i} \lambda_{ij} a^{\nu_{ij}} \right)$ be functions

satisfying Assumptions A.2 and 4.1, respectively. Then, if Algorithm 3 is performed, the following holds:

$$\mathbb{E}[R_t] \leq t q_1 \left( \frac{\pi^2}{6t} \right) + 2t \sum_{i=1}^{n} \zeta_i h_i \left( \frac{1}{t} \sum_{j=1}^{s_i} 2^{\nu_{ij}} \lambda_{ij} \left( t C_{2,\nu_{ij},t} \right)^{1-\nu'_{ij}/2} (C_1 \gamma_t)^{\nu'_{ij}/2} \right),$$

where $\nu'_{ij} = \min\{\nu_{ij}, 1\}, C_1 = \frac{2}{\log(1+\sigma_{\text{noise}}^{-2})}, C_{2,\nu_{ij},t} = \mathbb{E}[\beta_t^{\nu_{ij}/(2-\nu'_{ij})}]$, and the expectation is taken over all sources of randomness, including $f, \varepsilon_1, \ldots, \varepsilon_t, \beta_1, \ldots, \beta_t$.

Unlike the case where $\mathcal{X}$ is finite, since $C_{2,\nu_{ij},t}$ diverges with the order of $(\log t)^{\nu_{ij}/(2-\nu'_{ij})}$, $(C_{2,\nu_{ij},t})^{1-\nu'_{ij}/2}$ also diverges with the order of $(\log t)^{\nu_{ij}/2}$. Hence, even if $\lim_{t\to\infty} \gamma_t/t = 0$, that is, $\gamma_t$ is sublinear, if $\gamma_t$ diverges with the order of $\frac{t}{(\log t)^{\nu_{ij}/\nu'_{ij}}}$ or higher, the argument for the function $h_i(\cdot)$ diverges to infinity. On the other hand, if the order of $\gamma_t$ is slower than $\frac{t}{(\log t)^{\nu_{ij}/\nu'_{ij}}}$, then $\lim_{t\to\infty} \mathbb{E}[R_t]/t = 0$ under the assumptions that $q_1(\cdot)$ and $h_i(\cdot)$ are continuous at 0. Next, as in the case of $\mathcal{X}$, for six robustness measures including the expectation measure, the following corollary holds.

**Corollary A.1.** Under the assumptions of Theorem A.1, for the expectation, worst-case, best-case, $\alpha$-value-at-risk, $\alpha$-conditional value-at-risk, and mean absolute deviation measures, the following holds:

$$\mathbb{E}[R_t] \leq C \left( \frac{\pi^2}{6} + 4\sqrt{C_1 t (2 \log \kappa_t^{(1)} + 2) \gamma_t} \right),$$

where $C$ is 2 for the mean absolute deviation measure, and 1 for the other measures.

Furthermore, for the index $\hat{t}$ given by equation 9, the following theorem holds.

**Theorem A.2.** Under the assumptions of Theorem A.1, the following holds:

$$\mathbb{E}[r_{\hat{t}}] \leq \frac{\mathbb{E}[R_t]}{t} \leq q_1 \left( \frac{\pi^2}{6t} \right) + 2 \sum_{i=1}^{n} \zeta_i h_i \left( \frac{1}{t} \sum_{j=1}^{s_i} 2^{\nu_{ij}} \lambda_{ij} \left( t C_{2,\nu_{ij},t} \right)^{1-\nu'_{ij}/2} (C_1 \gamma_t)^{\nu'_{ij}/2} \right),$$

where $\hat{t}$ is given by equation 9, and functions $q_1(\cdot)$, $h_i(\cdot)$ and all coefficients are as defined in Theorem A.1. Moreover, for the expectation, worst-case, best-case, $\alpha$-value-at-risk, $\alpha$-conditional value-at-risk, and mean absolute deviation measures, the following holds:

$$\mathbb{E}[r_{\hat{t}}] \leq C \left( \frac{\pi^2}{6t} + 4\sqrt{\frac{C_1 (2 \log \kappa_t^{(1)} + 2) \gamma_t}{t}} \right),$$

where $C$ is given in Corollary A.1. In addition, for the expectation measure, $\hat{t}$ satisfies $\hat{t} = t$ and

$$\mathbb{E}[r_{\hat{t}}] = \mathbb{E}[r_t] \leq \frac{\pi^2}{6t} + 4\sqrt{\frac{C_1 (2 \log \kappa_t^{(1)} + 2) \gamma_t}{t}}.$$

Proofs are given by using the same argument as in the proof of Theorem 4.2 and 4.3.

## A.2 Extension to Continuous Settings when $\mathcal{X}$ is Finite and $\Omega$ is Continuous

Let $\mathcal{X}$ be a finite set, and let $\Omega$ be a continuous set. Suppose that $\Omega$ is a compact and convex set with $\Omega \subset [0, r]^{d_2}$. In this setting, the difference between Algorithm 1 and an extending method is not only the difference in the distribution of $\beta_t$. Specifically, the method for estimating the optimal solution $\hat{x}_t$ and the method for selecting $x_t$ also need to be changed.

Let $t \geq 1$, and let $\Omega_t$ be a finite subset of $\Omega$. For $w \in \Omega$, let $[w]_t$ be the element of $\Omega_t$ closest to $w$. Then, for $(x, w) \in \mathcal{X} \times \Omega$, we define $\mu_{t-1}^\dagger(x, w)$, $l_{t-1}^\dagger(x, w)$ and $u_{t-1}^\dagger(x, w)$ as follows:

$$\mu_{t-1}^\dagger(x, w) = \mu_{t-1}(x, [w]_t), \ l_{t-1}^\dagger(x, w) = l_{t-1}(x, [w]_t) = \mu_{t-1}(x, [w]_t) - \beta_t^{1/2} \sigma_{t-1}(x, [w]_t),$$

$$u_{t-1}^\dagger(x, w) = u_{t-1}(x, [w]_t) = \mu_{t-1}(x, [w]_t) + \beta_t^{1/2} \sigma_{t-1}(x, [w]_t).$$

---

**Algorithm 4** RRGP-UCB for robustness measures when $\mathcal{X}$ is finite and $\Omega$ is continuous.

---

**Input:** GP prior $\mathcal{GP}(0, k)$, $\{\kappa_t^{(2)}\}_{t\in\mathbb{N}}$, $1 \le \kappa_1^{(2)} \le \kappa_2^{(2)} \le \cdots$, finite subsets $\Omega_1, \Omega_2, \ldots \subset \Omega$

    **for** $t = 1, 2, \ldots$ **do**

        Generate $\xi_t$ from chi-squared distribution with two degrees of freedom

        Compute $\beta_t = 2 \log \kappa_t^{(2)} + \xi_t$

        Compute $Q_{t-1}^\dagger(\boldsymbol{x}, \boldsymbol{w})$ for each $(\boldsymbol{x}, \boldsymbol{w}) \in \mathcal{X} \times \Omega$

        Compute $Q_{t-1}^\dagger(\boldsymbol{x})$ for each $\boldsymbol{x} \in \mathcal{X}$

        Estimate $\hat{\boldsymbol{x}}_t^\dagger$ by $\hat{\boldsymbol{x}}_t^\dagger = \arg\max_{\boldsymbol{x}\in\mathcal{X}} \rho(\mu_{t-1}^\dagger(\boldsymbol{x}, \cdot))$

        Select next evaluation point $\boldsymbol{x}_t$ by equation 12

        Select next evaluation point $\boldsymbol{w}_t$ by $\boldsymbol{w}_t = \arg\max_{\boldsymbol{w}\in\Omega} \sigma_{t-1}^2(\boldsymbol{x}_t, \boldsymbol{w})$

        Observe $y_t = f(\boldsymbol{x}_t, \boldsymbol{w}_t) + \varepsilon_t$ at point $(\boldsymbol{x}_t, \boldsymbol{w}_t)$

        Update GP by adding observed data

    **end for**

---

Furthermore, for each $\boldsymbol{x}$, we define a set of functions with respect to $\boldsymbol{w}$, $G_{t-1}^\dagger(\boldsymbol{x})$, as follows:

$$G_{t-1}^\dagger(\boldsymbol{x}) = \{g(\boldsymbol{x}, \cdot) \mid \text{for all } \boldsymbol{w} \in \Omega, g(\boldsymbol{x}, \boldsymbol{w}) \in Q_{t-1}^\dagger(\boldsymbol{x}, \boldsymbol{w})\},$$

where $Q_{t-1}^\dagger(\boldsymbol{x}, \boldsymbol{w}) = [l_{t-1}^\dagger(\boldsymbol{x}, \boldsymbol{w}), u_{t-1}^\dagger(\boldsymbol{x}, \boldsymbol{w})]$. Suppose that $\mathrm{lcb}_{t-1}^\dagger(\boldsymbol{x})$ and $\mathrm{ucb}_{t-1}^\dagger(\boldsymbol{x})$ satisfy the following inequalities:

$$\mathrm{lcb}_{t-1}^\dagger(\boldsymbol{x}) \le \inf_{g(\boldsymbol{x}, \cdot) \in G_{t-1}^\dagger(\boldsymbol{x})} \rho(g(\boldsymbol{x}, \cdot)), \qquad \sup_{g(\boldsymbol{x}, \cdot) \in G_{t-1}^\dagger(\boldsymbol{x})} \rho(g(\boldsymbol{x}, \cdot)) \le \mathrm{ucb}_{t-1}^\dagger(\boldsymbol{x}). \tag{10}$$

For commonly used robustness measures described in Table 3 in Inatsu et al. (2024a), we can use $\mathrm{lcb}_{t-1}^{(m)}(\boldsymbol{x})$ and $\mathrm{ucb}_{t-1}^{(m)}(\boldsymbol{x})$ in the table as $\mathrm{lcb}_{t-1}^\dagger(\boldsymbol{x})$ and $\mathrm{ucb}_{t-1}^\dagger(\boldsymbol{x})$, respectively. Using this, we define the credible interval $Q_{t-1}^\dagger(\boldsymbol{x}) = [\mathrm{lcb}_{t-1}^\dagger(\boldsymbol{x}), \mathrm{ucb}_{t-1}^\dagger(\boldsymbol{x})]$. Here, we emphasize that $Q_{t-1}^\dagger(\boldsymbol{x})$ is the credible interval for $\rho(f_t^\dagger(\boldsymbol{x}, \cdot)) \equiv F_t^\dagger(\boldsymbol{x})$, not $F(\boldsymbol{x})$, where $f_t^\dagger(\boldsymbol{x}, \boldsymbol{w}) \equiv f(\boldsymbol{x}, [\boldsymbol{w}]_t)$. We define the estimated solution $\hat{\boldsymbol{x}}_t^\dagger$ by using $\mu_{t-1}^\dagger(\boldsymbol{x}, \cdot)$ as follows:

$$\hat{\boldsymbol{x}}_t^\dagger = \arg\max_{\boldsymbol{x}\in\mathcal{X}} \rho(\mu_{t-1}^\dagger(\boldsymbol{x}, \cdot)). \tag{11}$$

The next point to be evaluated $\boldsymbol{x}_t$ is selected as follows.

**Definition A.1** (Selection rule for $\boldsymbol{x}_t$ when $\Omega$ is continuous)**.** Suppose that $\xi_1, \ldots, \xi_t$ are random variables following the chi-squared distribution with two degrees of freedom, where $f, \varepsilon_1, \ldots, \varepsilon_t, \xi_1, \ldots, \xi_t$ are mutually independent. For the sequence $\kappa_t^{(2)}$ with $1 \le \kappa_1^{(2)} \le \kappa_2^{(2)} \le \cdots$, we define $\beta_t = 2 \log \kappa_t^{(2)} + \xi_t$. Then, for $\mathrm{lcb}_{t-1}^\dagger(\boldsymbol{x})$ and $\mathrm{ucb}_{t-1}^\dagger(\boldsymbol{x})$ satisfying equation 10, and $\hat{\boldsymbol{x}}_t^\dagger$ given by equation 11, we select $\boldsymbol{x}_t$ as follows:

$$\boldsymbol{x}_t = \arg\max_{\boldsymbol{x}\in\{\tilde{\boldsymbol{x}}_t^\dagger, \hat{\boldsymbol{x}}_t^\dagger\}} (\mathrm{ucb}_{t-1}^\dagger(\boldsymbol{x}) - \mathrm{lcb}_{t-1}^\dagger(\boldsymbol{x})), \tag{12}$$

where $\tilde{\boldsymbol{x}}_t^\dagger = \arg\max_{\boldsymbol{x}\in\mathcal{X}} (\mathrm{ucb}_{t-1}^\dagger(\boldsymbol{x}) - \max_{\boldsymbol{x}\in\mathcal{X}} \mathrm{lcb}_{t-1}^\dagger(\boldsymbol{x}))_+$.

For $\boldsymbol{w}_t$, we use the same rule as in Algorithm 1, that is, $\boldsymbol{w}_t$ is selected by

$$\boldsymbol{w}_t = \arg\max_{\boldsymbol{w}\in\Omega} \sigma_{t-1}^2(\boldsymbol{x}_t, \boldsymbol{w}).$$

The pseudo-code for the proposed method is provided in Algorithm 4.

### A.2.1 Theoretical Analysis in the Continuous Setting when $\mathcal{X}$ is Finite and $\Omega$ is Continuous

First, we introduce the following two assumptions.

**Assumption A.3.** There exist $a_2, b_2 > 0$ such that

$$\mathbb{P}\left(\sup_{\boldsymbol{w}\in\Omega} \left|\frac{\partial f(\boldsymbol{x}, \boldsymbol{w})}{\partial w_j}\right| > L\right) \le a_2 \exp\left(-\left(\frac{L}{b_2}\right)^2\right) \quad \text{for } j \in \{1, \ldots, d_2\}, \boldsymbol{x} \in \mathcal{X}.$$

**Assumption A.4.** There exists a non-decreasing concave function $q_2(a)$ such that $q_2(0) = 0$ and

$$|\rho(f(\boldsymbol{x}, \cdot)) - \rho(f_t^\dagger(\boldsymbol{x}, \cdot))| \leq q_2 \left( \max_{\boldsymbol{w} \in \Omega} |f(\boldsymbol{x}, \boldsymbol{w}) - f(\boldsymbol{x}, [\boldsymbol{w}]_t)| \right)$$

for any $\boldsymbol{x} \in \mathcal{X}$, $\Omega_t$ and $f(\boldsymbol{x}, \boldsymbol{w})$.

For Assumption A.3, if $\boldsymbol{x}$ is fixed, then $f(\boldsymbol{x}, \boldsymbol{w})$ is a GP on $\Omega$. Hence, as in the case of Assumption A.1, we can obtain a sufficient condition for Assumption A.3 by using the derivative of the kernel function. Furthermore, for Assumption A.4, we can use $q^{(m)}(a)$ in Table 4 in Inatsu et al. (2024a) as $q_2(a)$ if the target robustness measure is the Bayes risk (expectation), worst-case, best-case, $\alpha$-value-at-risk, $\alpha$-conditional value-at-risk, or mean absolute deviation measure described in the table. Details are given in Appendix C.5. In addition, for the optimal solution $\boldsymbol{x}^* = \arg\max_{\boldsymbol{x} \in \mathcal{X}} F(\boldsymbol{x})$ and estimated solution $\hat{\boldsymbol{x}}_t^\dagger$, we define the regret $r_t^\dagger$ and cumulative regret $R_t^\dagger$ as follows:

$$r_t^\dagger = F(\boldsymbol{x}^*) - F(\hat{\boldsymbol{x}}_t^\dagger), R_t^\dagger = \sum_{k=1}^t r_k^\dagger.$$

Here, for $\mathrm{ucb}_{t-1}^\dagger(\boldsymbol{x}_t)$, $\mathrm{lcb}_{t-1}^\dagger(\boldsymbol{x}_t)$ and $2\beta_t^{1/2}\sigma_{t-1}(\boldsymbol{x}_t, \boldsymbol{w}_t)$, we introduce the following assumption.

**Assumption A.5.** There exists a function $q^\dagger(x) \in \mathcal{Q}$ such that

$$\mathrm{ucb}_{t-1}^\dagger(\boldsymbol{x}_t) - \mathrm{lcb}_{t-1}^\dagger(\boldsymbol{x}_t) \leq q^\dagger(2\beta_t^{1/2} \max_{\boldsymbol{w} \in \Omega} \sigma_{t-1}(\boldsymbol{x}_t, \boldsymbol{w}))$$

for any $t \geq 1$, $\boldsymbol{x}_t$, $\beta_t$ and $\sigma_{t-1}(\boldsymbol{x}_t, \boldsymbol{w})$.

As in the case of $q(a)$, for the Bayes risk (expectation), worst-case, best-case, $\alpha$-value-at-risk, $\alpha$-conditional value-at-risk, and mean absolute deviation measures described in Table 4 in Inatsu et al. (2024a), we can use $q^{(m)}(a)$ in the table as $q^\dagger(a)$. Then, the following theorem holds.

**Theorem A.3.** Assume that the regularity assumption, Assumptions A.3, A.4 and A.5, and equation 10 hold. Let $\tau_t^\dagger = \lceil b_2 d_2 r t^2 (\sqrt{\log(a_2 d_2 |\mathcal{X}|)} + \sqrt{\pi}/2) \rceil$, and let $\Omega_t$ be a set of discretization for $\Omega$ with each coordinate equally divided into $\tau_t^\dagger$. Suppose that $\xi_1, \ldots, \xi_t$ are random variables following the chi-squared distribution with two degrees of freedom, where $f, \varepsilon_1, \ldots, \varepsilon_t, \xi_1, \ldots, \xi_t$ are mutually independent. Let $\kappa_t^{(2)} = \lceil b_2 d_2 r t^2 (\sqrt{\log(a_2 d_2 |\mathcal{X}|)} + \sqrt{\pi}/2) \rceil^{d_2} |\mathcal{X}|$, and define $\beta_t = 2 \log \kappa_t^{(2)} + \xi_t$. Let $q_2(a)$ and $q^\dagger(a) = \sum_{i=1}^n \zeta_i h_i^\dagger \left( \sum_{j=1}^{s_i} \lambda_{ij} a^{\nu_{ij}} \right)$ be functions satisfying Assumptions A.4 and A.5, respectively. Then, if Algorithm 4 is performed, the following holds:

$$\mathbb{E}[R_t^\dagger] \leq 2t q_2 \left( \frac{\pi^2}{6t} \right) + 2t \sum_{i=1}^n \zeta_i h_i^\dagger \left( \frac{1}{t} \sum_{j=1}^{s_i} 2^{\nu_{ij}} \lambda_{ij} \left( t C_{2,\nu_{ij},t} \right)^{1 - \nu_{ij}'/2} (C_1 \gamma_t)^{\nu_{ij}'/2} \right),$$

where $\nu_{ij}' = \min\{\nu_{ij}, 1\}$, $C_1 = \frac{2}{\log(1 + \sigma_{\mathrm{noise}}^{-2})}$, $C_{2,\nu_{ij},t} = \mathbb{E}[\beta_t^{\nu_{ij}/(2-\nu_{ij}')}]$, and the expectation is taken over all sources of randomness, including $f, \varepsilon_1, \ldots, \varepsilon_t, \beta_1, \ldots, \beta_t$.

Here, since $C_{2,\nu_{ij},t}$ diverges with the order of $(\log t)^{\nu_{ij}/(2-\nu_{ij}')}$, if $\gamma_t$ diverges with the order of $\frac{t}{(\log t)^{\nu_{ij}/\nu_{ij}'}}$ or higher, the argument for the function $h_i^\dagger(\cdot)$ tends to infinity. On the other hand, if the order of $\gamma_t$ is slower than $\frac{t}{(\log t)^{\nu_{ij}/\nu_{ij}'}}$, $\lim_{t \to \infty} \mathbb{E}[R_t^\dagger]/t = 0$ holds if $q_2(\cdot)$ and $h_i^\dagger(\cdot)$ are continuous at 0. Next, for six robustness measures including the expectation measure, the following corollary holds.

**Corollary A.2.** Under the assumptions of Theorem A.3, for the expectation, worst-case, best-case, $\alpha$-value-at-risk, $\alpha$-conditional value-at-risk, and mean absolute deviation measures, the following holds:

$$\mathbb{E}[R_t^\dagger] \leq C \left( \frac{\pi^2}{3} + 4 \sqrt{C_1 t (2 \log \kappa_t^{(2)} + 2) \gamma_t} \right),$$

where $C$ is 2 for the mean absolute deviation measure, and 1 for the other measures.

Here, we define $\hat{t}$ as follows:

$$\hat{t} = \underset{1 \leq i \leq t}{\arg\min} \, \mathbb{E}_{t-1}[F(\boldsymbol{x}^*) - F(\hat{\boldsymbol{x}}_i^\dagger)]. \tag{13}$$

Then, the following theorem holds.

**Theorem A.4.** Under the assumptions of Theorem A.3, the following holds:

$$\mathbb{E}[r_{\hat{t}}^\dagger] \leq \frac{\mathbb{E}[R_t^\dagger]}{t} \leq 2q_2 \left( \frac{\pi^2}{6t} \right) + 2 \sum_{i=1}^n \zeta_i h_i^\dagger \left( \frac{1}{t} \sum_{j=1}^{s_i} 2^{\nu_{ij}} \lambda_{ij} \left( t C_{2,\nu_{ij},t} \right)^{1-\nu_{ij}'/2} \left( C_1 \gamma_t \right)^{\nu_{ij}'/2} \right),$$

where $\hat{t}$ is given by equation 13, and functions $q_2(\cdot)$, $h_i^\dagger(\cdot)$ and all coefficients are as defined in Theorem A.3. Moreover, for the expectation, worst-case, best-case, $\alpha$-value-at-risk, $\alpha$-conditional value-at-risk, and mean absolute deviation measures, the following holds:

$$\mathbb{E}[r_{\hat{t}}^\dagger] \leq C \left( \frac{\pi^2}{3t} + 4 \sqrt{\frac{C_1 (2 \log \kappa_t^{(2)} + 2) \gamma_t}{t}} \right), \tag{14}$$

where $C$ is given in Corollary A.2.

Proofs are given by using the same argument as in the proof of Theorems 4.2 and 4.3. Finally, we consider $\hat{t}$ in the expectation measure. Under the expectation measure, since $\hat{\boldsymbol{x}}_t^\dagger$ corresponds to the posterior mean of $F_t^\dagger(\boldsymbol{x})$, there is a gap with the index $\hat{t}$ given in equation 13. As a result, even in the case of the expectation measure, $\hat{t}$ does not necessarily equal $t$. Nevertheless, the upper bound of $\mathbb{E}[r_t^\dagger]$ can be expressed as the right-hand side in equation 14 plus $2t^{-2}$.

**Theorem A.5.** Under the assumptions of Theorem A.3, for the expectation measure, the following holds:

$$\mathbb{E}[r_t^\dagger] \leq \frac{2}{t^2} + \frac{\pi^2}{3t} + 4 \sqrt{\frac{C_1 (2 \log \kappa_t^{(2)} + 2) \gamma_t}{t}}.$$

The proof is given in Appendix C.9.

## A.3 Extension to Continuous Settings when $\mathcal{X}$ and $\Omega$ are Continuous

Let $\mathcal{X}$ and $\Omega$ be continuous sets. Suppose that both $\mathcal{X}$ and $\Omega$ are compact and convex sets with $\mathcal{X} \times \Omega \subset [0, r]^{d_1 + d_2}$. Let $d = d_1 + d_2$. In this setting, there is no difference from Algorithm 4 in terms of implementation, but the way that the partition of $\Omega$ and theoretical choice of $\kappa_t^{(2)}$ is different. Therefore, by replacing the notations in Algorithm 4, we show the pseudo-code of the proposed method in Algorithm 5.

### A.3.1 Theoretical Analysis in the Continuous Setting when $\mathcal{X}$ and $\Omega$ are Continuous

To derive the theoretical guarantee, we introduce the following two assumptions.

**Assumption A.6.** Let $\mathcal{X} \times \Omega \equiv \Theta$ and $(\boldsymbol{x}, \boldsymbol{w}) \equiv \boldsymbol{\theta}$. Then, there exist $a_3, b_3 > 0$ such that

$$\mathbb{P} \left( \sup_{\boldsymbol{\theta} \in \Theta} \left| \frac{\partial f(\boldsymbol{\theta})}{\partial \theta_j} \right| > L \right) \leq a_3 \exp \left( - \left( \frac{L}{b_3} \right)^2 \right) \quad \text{for } j \in \{1, \ldots, d\}.$$

**Assumption A.7.** There exists a non-decreasing and concave function $q_3(a)$ such that $q_3(0) = 0$ and

$$|\rho(f(\boldsymbol{x}, \cdot)) - \rho(f_t^\dagger(\boldsymbol{x}, \cdot))| \leq q_3 \left( \max_{\boldsymbol{w} \in \Omega} |f(\boldsymbol{x}, \boldsymbol{w}) - f(\boldsymbol{x}, [\boldsymbol{w}]_t)| \right),$$

$$|\rho(f(\boldsymbol{x}, \cdot)) - \rho(f_t^\dagger([\boldsymbol{x}]_t, \cdot))| \leq q_3 \left( \max_{\boldsymbol{w} \in \Omega} |f(\boldsymbol{x}, \boldsymbol{w}) - f([\boldsymbol{x}]_t, [\boldsymbol{w}]_t)| \right)$$

for any $\boldsymbol{x} \in \mathcal{X}$, $\Omega_t$ and $f(\boldsymbol{x}, \boldsymbol{w})$.

---

**Algorithm 5** RRGP-UCB for robustness measures when $\mathcal{X}$ and $\Omega$ are continuous.

---

**Input:** GP prior $\mathcal{GP}(0,\ k)$, $\{\kappa_t^{(3)}\}_{t\in\mathbb{N}}$, $1 \leq \kappa_1^{(3)} \leq \kappa_2^{(3)} \leq \cdots$, finite subsets $\Omega_1, \Omega_2, \ldots \subset \Omega$

  **for** $t = 1, 2, \ldots$ **do**

    Generate $\xi_t$ from chi-squared distribution with two degrees of freedom

    Compute $\beta_t = 2\log\kappa_t^{(3)} + \xi_t$

    Compute $Q_{t-1}^\dagger(\boldsymbol{x}, \boldsymbol{w})$ for each $(\boldsymbol{x}, \boldsymbol{w}) \in \mathcal{X} \times \Omega$

    Compute $Q_{t-1}^\dagger(\boldsymbol{x})$ for each $\boldsymbol{x} \in \mathcal{X}$

    Estimate $\hat{\boldsymbol{x}}_t^\dagger$ by $\hat{\boldsymbol{x}}_t^\dagger = \arg\max_{\boldsymbol{x}\in\mathcal{X}} \rho(\mu_{t-1}^\dagger(\boldsymbol{x}, \cdot))$

    Select next evaluation point $\boldsymbol{x}_t$ by equation 12

    Select next evaluation point $\boldsymbol{w}_t$ by $\boldsymbol{w}_t = \arg\max_{\boldsymbol{w}\in\Omega} \sigma_{t-1}^2(\boldsymbol{x}_t, \boldsymbol{w})$

    Observe $y_t = f(\boldsymbol{x}_t, \boldsymbol{w}_t) + \varepsilon_t$ at point $(\boldsymbol{x}_t, \boldsymbol{w}_t)$

    Update GP by adding observed data

  **end for**

---

For Assumption A.7, we can use $q^{(m)}(a)$ in Table 4 in Inatsu et al. (2024a) as $q_3(a)$ if the target robustness measure is the Bayes risk (expectation), worst-case, best-case, $\alpha$-value-at-risk, $\alpha$-conditional value-at-risk, or mean absolute deviation measures described in the table. Details are described in Appendix C.5. Then, the following theorem holds.

**Theorem A.6.** Assume that the regularity assumption, Assumptions A.5, A.6 and A.7, and equation 10 hold. Let $\tilde{\tau}_t = \lceil b_3 drt^2(\sqrt{\log(a_3 d)} + \sqrt{\pi}/2) \rceil$, and let $\mathcal{X}_t \times \Omega_t$ be a set of discretization for $\mathcal{X} \times \Omega$ with each coordinate equally divided into $\tilde{\tau}_t$. Suppose that $\xi_1, \ldots, \xi_t$ are random variables following the chi-squared distribution with two degrees of freedom, where $f, \varepsilon_1, \ldots, \varepsilon_t, \xi_1, \ldots, \xi_t$ are mutually independent. Let $\kappa_t^{(3)} = (1 + \tilde{\tau}_t^{d_1})\tilde{\tau}_t^{d_2}$, and define $\beta_t = 2\log\kappa_t^{(3)} + \xi_t$. Let $q_3(a)$ and $q^\dagger(a) = \sum_{i=1}^n \zeta_i h_i^\dagger\left(\sum_{j=1}^{s_i} \lambda_{ij} a^{\nu_{ij}}\right)$ be functions satisfying Assumptions A.7 and A.5, respectively. Then, if Algorithm 5 is performed, the following holds:

$$\mathbb{E}[R_t^\dagger] \leq 2tq_3\left(\frac{\pi^2}{6t}\right) + 2t\sum_{i=1}^n \zeta_i h_i^\dagger\left(\frac{1}{t}\sum_{j=1}^{s_i} 2^{\nu_{ij}}\lambda_{ij}\left(tC_{2,\nu_{ij},t}\right)^{1-\nu_{ij}'/2}\left(C_1\gamma_t\right)^{\nu_{ij}'/2}\right),$$

where $\nu_{ij}' = \min\{\nu_{ij}, 1\}$, $C_1 = \frac{2}{\log(1+\sigma_{\text{noise}}^{-2})}$, $C_{2,\nu_{ij},t} = \mathbb{E}[\beta_t^{\nu_{ij}/(2-\nu_{ij}')}]$, and the expectation is taken over all sources of randomness, including $f, \varepsilon_1, \ldots, \varepsilon_t, \beta_1, \ldots, \beta_t$.

The proof is given in Appendix C.10. Here, since $C_{2,\nu_{ij},t}$ diverges with the order of $(\log t)^{\nu_{ij}/(2-\nu_{ij}')}$, if the order of $\gamma_t$ is $\frac{t}{(\log t)^{\nu_{ij}/\nu_{ij}'}}$ or higher, then the argument for the function $h_i^\dagger(\cdot)$ tends to infinity. On the other hand, if the order of $\gamma_t$ is slower than $\frac{t}{(\log t)^{\nu_{ij}/\nu_{ij}'}}$, $\lim_{t\to\infty} \mathbb{E}[R_t^\dagger]/t = 0$ holds if $q_3(\cdot)$ and $h_i^\dagger(\cdot)$ are continuous at 0. Next, for six measures including the expectation measure, the following corollary holds.

**Corollary A.3.** Under the assumptions of Theorem A.6, for the expectation, worst-case, best-case, $\alpha$-value-at-risk, $\alpha$-conditional value-at-risk, and mean absolute deviation measures, the following holds:

$$\mathbb{E}[R_t^\dagger] \leq C\left(\frac{\pi^2}{3} + 4\sqrt{C_1 t(2\log\kappa_t^{(3)} + 2)\gamma_t}\right),$$

where $C$ is 2 for the mean absolute deviation measure, and 1 for the other measures.

Furthermore, for $\hat{t}$ given by equation 13, the following theorem holds.

**Theorem A.7.** Under the assumptions of Theorem A.6, the following holds:

$$\mathbb{E}[r_{\hat{t}}^\dagger] \leq \frac{\mathbb{E}[R_t^\dagger]}{t} \leq 2q_3\left(\frac{\pi^2}{6t}\right) + 2\sum_{i=1}^n \zeta_i h_i^\dagger\left(\frac{1}{t}\sum_{j=1}^{s_i} 2^{\nu_{ij}}\lambda_{ij}\left(tC_{2,\nu_{ij},t}\right)^{1-\nu_{ij}'/2}\left(C_1\gamma_t\right)^{\nu_{ij}'/2}\right),$$

---

**Algorithm 6** RRGP-UCB for robustness measures in the uncontrollable setting when $\Omega$ is finite.

---

**Input:** GP prior $\mathcal{GP}(0, k)$, finite set $\Omega$, $\{\kappa_t^{(4)}\}_{t\in\mathbb{N}}$, $1 \leq \kappa_1^{(4)} \leq \kappa_2^{(4)} \leq \cdots$
  **for** $t = 1, 2, \ldots$ **do**
    Generate $\xi_t$ from chi-squared distribution with two degrees of freedom
    Compute $\beta_t = 2\log\kappa_t^{(4)} + \xi_t$
    Compute $Q_{t-1}(\boldsymbol{x}, \boldsymbol{w})$ for each $(\boldsymbol{x}, \boldsymbol{w}) \in \mathcal{X} \times \Omega$
    Compute $Q_{t-1}(\boldsymbol{x})$ for each $\boldsymbol{x} \in \mathcal{X}$
    Estimate $\hat{\boldsymbol{x}}_t$ by $\hat{\boldsymbol{x}}_t = \arg\max_{\boldsymbol{x}\in\mathcal{X}} \rho(\mu_{t-1}(\boldsymbol{x}, \cdot))$
    Select next evaluation point $\boldsymbol{x}_t$ by equation 4
    Generate $\boldsymbol{w}_t$ from $P^*$
    Observe $y_t = f(\boldsymbol{x}_t, \boldsymbol{w}_t) + \varepsilon_t$ at point $(\boldsymbol{x}_t, \boldsymbol{w}_t)$
    Update GP by adding observed data
  **end for**

---

where $\hat{t}$ is given by equation 13, and functions $q_3(\cdot)$, $h_i^\dagger(\cdot)$ and all coefficients are as defined in Theorem A.6. Moreover, for the expectation, worst-case, best-case, $\alpha$-value-at-risk, $\alpha$-conditional value-at-risk, and mean absolute deviation measures, the following holds:

$$\mathbb{E}[r_{\hat{t}}^\dagger] \leq C\left(\frac{\pi^2}{3t} + 4\sqrt{\frac{C_1(2\log\kappa_t^{(3)} + 2)\gamma_t}{t}}\right),$$

where $C$ is given in Corollary A.3.

The proof is given by using the same argument as in the proof of Theorems 4.2 and 4.3. Finally, for the expectation measure, the following theorem holds.

**Theorem A.8.** Under the assumptions of Theorem A.6, for the expectation measure, the following holds:

$$\mathbb{E}[r_t^\dagger] \leq \frac{2}{t^2} + \frac{\pi^2}{3t} + 4\sqrt{\frac{C_1(2\log\kappa_t^{(3)} + 2)\gamma_t}{t}}.$$

As in Appendix C.9, we can prove Theorem A.8.

## B  Extension of the Proposed Method to Uncontrollable Settings

In this section, we consider the case of uncontrollable settings, i.e., $\boldsymbol{w}_1, \ldots, \boldsymbol{w}_t$ follow the distribution $P^*$ and cannot be controlled even during optimization. Hereafter, we assume that $\boldsymbol{w}_1, \ldots, \boldsymbol{w}_t$ are mutually independent.

### B.1  Extension to Uncontrollable Settings when $\Omega$ is Finite

Let $\Omega$ be a finite set. In this case, if $\mathcal{X}$ is finite or continuous, the only difference between the proposed method and Algorithm 1 or 3 is whether or not $\boldsymbol{w}_t$ is sampled from $P^*$. Moreover, if we set $\kappa_t^{(1)} = |\mathcal{X} \times \Omega|$ in Algorithm 3, we can express the case when $\mathcal{X}$ is finite, i.e., Algorithm 1. We give the pseudo-code for the case where $\Omega$ is finite in the uncontrollable setting in Algorithm 6. Note that the $\mathcal{X}$ in Algorithm 6 includes both the finite and continuous cases.

### B.2  Theoretical Analysis in Uncontrollable Settings when $\Omega$ is Finite

Let $\Omega = \{\boldsymbol{w}^{(1)}, \ldots, \boldsymbol{w}^{(J)}\}$, $p_j = \mathbb{P}_{\boldsymbol{w}}(\boldsymbol{w} = \boldsymbol{w}^{(j)})$ and $p_{min} = \min_{1\leq j\leq J} p_j$. Next, we introduce the following assumption.

**Assumption B.1.** For any $j \in \{1, \ldots, J\}$, $p_j > 0$ holds.

Then, for the upper bound of the inequality for the theoretical analysis for finite or continuous $\mathcal{X}$, it is sufficient to replace $C_1$ in the upper bound of the inequality in the results of Section 4 or Appendix A.1.1 with $C' = p_{min}^{-1} C_1$.

**Theorem B.1.** Assume that the regularity assumption, Assumptions 4.1 and B.1, and equation 2 hold. Suppose that $\mathcal{X}$ and $\Omega$ are finite. Also suppose that $\boldsymbol{w}_1, \ldots, \boldsymbol{w}_t$ follow $P^*$, and $\xi_1, \ldots, \xi_t$ are random variables following the chi-squared distribution with two degrees of freedom, where $f, \varepsilon_1, \ldots, \varepsilon_t, \boldsymbol{w}_1, \ldots, \boldsymbol{w}_t$, $\xi_1, \ldots, \xi_t$ are mutually independent. Let $\kappa_t^{(4)} = |\mathcal{X} \times \Omega|$, and define $\beta_t = 2 \log \kappa_t^{(4)} + \xi_t$. Let $q(a) = \sum_{i=1}^{n} \zeta_i h_i \left( \sum_{j=1}^{s_i} \lambda_{ij} a^{\nu_{ij}} \right)$ be a function satisfying Assumption 4.1. Then, under the uncontrollable setting, if Algorithm 6 is performed, the following holds:

$$\mathbb{E}[R_t] \leq 2t \sum_{i=1}^{n} \zeta_i h_i \left( \frac{1}{t} \sum_{j=1}^{s_i} 2^{\nu_{ij}} \lambda_{ij} \left( t C_{2,\nu_{ij}} \right)^{1-\nu'_{ij}/2} \left( C'_1 \gamma_t \right)^{\nu'_{ij}/2} \right),$$

where $\nu'_{ij} = \min\{\nu_{ij}, 1\}, C'_1 = \frac{2 p_{min}^{-1}}{\log(1+\sigma_{\text{noise}}^{-2})}, C_{2,\nu_{ij}} = \mathbb{E}[\beta_t^{\nu_{ij}/(2-\nu'_{ij})}]$, and the expectation is taken over all sources of randomness, including $f, \varepsilon_1, \ldots, \varepsilon_t, \boldsymbol{w}_1, \ldots, \boldsymbol{w}_t, \beta_1, \ldots, \beta_t$. In addition, for the expectation, worst-case, best-case, $\alpha$-value-at-risk, $\alpha$-conditional value-at-risk, and mean absolute deviation measures, the following inequality holds:

$$\mathbb{E}[R_t] \leq C \sqrt{t C'_1 (2 \log(|\mathcal{X} \times \Omega|) + 2) \gamma_t},$$

where $C$ is 8 for the mean absolute deviation measure and 4 for the other measures.

**Theorem B.2.** Under the assumptions of Theorem B.1, the following holds:

$$\mathbb{E}[r_{\hat{t}}] \leq \frac{\mathbb{E}[R_t]}{t} \leq 2 \sum_{i=1}^{n} \zeta_i h_i \left( \frac{1}{t} \sum_{j=1}^{s_i} 2^{\nu_{ij}} \lambda_{ij} \left( t C_{2,\nu_{ij}} \right)^{1-\nu'_{ij}/2} \left( C'_1 \gamma_t \right)^{\nu'_{ij}/2} \right),$$

where $\hat{t}$ is given by equation 9, and the function $h_i(\cdot)$ and all coefficients are as defined in Theorem B.1. In addition, for the expectation, worst-case, best-case, $\alpha$-value-at-risk, $\alpha$-conditional value-at-risk, and mean absolute deviation measures, the following holds:

$$\mathbb{E}[r_{\hat{t}}] \leq \frac{\mathbb{E}[R_t]}{t} \leq C \sqrt{\frac{C'_1 (2 \log(|\mathcal{X} \times \Omega|) + 2) \gamma_t}{t}},$$

where $C$ is given in Theorem B.1. Furthermore, in the expectation measure, the following holds:

$$\mathbb{E}[r_t] \leq \frac{\mathbb{E}[R_t]}{t} \leq 4 \sqrt{\frac{C'_1 (2 \log(|\mathcal{X} \times \Omega|) + 2) \gamma_t}{t}}.$$

**Theorem B.3.** Assume that the regularity assumption, Assumptions 4.1, A.1, A.2 and B.1, and equation 2 hold. Suppose that $\mathcal{X}$ and $\Omega$ are continuous and finite, respectively. Also suppose that $\boldsymbol{w}_1, \ldots, \boldsymbol{w}_t$ follow $P^*$, and $\xi_1, \ldots, \xi_t$ are random variables following the chi-squared distribution with two degrees of freedom, where $f, \varepsilon_1, \ldots, \varepsilon_t, \boldsymbol{w}_1, \ldots, \boldsymbol{w}_t, \xi_1, \ldots, \xi_t$ are mutually independent. Let $\kappa_t^{(4)} = (1 + \lceil b_1 d_1 r t^2 (\sqrt{\log(a_1 d_1 |\Omega|)} + \sqrt{\pi}/2) \rceil^{d_1}) |\Omega|$, and define $\beta_t = 2 \log \kappa_t^{(4)} + \xi_t$. Let $q_1(a)$ and $q(a) = \sum_{i=1}^{n} \zeta_i h_i \left( \sum_{j=1}^{s_i} \lambda_{ij} a^{\nu_{ij}} \right)$ be functions satisfying Assumptions A.2 and 4.1, respectively. Then, under the uncontrollable setting, if Algorithm 6 is performed, the following holds:

$$\mathbb{E}[R_t] \leq t q_1 \left( \frac{\pi^2}{6t} \right) + 2t \sum_{i=1}^{n} \zeta_i h_i \left( \frac{1}{t} \sum_{j=1}^{s_i} 2^{\nu_{ij}} \lambda_{ij} \left( t C_{2,\nu_{ij},t} \right)^{1-\nu'_{ij}/2} \left( C'_1 \gamma_t \right)^{\nu'_{ij}/2} \right),$$

where $\nu'_{ij} = \min\{\nu_{ij}, 1\}, C'_1 = \frac{2 p_{min}^{-1}}{\log(1+\sigma_{\text{noise}}^{-2})}, C_{2,\nu_{ij},t} = \mathbb{E}[\beta_t^{\nu_{ij}/(2-\nu'_{ij})}]$, and the expectation is taken over all sources of randomness, including $f, \varepsilon_1, \ldots, \varepsilon_t, \boldsymbol{w}_1, \ldots, \boldsymbol{w}_t, \beta_1, \ldots, \beta_t$. In addition, for the expectation,

---

**Algorithm 7** RRGP-UCB for robustness measures in the uncontrollable setting when $\Omega$ is continuous.

---

**Input:** GP prior $\mathcal{GP}(0,\ k)$, continuous set $\Omega$, $\{\kappa_t^{(5)}\}_{t\in\mathbb{N}}$, $1 \leq \kappa_1^{(5)} \leq \kappa_2^{(5)} \leq \cdots$, finite subsets $\Omega_1, \Omega_2, \ldots \subset \Omega$

    **for** $t = 1, 2, \ldots$ **do**
        Generate $\xi_t$ from chi-squared distribution with two degrees of freedom
        Compute $\beta_t = 2\log\kappa_t^{(5)} + \xi_t$
        Compute $Q_{t-1}^{\dagger}(\boldsymbol{x}, \boldsymbol{w})$ for each $(\boldsymbol{x}, \boldsymbol{w}) \in \mathcal{X} \times \Omega$
        Compute $Q_{t-1}^{\dagger}(\boldsymbol{x})$ for each $\boldsymbol{x} \in \mathcal{X}$
        Estimate $\hat{\boldsymbol{x}}_t^{\dagger}$ by $\hat{\boldsymbol{x}}_t^{\dagger} = \arg\max_{\boldsymbol{x}\in\mathcal{X}} \rho(\mu_{t-1}^{\dagger}(\boldsymbol{x}, \cdot))$
        Select next evaluation point $\boldsymbol{x}_t$ by equation 12
        Generate $\boldsymbol{w}_t$ from $P^*$
        Observe $y_t = f(\boldsymbol{x}_t, \boldsymbol{w}_t) + \varepsilon_t$ at point $(\boldsymbol{x}_t, \boldsymbol{w}_t)$
        Update GP by adding observed data
    **end for**

---

worst-case, best-case, $\alpha$-value-at-risk, $\alpha$-conditional value-at-risk, and mean absolute deviation measures, the following holds:

$$\mathbb{E}[R_t] \leq C\left(\frac{\pi^2}{6} + 4\sqrt{C_1' t(2\log\kappa_t^{(4)} + 2)\gamma_t}\right),$$

where $C$ is 2 for the mean absolute deviation measure, and 1 for the other measures.

**Theorem B.4.** Under the assumptions of Theorem B.3, the following holds:

$$\mathbb{E}[r_{\hat{t}}] \leq \frac{\mathbb{E}[R_t]}{t} \leq q_1\left(\frac{\pi^2}{6t}\right) + 2\sum_{i=1}^{n}\zeta_i h_i\left(\frac{1}{t}\sum_{j=1}^{s_i} 2^{\nu_{ij}}\lambda_{ij}\left(tC_{2,\nu_{ij},t}\right)^{1-\nu_{ij}'/2}\left(C_1'\gamma_t\right)^{\nu_{ij}'/2}\right),$$

where $\hat{t}$ is given by equation 9, and functions $q_1(\cdot)$, $h_i(\cdot)$ and all coefficients are as defined in Theorem B.3. In addition, for the expectation, worst-case, best-case, $\alpha$-value-at-risk, $\alpha$-conditional value-at-risk, and mean absolute deviation measures, the following holds:

$$\mathbb{E}[r_{\hat{t}}] \leq C\left(\frac{\pi^2}{6t} + 4\sqrt{\frac{C_1'(2\log\kappa_t^{(4)} + 2)\gamma_t}{t}}\right),$$

where, $C$ is given in Theorem B.3. Moreover, for the expectation measure, $\hat{t}$ satisfies $\hat{t} = t$, i.e., the following holds:

$$\mathbb{E}[r_t] \leq \frac{\pi^2}{6t} + 4\sqrt{\frac{C_1'(2\log\kappa_t^{(4)} + 2)\gamma_t}{t}}.$$

Proofs are given in Appendix C.11.

## B.3 Extension to Uncontrollable Settings when $\Omega$ is Continuous

Let $\Omega$ be a continuous set. In this case, if $\mathcal{X}$ is finite or continuous, the only difference between the proposed method and Algorithm 4 or 5 is whether or not $\boldsymbol{w}_t$ is sampled from $P^*$. We give the pseudo-code for the case where $\Omega$ is continuous in the uncontrollable setting in Algorithm 7. Note that $\mathcal{X}$ in Algorithm 7 includes both the finite and continuous cases.

## B.4 Theoretical Analysis in Uncontrollable Settings when $\Omega$ is Continuous

First, we introduce a similar assumption to Assumption B.1. When $\Omega$ is finite, Assumption B.1 means that there is no $\boldsymbol{w}^{(j)} \in \Omega$ such that $\mathbb{P}_{\boldsymbol{w}}(\boldsymbol{w} = \boldsymbol{w}^{(j)}) = 0$, and this requires that the points that cannot be realized values of $\boldsymbol{w}$ are not included in $\Omega$. On the other hand, when $\Omega$ is continuous, a similar assumption is that there is no $\boldsymbol{a} \in \Omega$ and $\epsilon > 0$ such that $\mathbb{P}_{\boldsymbol{w}}(\boldsymbol{w} \in \text{Nei}(\boldsymbol{a}; \epsilon)) = 0$, where $\text{Nei}(\boldsymbol{a}; \epsilon)$ is the open ball with center $\boldsymbol{a}$ and radius $\epsilon > 0$. Therefore, we introduce the following assumption:

**Assumption B.2.** For any $\boldsymbol{a} \in \Omega$ and $\epsilon > 0$, $\mathbb{P}_{\boldsymbol{w}}(\boldsymbol{w} \in \mathrm{Nei}(\boldsymbol{a}; \epsilon)) > 0$ holds.

Furthermore, we introduce a new assumption on the partition of $\Omega$. Here, let $\mathcal{S} = \{\tilde{\Omega}_1, \ldots, \tilde{\Omega}_s\}$ be a family of subsets in $\Omega$. Then, $\mathcal{S}$ is the partition of $\Omega$ if $\mathcal{S}$ satisfies $\bigcup_{i=1}^{s} \tilde{\Omega}_i = \Omega$ and $\tilde{\Omega}_i \cap \tilde{\Omega}_j = \emptyset$ for any $i \neq j$.

**Assumption B.3.** There exist partitions $\mathcal{S}_1, \mathcal{S}_2, \ldots$ of $\Omega$ satisfying the following two conditions:

1. For any $t \geq 1$, $p_{min,t} \equiv \min_{1 \leq i \leq t} \min_{\tilde{\Omega} \in \mathcal{S}_i} \mathbb{P}_{\boldsymbol{w}}(\boldsymbol{w} \in \tilde{\Omega}) > 0$.

2. There exists a non-stochastic sequence $\iota_1, \iota_2, \ldots$ such that

$$|\sigma_{t-1}^2(\boldsymbol{x}_t, \boldsymbol{a}) - \sigma_{t-1}^2(\boldsymbol{x}_t, \boldsymbol{b})| \leq \iota_t$$

for any $t \geq 1$, $\{(\boldsymbol{x}_1, \boldsymbol{w}_1, y_1, \beta_1), \ldots, (\boldsymbol{x}_{t-1}, \boldsymbol{w}_{t-1}, y_{t-1}, \beta_{t-1}), \boldsymbol{x}_t, \beta_t\}$, $\tilde{\Omega} \in \mathcal{S}_t$ and $\boldsymbol{a}, \boldsymbol{b} \in \tilde{\Omega}$.

Then, the following theorem holds.

**Theorem B.5.** Assume that the regularity assumption, Assumptions A.3, A.4, A.5, B.2 and B.3, and equation 10 hold. Suppose that $\mathcal{X}$ and $\Omega$ are finite and continuous, respectively. Let $\tau_t^\dagger = \lceil b_2 d_2 r t^2 (\sqrt{\log(a_2 d_2 |\mathcal{X}|)} + \sqrt{\pi}/2) \rceil$, and let $\Omega_t$ be a set of discretization for $\Omega$ with each coordinate equally divided into $\tau_t^\dagger$. Suppose that $\boldsymbol{w}_1, \ldots, \boldsymbol{w}_t$ follow $P^*$, and $\xi_1, \ldots, \xi_t$ are random variables following the chi-squared distribution with two degrees of freedom, where $f, \varepsilon_1, \ldots, \varepsilon_t, \boldsymbol{w}_1, \ldots, \boldsymbol{w}_t, \xi_1, \ldots, \xi_t$ are mutually independent. Define $\kappa_t^{(5)} = \lceil b_2 d_2 r t^2 (\sqrt{\log(a_2 d_2 |\mathcal{X}|)} + \sqrt{\pi}/2) \rceil^{d_2} |\mathcal{X}|$ and $\beta_t = 2 \log \kappa_t^{(5)} + \xi_t$. Let $q_2(a)$ and $q^\dagger(a) = \sum_{i=1}^{n} \zeta_i h_i^\dagger \left( \sum_{j=1}^{s_i} \lambda_{ij} a^{\nu_{ij}} \right)$ be functions satisfying Assumptions A.4 and A.5, respectively. For the sequence $\iota_1, \ldots, \iota_t$ satisfying Assumption B.3, define $\varphi_t = \iota_1 + \cdots + \iota_t$. Then, under the uncontrollable setting, if Algorithm 7 is performed, the following holds:

$$\mathbb{E}[R_t^\dagger] \leq 2 t q_2 \left( \frac{\pi^2}{6t} \right) + 2t \sum_{i=1}^{n} \zeta_i h_i^\dagger \left( \frac{1}{t} \sum_{j=1}^{s_i} 2^{\nu_{ij}} \lambda_{ij} \left( t C_{2,\nu_{ij},t} \right)^{1 - \nu_{ij}'/2} (\varphi_t + p_{min,t}^{-1} C_1 \gamma_t)^{\nu_{ij}'/2} \right),$$

where $\nu_{ij}' = \min\{\nu_{ij}, 1\}$, $C_1 = \frac{2}{\log(1 + \sigma_{\mathrm{noise}}^{-2})}$, $C_{2,\nu_{ij},t} = \mathbb{E}[\beta_t^{\nu_{ij}/(2 - \nu_{ij}')}]$, $p_{min,t}^{-1}$ is given in Assumption B.3, and the expectation is taken over all sources of randomness, including $f, \varepsilon_1, \ldots, \varepsilon_t, \boldsymbol{w}_1, \ldots, \boldsymbol{w}_t, \beta_1, \ldots, \beta_t$. In addition, for the expectation, worst-case, best-case, $\alpha$-value-at-risk, $\alpha$-conditional value-at-risk, and mean absolute deviation measures, the following holds:

$$\mathbb{E}[R_t^\dagger] \leq C \left( \frac{\pi^2}{3} + 4 \sqrt{t(2 \log \kappa_t^{(5)} + 2)(\varphi_t + p_{min,t}^{-1} C_1 \gamma_t)} \right),$$

where $C$ is 2 for the mean absolute deviation measure, and 1 for the other measures.

**Theorem B.6.** Under the assumptions of Theorem B.5, for $\hat{t}$ defined by equation 13, the following holds:

$$\mathbb{E}[r_{\hat{t}}^\dagger] \leq \frac{\mathbb{E}[R_t^\dagger]}{t} \leq 2 q_2 \left( \frac{\pi^2}{6t} \right) + 2 \sum_{i=1}^{n} \zeta_i h_i^\dagger \left( \frac{1}{t} \sum_{j=1}^{s_i} 2^{\nu_{ij}} \lambda_{ij} \left( t C_{2,\nu_{ij},t} \right)^{1 - \nu_{ij}'/2} (\varphi_t + p_{min,t}^{-1} C_1 \gamma_t)^{\nu_{ij}'/2} \right).$$

In addition, for the expectation, worst-case, best-case, $\alpha$-value-at-risk, $\alpha$-conditional value-at-risk, and mean absolute deviation measures, the following holds:

$$\mathbb{E}[r_{\hat{t}}^\dagger] \leq C \left( \frac{\pi^2}{3t} + 4 \sqrt{\frac{(2 \log \kappa_t^{(5)} + 2)(\varphi_t + p_{min,t}^{-1} C_1 \gamma_t)}{t}} \right).$$

Moreover, for the expectation measure, the following holds:

$$\mathbb{E}[r_{\hat{t}}^\dagger] \leq \frac{2}{t^2} + \frac{\pi^2}{3t} + 4 \sqrt{\frac{(2 \log \kappa_t^{(5)} + 2)(\varphi_t + p_{min,t}^{-1} C_1 \gamma_t)}{t}}.$$

**Theorem B.7.** Assume that the regularity assumption, Assumptions A.5, A.6, A.7, B.2 and B.3, and equation 10 hold. Suppose that $\mathcal{X}$ and $\Omega$ are continuous. Let $\tilde{\tau}_t = \lceil b_3 d r t^2 (\sqrt{\log(a_3 d)} + \sqrt{\pi}/2) \rceil$, and let $\mathcal{X}_t \times \Omega_t$ be a set of discretization for $\mathcal{X} \times \Omega$ with each coordinate equally divided into $\tilde{\tau}_t$. Suppose that $\boldsymbol{w}_1, \ldots, \boldsymbol{w}_t$ follow $P^*$, and $\xi_1, \ldots, \xi_t$ are random variables following the chi-squared distribution with two degrees of freedom, where $f, \varepsilon_1, \ldots, \varepsilon_t, \boldsymbol{w}_1, \ldots, \boldsymbol{w}_t, \xi_1, \ldots, \xi_t$ are mutually independent. Define $\kappa_t^{(5)} = (1 + \tilde{\tau}_t^{d_1}) \tilde{\tau}_t^{d_2}$ and $\beta_t = 2 \log \kappa_t^{(5)} + \xi_t$. Let $q_3(a)$ and $q^\dagger(a) = \sum_{i=1}^n \zeta_i h_i^\dagger \left( \sum_{j=1}^{s_i} \lambda_{ij} a^{\nu_{ij}} \right)$ be functions satisfying Assumptions A.7 and A.5, respectively. For the sequence $\iota_1, \ldots, \iota_t$ satisfying Assumption B.3, let $\varphi_t = \iota_1 + \cdots + \iota_t$. Then, under the uncontrollable setting, if Algorithm 7 is performed, the following holds:

$$\mathbb{E}[R_t^\dagger] \leq 2t q_3 \left( \frac{\pi^2}{6t} \right) + 2t \sum_{i=1}^n \zeta_i h_i^\dagger \left( \frac{1}{t} \sum_{j=1}^{s_i} 2^{\nu_{ij}} \lambda_{ij} \left( t C_{2, \nu_{ij}, t} \right)^{1 - \nu_{ij}'/2} \left( \varphi_t + p_{min,t}^{-1} C_1 \gamma_t \right)^{\nu_{ij}'/2} \right),$$

where $\nu_{ij}' = \min\{\nu_{ij}, 1\}$, $C_1 = \frac{2}{\log(1 + \sigma_{\text{noise}}^{-2})}$, $C_{2, \nu_{ij}, t} = \mathbb{E}[\beta_t^{\nu_{ij}/(2 - \nu_{ij})}]$, $p_{min,t}^{-1}$ is given in Assumption B.3, and the expectation is taken over all sources of randomness, including $f, \varepsilon_1, \ldots, \varepsilon_t, \boldsymbol{w}_1, \ldots, \boldsymbol{w}_t, \beta_1, \ldots, \beta_t$. In addition, for the expectation, worst-case, best-case, $\alpha$-value-at-risk, $\alpha$-conditional value-at-risk, and mean absolute deviation measures, the following holds:

$$\mathbb{E}[R_t^\dagger] \leq C \left( \frac{\pi^2}{3} + 4 \sqrt{t(2 \log \kappa_t^{(5)} + 2)(\varphi_t + p_{min,t}^{-1} C_1 \gamma_t)} \right),$$

where $C$ is 2 for the mean absolute deviation measure, and 1 for the other measures.

**Theorem B.8.** Under the assumptions of Theorem B.7, for $\hat{t}$ defined by equation 13, the following holds:

$$\mathbb{E}[r_{\hat{t}}^\dagger] \leq \frac{\mathbb{E}[R_t^\dagger]}{t} \leq 2 q_3 \left( \frac{\pi^2}{6t} \right) + 2 \sum_{i=1}^n \zeta_i h_i^\dagger \left( \frac{1}{t} \sum_{j=1}^{s_i} 2^{\nu_{ij}} \lambda_{ij} \left( t C_{2, \nu_{ij}, t} \right)^{1 - \nu_{ij}'/2} \left( \varphi_t + p_{min,t}^{-1} C_1 \gamma_t \right)^{\nu_{ij}'/2} \right).$$

In addition, for the expectation, worst-case, best-case, $\alpha$-value-at-risk, $\alpha$-conditional value-at-risk, and mean absolute deviation measures, the following holds:

$$\mathbb{E}[r_{\hat{t}}^\dagger] \leq C \left( \frac{\pi^2}{3t} + 4 \sqrt{\frac{(2 \log \kappa_t^{(5)} + 2)(\varphi_t + p_{min,t}^{-1} C_1 \gamma_t)}{t}} \right).$$

Moreover, for the expectation measure, the following holds:

$$\mathbb{E}[r_{\hat{t}}^\dagger] \leq \frac{2}{t^2} + \frac{\pi^2}{3t} + 4 \sqrt{\frac{(2 \log \kappa_t^{(5)} + 2)(\varphi_t + p_{min,t}^{-1} C_1 \gamma_t)}{t}}.$$

Proofs are described in Appendix C.12. In the next section, we provide specific examples that satisfy Assumption B.3.

## B.5 Specific Examples Satisfying Assumption B.3

For simplicity, assume that $\mathcal{X} = \Omega = [0, 1]$. Let $P^*$ be the uniform distribution on $\Omega$. For each $t \geq 1$, we define $\varrho_t = \lceil t^{1/2} \rceil$. Here, we consider a partition of $\Omega$ into $\varrho_t$ intervals with length $\varrho_t^{-1}$, that is, $\mathcal{S}_t = \{[(j-1)/\varrho_t, j/\varrho_t) \mid j = 1, \ldots, \varrho_t - 1\} \cup \{[(\varrho_t - 1)/\varrho_t, 1]\}$. Then, for any $\tilde{\Omega}_j \in \mathcal{S}_t$, $p_{min,t} = \varrho_t^{-1} > 0$ because $\mathbb{P}_{\boldsymbol{w}}(\boldsymbol{w} \in \tilde{\Omega}_j) = \varrho_t^{-1}$. Next, since the difference between posterior variances satisfies

$$\sigma_{t-1}^2(\boldsymbol{x}_t, \boldsymbol{a}) - \sigma_{t-1}^2(\boldsymbol{x}_t, \boldsymbol{b}) = (\sigma_{t-1}(\boldsymbol{x}_t, \boldsymbol{a}) + \sigma_{t-1}(\boldsymbol{x}_t, \boldsymbol{b}))(\sigma_{t-1}(\boldsymbol{x}_t, \boldsymbol{a}) - \sigma_{t-1}(\boldsymbol{x}_t, \boldsymbol{b})),$$

the following holds:

$$|\sigma_{t-1}^2(\boldsymbol{x}_t, \boldsymbol{a}) - \sigma_{t-1}^2(\boldsymbol{x}_t, \boldsymbol{b})| = |\sigma_{t-1}(\boldsymbol{x}_t, \boldsymbol{a}) + \sigma_{t-1}(\boldsymbol{x}_t, \boldsymbol{b})||\sigma_{t-1}(\boldsymbol{x}_t, \boldsymbol{a}) - \sigma_{t-1}(\boldsymbol{x}_t, \boldsymbol{b})| \leq 2|\sigma_{t-1}(\boldsymbol{x}_t, \boldsymbol{a}) - \sigma_{t-1}(\boldsymbol{x}_t, \boldsymbol{b})|.$$

Furthermore, from Theorem E.4 in Kusakawa et al. (2022), for linear, Gaussian and Matérn kernels, the posterior standard deviation is a $K$-Lipschitz continuous function for any $t \geq 1$ and observed data. Therefore, if the true kernel function is the Gaussian kernel $k((\boldsymbol{x}, \boldsymbol{w}), (\boldsymbol{x}', \boldsymbol{w}')) \equiv k(\boldsymbol{\theta}, \boldsymbol{\theta}') = \exp(-\|\boldsymbol{\theta} - \boldsymbol{\theta}'\|_2^2/2)$, the following inequality holds:

$$|\sigma_{t-1}(\boldsymbol{x}_t, \boldsymbol{a}) - \sigma_{t-1}(\boldsymbol{x}_t, \boldsymbol{b})| \leq \sqrt{2}\|\boldsymbol{a} - \boldsymbol{b}\|_1.$$

Moreover, since the length of $\tilde{\Omega}_j$ is $\varrho_t^{-1}$, the following holds:

$$|\sigma_{t-1}^2(\boldsymbol{x}_t, \boldsymbol{a}) - \sigma_{t-1}^2(\boldsymbol{x}_t, \boldsymbol{b})| \leq 2|\sigma_{t-1}(\boldsymbol{x}_t, \boldsymbol{a}) - \sigma_{t-1}(\boldsymbol{x}_t, \boldsymbol{b})| \leq 2\sqrt{2}\|\boldsymbol{a} - \boldsymbol{b}\|_1 = 2\sqrt{2}\varrho_t^{-1}.$$

This implies that $\iota_t = 2\sqrt{2}\varrho_t^{-1}$. Hence, the following inequality holds:

$$\varphi_t = 2\sqrt{2}\sum_{k=1}^{t} \varrho_k^{-1} \leq 2\sqrt{2}\sum_{k=1}^{t} k^{-1/2} \leq 2\sqrt{2}\left(1 + \int_1^t k^{-1/2}\mathrm{d}k\right) = 2\sqrt{2}(1 + 2\sqrt{t} - 2) \leq 6\sqrt{t}.$$

Thus, by substituting $\varphi_t \leq 6\sqrt{t}$ and $p_{min,t}^{-1} = \varrho_t = \lceil t^{1/2}\rceil$ into Theorem B.7, the following holds for the expectation measure:

$$\mathbb{E}[R_t^\dagger] \leq \frac{\pi^2}{3} + 4\sqrt{t(2\log \kappa_t^{(5)} + 2)(6\sqrt{t} + \lceil t^{1/2}\rceil C_1\gamma_t)}.$$

From the definition, $\kappa_t^{(5)}$ satisfies $\log \kappa_t^{(5)} = O(\log t)$. In addition, from Theorem 5 in Srinivas et al. (2010), for the Gaussian kernel, $\gamma_t$ satisfies $\gamma_t = O((\log t)^3)$. Therefore, we obtain

$$\mathbb{E}[R_t^\dagger] = O(t^{3/4}(\log t)^2).$$

## C Proofs

### C.1 Proof of Lemma 3.1

*Proof.* The value $\max_{\boldsymbol{x}\in\mathcal{X}} l_{t-1}(\boldsymbol{x})$ is a constant that does not depend on $\boldsymbol{x}$. This implies that $\tilde{\boldsymbol{x}}_t^{(f)} = \boldsymbol{x}_t^{(\mathrm{u})}$. Therefore, if $\boldsymbol{x}_t^{(f)} = \tilde{\boldsymbol{x}}_t^{(f)}$, then $u_{t-1}(\boldsymbol{x}_t^{(\mathrm{u})}) = u_{t-1}(\boldsymbol{x}_t^{(f)})$. On the other hand, if $\boldsymbol{x}_t^{(f)} = \hat{\boldsymbol{x}}_t^{(f)}$, that is,

$$2\beta_t^{1/2}\sigma_{t-1}(\tilde{\boldsymbol{x}}_t^{(f)}) = u_{t-1}(\tilde{\boldsymbol{x}}_t^{(f)}) - l_{t-1}(\tilde{\boldsymbol{x}}_t^{(f)}) \leq u_{t-1}(\hat{\boldsymbol{x}}_t^{(f)}) - l_{t-1}(\hat{\boldsymbol{x}}_t^{(f)}) = 2\beta_t^{1/2}\sigma_{t-1}(\hat{\boldsymbol{x}}_t^{(f)}),$$

then $\beta_t^{1/2}\sigma_{t-1}(\tilde{\boldsymbol{x}}_t^{(f)}) \leq \beta_t^{1/2}\sigma_{t-1}(\hat{\boldsymbol{x}}_t^{(f)})$. From the definition of $\hat{\boldsymbol{x}}_t^{(f)}$, $\mu_{t-1}(\tilde{\boldsymbol{x}}_t^{(f)}) \leq \mu_{t-1}(\hat{\boldsymbol{x}}_t^{(f)})$. Since $\beta_t^{1/2}\sigma_{t-1}(\tilde{\boldsymbol{x}}_t^{(f)}) \leq \beta_t^{1/2}\sigma_{t-1}(\hat{\boldsymbol{x}}_t^{(f)})$, the inequality $\mu_{t-1}(\tilde{\boldsymbol{x}}_t^{(f)}) + \beta_t^{1/2}\sigma_{t-1}(\tilde{\boldsymbol{x}}_t^{(f)}) \leq \mu_{t-1}(\hat{\boldsymbol{x}}_t^{(f)}) + \beta_t^{1/2}\sigma_{t-1}(\hat{\boldsymbol{x}}_t^{(f)})$ holds. This implies that $u_{t-1}(\tilde{\boldsymbol{x}}_t^{(f)}) \leq u_{t-1}(\hat{\boldsymbol{x}}_t^{(f)})$. Finally, since $\tilde{\boldsymbol{x}}_t^{(f)} = \boldsymbol{x}_t^{(\mathrm{u})}$, $u_{t-1}(\boldsymbol{x}_t^{(\mathrm{u})}) \leq u_{t-1}(\hat{\boldsymbol{x}}_t^{(f)})$ and $u_{t-1}(\boldsymbol{x}_t^{(\mathrm{u})}) = u_{t-1}(\hat{\boldsymbol{x}}_t^{(f)})$ hold. $\square$

### C.2 Proof of Theorem 4.1

*Proof.* For $t \geq 1$, we define $D_{t-1} = \{(\boldsymbol{x}_1, \boldsymbol{w}_1, y_1, \beta_1), \ldots, (\boldsymbol{x}_{t-1}, \boldsymbol{w}_{t-1}, y_{t-1}, \beta_{t-1})\}$ and $D_0 = \emptyset$. Let $\xi_t$ be a realization from the chi-squared distribution with two degrees of freedom, and $\delta = \frac{1}{\exp(\xi_t/2)}$. Then, from the proof of Lemma 5.1 in Srinivas et al. (2010), with probability at least $1 - \delta$, the following holds for any $(\boldsymbol{x}, \boldsymbol{w}) \in \mathcal{X} \times \Omega$:

$$l_{t-1,\delta}(\boldsymbol{x}, \boldsymbol{w}) \equiv \mu_{t-1}(\boldsymbol{x}, \boldsymbol{w}) - \beta_\delta^{1/2}\sigma_{t-1}(\boldsymbol{x}, \boldsymbol{w}) \leq f(\boldsymbol{x}, \boldsymbol{w}) \leq \mu_{t-1}(\boldsymbol{x}, \boldsymbol{w}) + \beta_\delta^{1/2}\sigma_{t-1}(\boldsymbol{x}, \boldsymbol{w}) \equiv u_{t-1,\delta}(\boldsymbol{x}, \boldsymbol{w}),$$

where $\beta_\delta = 2\log(|\mathcal{X} \times \Omega|/\delta)$. From the definition of $\delta$, since $\beta_\delta = 2\log(|\mathcal{X} \times \Omega|) + \xi_t = \beta_t$, the following inequality holds with probability at least $1 - \delta$:

$$l_{t-1}(\boldsymbol{x}, \boldsymbol{w}) \leq f(\boldsymbol{x}, \boldsymbol{w}) \leq u_{t-1}(\boldsymbol{x}, \boldsymbol{w}).$$

Hence, $f(\boldsymbol{x}, \cdot) \in G_{t-1}(\boldsymbol{x})$ holds. In addition, from the theorem's assumption, since $\mathrm{ucb}_{t-1}(\boldsymbol{x})$ and $\mathrm{lcb}_{t-1}(\boldsymbol{x})$ satisfy equation 2, the following holds:

$$\mathrm{lcb}_{t-1}(\boldsymbol{x}) \leq F(\boldsymbol{x}) \leq \mathrm{ucb}_{t-1}(\boldsymbol{x}).$$

Therefore, for $F(\boldsymbol{x}^*) - F(\hat{\boldsymbol{x}}_t)$, the following inequality holds:

$$
\begin{aligned}
F(\boldsymbol{x}^*) - F(\hat{\boldsymbol{x}}_t) &= (F(\boldsymbol{x}^*) - \max_{\boldsymbol{x} \in \mathcal{X}} \mathrm{lcb}_{t-1}(\boldsymbol{x})) + (\max_{\boldsymbol{x} \in \mathcal{X}} \mathrm{lcb}_{t-1}(\boldsymbol{x}) - F(\hat{\boldsymbol{x}}_t)) \\
&\leq (\mathrm{ucb}_{t-1}(\boldsymbol{x}^*) - \max_{\boldsymbol{x} \in \mathcal{X}} \mathrm{lcb}_{t-1}(\boldsymbol{x})) + (\max_{\boldsymbol{x} \in \mathcal{X}} \mathrm{lcb}_{t-1}(\boldsymbol{x}) - F(\hat{\boldsymbol{x}}_t)) \\
&\leq (\mathrm{ucb}_{t-1}(\boldsymbol{x}^*) - \max_{\boldsymbol{x} \in \mathcal{X}} \mathrm{lcb}_{t-1}(\boldsymbol{x})) + (\rho(\mu_{t-1}(\hat{\boldsymbol{x}}_t, \cdot)) - F(\hat{\boldsymbol{x}}_t)) \\
&\leq (\mathrm{ucb}_{t-1}(\tilde{\boldsymbol{x}}_t) - \max_{\boldsymbol{x} \in \mathcal{X}} \mathrm{lcb}_{t-1}(\boldsymbol{x})) + (\mathrm{ucb}_{t-1}(\hat{\boldsymbol{x}}_t) - \mathrm{lcb}_{t-1}(\hat{\boldsymbol{x}}_t)) \\
&\leq (\mathrm{ucb}_{t-1}(\tilde{\boldsymbol{x}}_t) - \mathrm{lcb}_{t-1}(\tilde{\boldsymbol{x}}_t)) + (\mathrm{ucb}_{t-1}(\hat{\boldsymbol{x}}_t) - \mathrm{lcb}_{t-1}(\hat{\boldsymbol{x}}_t)) \\
&\leq (\mathrm{ucb}_{t-1}(\boldsymbol{x}_t) - \mathrm{lcb}_{t-1}(\boldsymbol{x}_t)) + (\mathrm{ucb}_{t-1}(\boldsymbol{x}_t) - \mathrm{lcb}_{t-1}(\boldsymbol{x}_t)) \\
&= 2(\mathrm{ucb}_{t-1}(\boldsymbol{x}_t) - \mathrm{lcb}_{t-1}(\boldsymbol{x}_t)),
\end{aligned}
$$

where the third inequality is derived by $\mu_{t-1}(\boldsymbol{x}, \cdot) \in G_{t-1}(\boldsymbol{x})$, the definition of $\hat{\boldsymbol{x}}_t$, and $\max_{\boldsymbol{x} \in \mathcal{X}} \mathrm{lcb}_{t-1}(\boldsymbol{x}) \leq \max_{\boldsymbol{x} \in \mathcal{X}} \rho(\mu_{t-1}(\boldsymbol{x}, \cdot)) = \rho(\mu_{t-1}(\hat{\boldsymbol{x}}_t, \cdot))$. Moreover, from Assumption 4.1, there exists a function

$$
q(a) = \sum_{i=1}^{n} \zeta_i h_i \left( \sum_{j=1}^{s_i} \lambda_{ij} a^{\nu_{ij}} \right)
$$

such that $\mathrm{ucb}_{t-1}(\boldsymbol{x}_t) - \mathrm{lcb}_{t-1}(\boldsymbol{x}_t) \leq q(2\beta_t^{1/2} \sigma_{t-1}(\boldsymbol{x}_t, \boldsymbol{w}_t))$. Thus, we have

$$
F(\boldsymbol{x}^*) - F(\hat{\boldsymbol{x}}_t) \leq 2q(2\beta_t^{1/2} \sigma_{t-1}(\boldsymbol{x}_t, \boldsymbol{w}_t)). \tag{15}
$$

Let $\mathcal{F}_{t-1}(\cdot)$ be a distribution function of $F(\boldsymbol{x}^*) - F(\hat{\boldsymbol{x}}_t)$ under the given $D_{t-1}$. Then, from equation 15 we get

$$
\mathcal{F}_{t-1}(2q(2\beta_t^{1/2} \sigma_{t-1}(\boldsymbol{x}_t, \boldsymbol{w}_t))) \geq 1 - \delta.
$$

By taking the generalized inverse function for both sides, we obtain

$$
\mathcal{F}_{t-1}^{-1}(1 - \delta) \leq 2q(2\beta_t^{1/2} \sigma_{t-1}(\boldsymbol{x}_t, \boldsymbol{w}_t)).
$$

Taking the expectation with respect to $\xi_t$ for both sides, we have

$$
\mathbb{E}_{\xi_t}[\mathcal{F}_{t-1}^{-1}(1 - \delta)] \leq \mathbb{E}_{\xi_t}[2q(2\beta_t^{1/2} \sigma_{t-1}(\boldsymbol{x}_t, \boldsymbol{w}_t))].
$$

Here, since $\xi_t$ follows the chi-squared distribution with two degrees of freedom, $\delta$ follows the uniform distribution on the interval $(0, 1)$. Hence, noting that $1 - \delta$ also follows the uniform distribution on $(0, 1)$, $F(\boldsymbol{x}^*) - F(\hat{\boldsymbol{x}}_t)$ does not depend on $\delta$, from the property of the generalized inverse function, under the given $D_{t-1}$ the distribution of $\mathcal{F}_{t-1}^{-1}(1-\delta)$ is equal to that of $F(\boldsymbol{x}^*) - F(\hat{\boldsymbol{x}}_t)$. Let $\mathbb{E}_{t-1}[\cdot]$ be a conditional expectation under the given $D_{t-1}$. Then, we get

$$
\mathbb{E}_{\xi_t}[\mathcal{F}_{t-1}^{-1}(1 - \delta)] = \mathbb{E}_{t-1}[F(\boldsymbol{x}^*) - F(\hat{\boldsymbol{x}}_t)].
$$

Furthermore, since $\beta_t = 2\log(|\mathcal{X} \times \Omega|) + \xi_t$, noting that

$$
\mathbb{E}_{\xi_t}[2q(2\beta_t^{1/2} \sigma_{t-1}(\boldsymbol{x}_t, \boldsymbol{w}_t))] = \mathbb{E}_{\beta_t}[2q(2\beta_t^{1/2} \sigma_{t-1}(\boldsymbol{x}_t, \boldsymbol{w}_t))],
$$

we obtain

$$
\mathbb{E}_{t-1}[F(\boldsymbol{x}^*) - F(\hat{\boldsymbol{x}}_t)] \leq \mathbb{E}_{\beta_t}[2q(2\beta_t^{1/2} \sigma_{t-1}(\boldsymbol{x}_t, \boldsymbol{w}_t))].
$$

Thus, by taking the expectation with respect to $D_{t-1}$ for both sides, we have

$$
\mathbb{E}[r_t] = \mathbb{E}[F(\boldsymbol{x}^*) - F(\hat{\boldsymbol{x}}_t)] \leq \mathbb{E}[2q(2\beta_t^{1/2} \sigma_{t-1}(\boldsymbol{x}_t, \boldsymbol{w}_t))].
$$

Therefore, the following inequality holds:

$$\mathbb{E}[R_t] = \mathbb{E}\left[\sum_{k=1}^{t} r_k\right] \leq \mathbb{E}\left[\sum_{k=1}^{t} 2q(2\beta_k^{1/2}\sigma_{k-1}(\boldsymbol{x}_k,\boldsymbol{w}_k))\right] = 2\mathbb{E}\left[\sum_{k=1}^{t}\sum_{i=1}^{n}\zeta_i h_i\left(\sum_{j=1}^{s_i}\lambda_{ij}(2\beta_k^{1/2}\sigma_{k-1}(\boldsymbol{x}_k,\boldsymbol{w}_k))^{\nu_{ij}}\right)\right]$$

$$= 2\sum_{i=1}^{n}\zeta_i\mathbb{E}\left[\sum_{k=1}^{t}h_i\left(\sum_{j=1}^{s_i}\lambda_{ij}(2\beta_k^{1/2}\sigma_{k-1}(\boldsymbol{x}_k,\boldsymbol{w}_k))^{\nu_{ij}}\right)\right]. \quad (16)$$

In addition, since $h_i(\cdot)$ is a concave function, the following holds:

$$\sum_{k=1}^{t}h_i\left(\sum_{j=1}^{s_i}\lambda_{ij}(2\beta_k^{1/2}\sigma_{k-1}(\boldsymbol{x}_k,\boldsymbol{w}_k))^{\nu_{ij}}\right) = t\sum_{k=1}^{t}\frac{1}{t}h_i\left(\sum_{j=1}^{s_i}\lambda_{ij}(2\beta_k^{1/2}\sigma_{k-1}(\boldsymbol{x}_k,\boldsymbol{w}_k))^{\nu_{ij}}\right)$$

$$\leq th_i\left(\sum_{k=1}^{t}\frac{1}{t}\sum_{j=1}^{s_i}\lambda_{ij}(2\beta_k^{1/2}\sigma_{k-1}(\boldsymbol{x}_k,\boldsymbol{w}_k))^{\nu_{ij}}\right)$$

$$= th_i\left(\frac{1}{t}\sum_{j=1}^{s_i}2^{\nu_{ij}}\lambda_{ij}\sum_{k=1}^{t}(\beta_k^{1/2}\sigma_{k-1}(\boldsymbol{x}_k,\boldsymbol{w}_k))^{\nu_{ij}}\right).$$

Furthermore, from Jensen's inequality, we get

$$\mathbb{E}\left[\sum_{k=1}^{t}h_i\left(\sum_{j=1}^{s_i}\lambda_{ij}(2\beta_k^{1/2}\sigma_{k-1}(\boldsymbol{x}_k,\boldsymbol{w}_k))^{\nu_{ij}}\right)\right] \leq th_i\left(\frac{1}{t}\sum_{j=1}^{s_i}2^{\nu_{ij}}\lambda_{ij}\mathbb{E}\left[\sum_{k=1}^{t}(\beta_k^{1/2}\sigma_{k-1}(\boldsymbol{x}_k,\boldsymbol{w}_k))^{\nu_{ij}}\right]\right). \quad (17)$$

Here, if $\nu_{ij} \leq 1$, by letting $\eta = 2/(2-\nu_{ij})$ and $\theta = 2/\nu_{ij}$, using Hölder's inequality we obtain

$$\sum_{k=1}^{t}(\beta_k^{1/2}\sigma_{k-1}(\boldsymbol{x}_k,\boldsymbol{w}_k))^{\nu_{ij}} = \sum_{k=1}^{t}(\beta_k^{\nu_{ij}/2}\sigma_{k-1}^{\nu_{ij}}(\boldsymbol{x}_k,\boldsymbol{w}_k)) \leq \left(\sum_{k=1}^{t}\beta_k^{\nu_{ij}\eta/2}\right)^{1/\eta}\left(\sum_{k=1}^{t}\sigma_{k-1}^{\nu_{ij}\theta}(\boldsymbol{x}_k,\boldsymbol{w}_k)\right)^{1/\theta}.$$

Moreover, by using Hölder's inequality for expected values, the following inequality holds:

$$\mathbb{E}\left[\sum_{k=1}^{t}(\beta_k^{1/2}\sigma_{k-1}(\boldsymbol{x}_k,\boldsymbol{w}_k))^{\nu_{ij}}\right] \leq \left(\mathbb{E}\left[\sum_{k=1}^{t}\beta_k^{\nu_{ij}\eta/2}\right]\right)^{1/\eta}\left(\mathbb{E}\left[\sum_{k=1}^{t}\sigma_{k-1}^{\nu_{ij}\theta}(\boldsymbol{x}_k,\boldsymbol{w}_k)\right]\right)^{1/\theta}.$$

From the definition, since $\beta_1,\ldots,\beta_t$ are independent and identically distributed random variables, the following equality holds:

$$\left(\mathbb{E}\left[\sum_{k=1}^{t}\beta_k^{\nu_{ij}\eta/2}\right]\right)^{1/\eta} = \left(t\mathbb{E}\left[\beta_t^{\nu_{ij}\eta/2}\right]\right)^{1/\eta} = \left(t\mathbb{E}\left[\beta_t^{\nu_{ij}/(2-\nu_{ij})}\right]\right)^{1-\nu_{ij}/2} = \left(t\mathbb{E}\left[\beta_t^{\nu_{ij}/(2-\nu'_{ij})}\right]\right)^{1-\nu'_{ij}/2}. \quad (18)$$

In addition, from the proof of Lemma 5.4 in Srinivas et al. (2010), the following inequality holds:

$$\sum_{k=1}^{t}\sigma_{k-1}^2(\boldsymbol{x}_k,\boldsymbol{w}_k) \leq C_1\gamma_t.$$

Hence, we have

$$\left(\mathbb{E}\left[\sum_{k=1}^{t}\sigma_{k-1}^{\nu_{ij}\theta}(\boldsymbol{x}_k,\boldsymbol{w}_k)\right]\right)^{1/\theta} = \left(\mathbb{E}\left[\sum_{k=1}^{t}\sigma_{k-1}^2(\boldsymbol{x}_k,\boldsymbol{w}_k)\right]\right)^{\nu_{ij}/2} \leq (C_1\gamma_t)^{\nu_{ij}/2} = (C_1\gamma_t)^{\nu'_{ij}/2}. \quad (19)$$

Thus, we get

$$\mathbb{E}\left[\sum_{k=1}^{t}(\beta_k^{1/2}\sigma_{k-1}(\boldsymbol{x}_k,\boldsymbol{w}_k))^{\nu_{ij}}\right] \leq \left(t\mathbb{E}\left[\beta_t^{\nu_{ij}/(2-\nu'_{ij})}\right]\right)^{1-\nu'_{ij}/2}(C_1\gamma_t)^{\nu'_{ij}/2} = \left(tC_{2,\nu_{ij}}\right)^{1-\nu'_{ij}/2}(C_1\gamma_t)^{\nu'_{ij}/2}.$$

Similarly, if $\nu_{ij} > 1$, by letting $\eta = \theta = 2$, using the same argument we obtain

$$\mathbb{E}\left[\sum_{k=1}^{t}(\beta_k^{1/2}\sigma_{k-1}(\boldsymbol{x}_k,\boldsymbol{w}_k))^{\nu_{ij}}\right] \leq \left(\mathbb{E}\left[\sum_{k=1}^{t}\beta_k^{\nu_{ij}}\right]\right)^{1/2}\left(\mathbb{E}\left[\sum_{k=1}^{t}\sigma_{k-1}^{2\nu_{ij}}(\boldsymbol{x}_k,\boldsymbol{w}_k)\right]\right)^{1/2}$$

$$= \left(\mathbb{E}\left[\sum_{k=1}^{t}\beta_k^{\nu_{ij}/(2-\nu'_{ij})}\right]\right)^{1-\nu'_{ij}/2}\left(\mathbb{E}\left[\sum_{k=1}^{t}\sigma_{k-1}^{2\nu_{ij}}(\boldsymbol{x}_k,\boldsymbol{w}_k)\right]\right)^{\nu'_{ij}/2}.$$

Therefore, since $\sigma_{k-1}(\boldsymbol{x},\boldsymbol{w}) \leq 1$, noting that $\sigma_{k-1}^{2\nu_{ij}}(\boldsymbol{x}_k,\boldsymbol{w}_k) \leq \sigma_{k-1}^2(\boldsymbol{x}_k,\boldsymbol{w}_k)$ we get

$$\mathbb{E}\left[\sum_{k=1}^{t}(\beta_k^{1/2}\sigma_{k-1}(\boldsymbol{x}_k,\boldsymbol{w}_k))^{\nu_{ij}}\right] \leq \left(\mathbb{E}\left[\sum_{k=1}^{t}\beta_k^{\nu_{ij}/(2-\nu'_{ij})}\right]\right)^{1-\nu'_{ij}/2}\left(\mathbb{E}\left[\sum_{k=1}^{t}\sigma_{k-1}^{2\nu_{ij}}(\boldsymbol{x}_k,\boldsymbol{w}_k)\right]\right)^{\nu'_{ij}/2}$$

$$\leq \left(\mathbb{E}\left[\sum_{k=1}^{t}\beta_k^{\nu_{ij}/(2-\nu'_{ij})}\right]\right)^{1-\nu'_{ij}/2}\left(\mathbb{E}\left[\sum_{k=1}^{t}\sigma_{k-1}^2(\boldsymbol{x}_k,\boldsymbol{w}_k)\right]\right)^{\nu'_{ij}/2} \tag{20}$$

$$\leq \left(tC_{2,\nu_{ij}}\right)^{1-\nu'_{ij}/2}(C_1\gamma_t)^{\nu'_{ij}/2}.$$

Hence, for any $\nu_{ij} > 0$, the following inequality holds:

$$\mathbb{E}\left[\sum_{k=1}^{t}(\beta_k^{1/2}\sigma_{k-1}(\boldsymbol{x}_k,\boldsymbol{w}_k))^{\nu_{ij}}\right] \leq \left(tC_{2,\nu_{ij}}\right)^{1-\nu'_{ij}/2}(C_1\gamma_t)^{\nu'_{ij}/2}. \tag{21}$$

Thus, noting that $h_i(\cdot)$ is a monotonic non-decreasing function, by substituting equation 21 into equation 17 we obtain

$$\mathbb{E}\left[\sum_{k=1}^{t}h_i\left(\sum_{j=1}^{s_i}\lambda_{ij}(2\beta_k^{1/2}\sigma_{k-1}(\boldsymbol{x}_k,\boldsymbol{w}_k))^{\nu_{ij}}\right)\right] \leq th_i\left(\frac{1}{t}\sum_{j=1}^{s_i}2^{\nu_{ij}}\lambda_{ij}\left(tC_{2,\nu_{ij}}\right)^{1-\nu'_{ij}/2}(C_1\gamma_t)^{\nu'_{ij}/2}\right). \tag{22}$$

Finally, by substituting equation 22 into equation 16, we get

$$\mathbb{E}[R_t] \leq 2t\sum_{i=1}^{n}\zeta_i h_i\left(\frac{1}{t}\sum_{j=1}^{s_i}2^{\nu_{ij}}\lambda_{ij}\left(tC_{2,\nu_{ij}}\right)^{1-\nu'_{ij}/2}(C_1\gamma_t)^{\nu'_{ij}/2}\right).$$

$\square$

## C.3 Proof of Theorem 4.2

*Proof.* For the expectation $\mathbb{E}[r_{\hat{t}}]$, the following equality holds:

$$\mathbb{E}[r_{\hat{t}}] = \mathbb{E}_{D_{t-1}}[\mathbb{E}_{t-1}[r_{\hat{t}}]] = \mathbb{E}_{D_{t-1}}[\mathbb{E}_{t-1}[F(\boldsymbol{x}^*) - F(\hat{\boldsymbol{x}}_{\hat{t}})]],$$

where $\mathbb{E}_{D_{t-1}}[\cdot]$ is the expectation with respect to $D_{t-1}$. From the definition of $\hat{t}$, the following inequality holds for any $1 \leq i \leq t$:

$$\mathbb{E}_{t-1}[F(\boldsymbol{x}^*) - F(\hat{\boldsymbol{x}}_{\hat{t}})] \leq \mathbb{E}_{t-1}[F(\boldsymbol{x}^*) - F(\hat{\boldsymbol{x}}_i)]. \tag{23}$$

This implies that

$$\mathbb{E}_{t-1}[F(\boldsymbol{x}^*) - F(\hat{\boldsymbol{x}}_{\hat{t}})] \leq \frac{1}{t}\sum_{i=1}^{t}\mathbb{E}_{t-1}[F(\boldsymbol{x}^*) - F(\hat{\boldsymbol{x}}_i)].$$

Therefore, using this we get

$$
\begin{aligned}
\mathbb{E}[r_{\hat{t}}] = \mathbb{E}_{D_{t-1}}[\mathbb{E}_{t-1}[F(\boldsymbol{x}^*) - F(\hat{\boldsymbol{x}}_{\hat{t}})]] &\leq \mathbb{E}_{D_{t-1}}\left[\frac{1}{t}\sum_{i=1}^t \mathbb{E}_{t-1}[F(\boldsymbol{x}^*) - F(\hat{\boldsymbol{x}}_i)]\right] \\
&= \frac{1}{t}\sum_{i=1}^t \mathbb{E}_{D_{t-1}}[\mathbb{E}_{t-1}[F(\boldsymbol{x}^*) - F(\hat{\boldsymbol{x}}_i)]] \\
&= \frac{1}{t}\sum_{i=1}^t \mathbb{E}[F(\boldsymbol{x}^*) - F(\hat{\boldsymbol{x}}_i)] = \frac{1}{t}\mathbb{E}\left[\sum_{i=1}^t F(\boldsymbol{x}^*) - F(\hat{\boldsymbol{x}}_i)\right] = \frac{\mathbb{E}[R_t]}{t}.
\end{aligned}
$$

$\square$

### C.4 Proof of Theorem 4.3

*Proof.* We show $\hat{t} = t$. Let $\Omega = \{\boldsymbol{w}^{(1)}, \ldots, \boldsymbol{w}^{(L)}\}$ and $\mathbb{P}(\boldsymbol{w} = \boldsymbol{w}^{(j)}) = p_j$, where $j \in \{1, \ldots, L\}$, and the probability is taken with respect to $\boldsymbol{w}$. Then, from the definition of the expectation measure, $F(\boldsymbol{x})$ can be expressed as follows:

$$
F(\boldsymbol{x}) = \sum_{j=1}^L p_j f(\boldsymbol{x}, \boldsymbol{w}^{(j)}).
$$

Therefore, the following equality holds:

$$
\mathbb{E}_{t-1}[F(\boldsymbol{x})] = \sum_{j=1}^L p_j \mathbb{E}_{t-1}[f(\boldsymbol{x}, \boldsymbol{w}^{(j)})] = \sum_{j=1}^L p_j \mu_{t-1}(\boldsymbol{x}, \boldsymbol{w}^{(j)}) = \rho(\mu_{t-1}(\boldsymbol{x}, \cdot)).
$$

Here, from the definition of $\hat{\boldsymbol{x}}_t$, the following inequality holds for any $\boldsymbol{x} \in \mathcal{X}$:

$$
\rho(\mu_{t-1}(\boldsymbol{x}, \cdot)) \leq \rho(\mu_{t-1}(\hat{\boldsymbol{x}}_t, \cdot)).
$$

Hence, for any $i \in \{1, \ldots, t\}$, the following holds:

$$
\mathbb{E}_{t-1}[F(\hat{\boldsymbol{x}}_i)] = \rho(\mu_{t-1}(\hat{\boldsymbol{x}}_i, \cdot)) \leq \rho(\mu_{t-1}(\hat{\boldsymbol{x}}_t, \cdot)) = \mathbb{E}_{t-1}[F(\hat{\boldsymbol{x}}_t)].
$$

This implies that $\hat{t} = t$. $\square$

### C.5 Equality between $q^{(m)}(a)$ in Table 4 in Inatsu et al. (2024a) and $q_1(a)$, $q_2(a)$, and $q_3(a)$

*Proof.* For any $\boldsymbol{x}, \boldsymbol{x}', f(\boldsymbol{x}, \boldsymbol{w})$ and $f'(\boldsymbol{x}', \boldsymbol{w})$, we derive $q(a)$ satisfying

$$
|\rho(f(\boldsymbol{x}, \cdot)) - \rho(f'(\boldsymbol{x}', \cdot))| \leq q\left(\sup_{\boldsymbol{w} \in \Omega} |f(\boldsymbol{x}, \boldsymbol{w}) - f'(\boldsymbol{x}', \boldsymbol{w})|\right).
$$

Since $\boldsymbol{x}, \boldsymbol{x}', f(\boldsymbol{x}, \boldsymbol{w})$ and $f'(\boldsymbol{x}', \boldsymbol{w})$ are arbitrary, we can assume that $|\rho(f(\boldsymbol{x}, \cdot)) - \rho(f'(\boldsymbol{x}', \cdot))| = \rho(f(\boldsymbol{x}, \cdot)) - \rho(f'(\boldsymbol{x}', \cdot))$ without loss of generality. For the distributionally robust, monotone Lipschitz map, and weighted sum measures, we can use $q^{(m)}(a)$ in Table 4 in Inatsu et al. (2024a) by introducing additional assumptions about monotonicity, concavity, and taking zero at point zero.

**Expectation** Let $\mathbb{E}_{\boldsymbol{w}}[\cdot]$ be an expectation with respect to $\boldsymbol{w}$. Then, the following holds:

$$
\begin{aligned}
\rho(f(\boldsymbol{x}, \cdot)) - \rho(f'(\boldsymbol{x}', \cdot)) &= \mathbb{E}_{\boldsymbol{w}}[f(\boldsymbol{x}, \boldsymbol{w})] - \mathbb{E}_{\boldsymbol{w}}[f'(\boldsymbol{x}', \boldsymbol{w})] \\
&= \mathbb{E}_{\boldsymbol{w}}[f(\boldsymbol{x}, \boldsymbol{w}) - f'(\boldsymbol{x}', \boldsymbol{w})] \\
&\leq \mathbb{E}_{\boldsymbol{w}}[|f(\boldsymbol{x}, \boldsymbol{w}) - f'(\boldsymbol{x}', \boldsymbol{w})|] \leq \sup_{\boldsymbol{w} \in \Omega} |f(\boldsymbol{x}, \boldsymbol{w}) - f'(\boldsymbol{x}', \boldsymbol{w})|.
\end{aligned}
$$

This implies that $q(a) = a$.

**Worst-Case**  For any $\epsilon > 0$, from the definition of infimum, there exists $\boldsymbol{w}^*$ such that

$$f'(\boldsymbol{x}', \boldsymbol{w}^*) \leq \inf_{\boldsymbol{w} \in \Omega} f'(\boldsymbol{x}', \boldsymbol{w}) + \epsilon.$$

Thus, we have

$$\inf_{\boldsymbol{w} \in \Omega} f(\boldsymbol{x}, \boldsymbol{w}) - \inf_{\boldsymbol{w} \in \Omega} f'(\boldsymbol{x}', \boldsymbol{w}) \leq \inf_{\boldsymbol{w} \in \Omega} f(\boldsymbol{x}, \boldsymbol{w}) - f'(\boldsymbol{x}', \boldsymbol{w}^*) + \epsilon \leq f(\boldsymbol{x}, \boldsymbol{w}^*) - f'(\boldsymbol{x}', \boldsymbol{w}^*) + \epsilon$$
$$\leq \sup_{\boldsymbol{w} \in \Omega} |f(\boldsymbol{x}, \boldsymbol{w}) - f'(\boldsymbol{x}', \boldsymbol{w})| + \epsilon.$$

Since $\epsilon > 0$ is arbitrary, we get

$$\rho(f(\boldsymbol{x}, \cdot)) - \rho(f'(\boldsymbol{x}', \cdot)) = \inf_{\boldsymbol{w} \in \Omega} f(\boldsymbol{x}, \boldsymbol{w}) - \inf_{\boldsymbol{w} \in \Omega} f'(\boldsymbol{x}', \boldsymbol{w}) \leq \sup_{\boldsymbol{w} \in \Omega} |f(\boldsymbol{x}, \boldsymbol{w}) - f'(\boldsymbol{x}', \boldsymbol{w})|.$$

This implies that $q(a) = a$.

**Best-Case**  For any $\epsilon > 0$, from the definition of supremum, there exists $\boldsymbol{w}^*$ such that

$$\sup_{\boldsymbol{w} \in \Omega} f(\boldsymbol{x}, \boldsymbol{w}^*) \leq f(\boldsymbol{x}, \boldsymbol{w}^*) + \epsilon.$$

Hence, we obtain

$$\sup_{\boldsymbol{w} \in \Omega} f(\boldsymbol{x}, \boldsymbol{w}) - \sup_{\boldsymbol{w} \in \Omega} f'(\boldsymbol{x}', \boldsymbol{w}) \leq f(\boldsymbol{x}, \boldsymbol{w}^*) + \epsilon - \sup_{\boldsymbol{w} \in \Omega} f'(\boldsymbol{x}', \boldsymbol{w}) \leq f(\boldsymbol{x}, \boldsymbol{w}^*) - f'(\boldsymbol{x}', \boldsymbol{w}^*) + \epsilon$$
$$\leq \sup_{\boldsymbol{w} \in \Omega} |f(\boldsymbol{x}, \boldsymbol{w}) - f'(\boldsymbol{x}', \boldsymbol{w})| + \epsilon.$$

Since $\epsilon > 0$ is arbitrary, we have

$$\rho(f(\boldsymbol{x}, \cdot)) - \rho(f'(\boldsymbol{x}', \cdot)) = \sup_{\boldsymbol{w} \in \Omega} f(\boldsymbol{x}, \boldsymbol{w}) - \sup_{\boldsymbol{w} \in \Omega} f'(\boldsymbol{x}', \boldsymbol{w}) \leq \sup_{\boldsymbol{w} \in \Omega} |f(\boldsymbol{x}, \boldsymbol{w}) - f'(\boldsymbol{x}', \boldsymbol{w})|.$$

This implies that $q(a) = a$.

**$\alpha$-Value-at-Risk**  Let $\alpha \in (0, 1)$ and $k = \sup_{\boldsymbol{w} \in \Omega} |f(\boldsymbol{x}, \boldsymbol{w}) - f'(\boldsymbol{x}', \boldsymbol{w})|$. Then, we have $f(\boldsymbol{x}, \boldsymbol{w}) \leq f'(\boldsymbol{x}', \boldsymbol{w}) + k$. Here, for any $b \in \mathbb{R}$, the inequality $\mathbb{P}_{\boldsymbol{w}}(f'(\boldsymbol{x}', \boldsymbol{w}) + k \leq b) \leq \mathbb{P}_{\boldsymbol{w}}(f(\boldsymbol{x}, \boldsymbol{w}) \leq b)$ holds, where $\mathbb{P}_{\boldsymbol{w}}(\cdot)$ is the probability with respect to $\boldsymbol{w}$. Here, if $\alpha \leq \mathbb{P}_{\boldsymbol{w}}(f'(\boldsymbol{x}', \boldsymbol{w}) + k \leq b)$ holds, then $\alpha \leq \mathbb{P}_{\boldsymbol{w}}(f(\boldsymbol{x}, \boldsymbol{w}) \leq b)$ holds. This implies that

$$v_f(\boldsymbol{x}; \alpha) \equiv \rho(f(\boldsymbol{x}, \cdot)) = \inf\{b \in \mathbb{R} \mid \alpha \leq \mathbb{P}_{\boldsymbol{w}}(f(\boldsymbol{x}, \boldsymbol{w}) \leq b)\} \leq \inf\{b \in \mathbb{R} \mid \alpha \leq \mathbb{P}_{\boldsymbol{w}}(f'(\boldsymbol{x}', \boldsymbol{w}) + k \leq b)\}$$
$$= \inf\{b \in \mathbb{R} \mid \alpha \leq \mathbb{P}_{\boldsymbol{w}}(f'(\boldsymbol{x}', \boldsymbol{w}) \leq b)\} + k$$
$$= \rho(f'(\boldsymbol{x}', \cdot)) + k.$$

Therefore, we get

$$\rho(f(\boldsymbol{x}, \cdot)) - \rho(f'(\boldsymbol{x}', \cdot)) \leq k = \sup_{\boldsymbol{w} \in \Omega} |f(\boldsymbol{x}, \boldsymbol{w}) - f'(\boldsymbol{x}', \boldsymbol{w})|.$$

This implies that $q(a) = a$.

**$\alpha$-Conditional Value-at-Risk**  Let $\alpha \in (0, 1)$. From Nguyen et al. (2021a), the $\alpha$-conditional value-at-risk can be rewritten as follows:

$$\rho(f(\boldsymbol{x}, \cdot)) = \frac{1}{\alpha} \int_0^\alpha v_f(\boldsymbol{x}; \alpha') \mathrm{d}\alpha'.$$

Thus, using the result of the $\alpha$-value-at-risk, we have

$$\rho(f(\boldsymbol{x}, \cdot)) - \rho(f'(\boldsymbol{x}', \cdot)) = \frac{1}{\alpha} \int_0^\alpha (v_f(\boldsymbol{x}; \alpha') - v_{f'}(\boldsymbol{x}'; \alpha')) \mathrm{d}\alpha' \leq \frac{1}{\alpha} \int_0^\alpha \sup_{\boldsymbol{w} \in \Omega} |f(\boldsymbol{x}, \boldsymbol{w}) - f'(\boldsymbol{x}', \boldsymbol{w})| \mathrm{d}\alpha'$$
$$= \sup_{\boldsymbol{w} \in \Omega} |f(\boldsymbol{x}, \boldsymbol{w}) - f'(\boldsymbol{x}', \boldsymbol{w})|.$$

This implies that $q(a) = a$.

**Mean Absolute Deviation**  From the triangle inequality $|a| - |b| \leq |a - b|$, the following holds:

$$
\begin{aligned}
\rho(f(\boldsymbol{x}, \cdot)) - \rho(f'(\boldsymbol{x}', \cdot)) &= \mathbb{E}_{\boldsymbol{w}}[|f(\boldsymbol{x}, \boldsymbol{w}) - \mathbb{E}_{\boldsymbol{w}}[f(\boldsymbol{x}, \boldsymbol{w})]|] - \mathbb{E}_{\boldsymbol{w}}[|f'(\boldsymbol{x}', \boldsymbol{w}) - \mathbb{E}_{\boldsymbol{w}}[f'(\boldsymbol{x}', \boldsymbol{w})]|] \\
&= \mathbb{E}_{\boldsymbol{w}}[|f(\boldsymbol{x}, \boldsymbol{w}) - \mathbb{E}_{\boldsymbol{w}}[f(\boldsymbol{x}, \boldsymbol{w})]| - |f'(\boldsymbol{x}', \boldsymbol{w}) - \mathbb{E}_{\boldsymbol{w}}[f'(\boldsymbol{x}', \boldsymbol{w})]|] \\
&\leq \mathbb{E}_{\boldsymbol{w}}[|(f(\boldsymbol{x}, \boldsymbol{w}) - f'(\boldsymbol{x}', \boldsymbol{w})) - (\mathbb{E}_{\boldsymbol{w}}[f(\boldsymbol{x}, \boldsymbol{w}) - f'(\boldsymbol{x}', \boldsymbol{w})])|] \\
&\leq \mathbb{E}_{\boldsymbol{w}}[|f(\boldsymbol{x}, \boldsymbol{w}) - f'(\boldsymbol{x}', \boldsymbol{w})|] + \mathbb{E}_{\boldsymbol{w}}[|\mathbb{E}_{\boldsymbol{w}}[f(\boldsymbol{x}, \boldsymbol{w}) - f'(\boldsymbol{x}', \boldsymbol{w})]|] \\
&\leq \sup_{\boldsymbol{w} \in \Omega} |f(\boldsymbol{x}, \boldsymbol{w}) - f'(\boldsymbol{x}', \boldsymbol{w})| + \sup_{\boldsymbol{w} \in \Omega} |f(\boldsymbol{x}, \boldsymbol{w}) - f'(\boldsymbol{x}', \boldsymbol{w})| \\
&= 2 \sup_{\boldsymbol{w} \in \Omega} |f(\boldsymbol{x}, \boldsymbol{w}) - f'(\boldsymbol{x}', \boldsymbol{w})|.
\end{aligned}
$$

This implies that $q(a) = 2a$.

**Distributionally Robust**  Let $P$ be a candidate distribution of $\boldsymbol{w}$, $\mathcal{A}$ be a family of candidate distributions, and $\rho_P(\cdot)$ be a robustness measure with respect to $P$. The distributionally robust measure is defined as $\rho(f(\boldsymbol{x}, \cdot)) = \inf_{P \in \mathcal{A}} \rho_P(f(\boldsymbol{x}, \cdot))$. Let $q_{\mathcal{A}}(\cdot)$ be a function satisfying $|\rho_P(f(\boldsymbol{x}, \cdot)) - \rho_P(f'(\boldsymbol{x}', \cdot))| \leq q_{\mathcal{A}}(\sup_{\boldsymbol{w} \in \Omega} |f(\boldsymbol{x}, \boldsymbol{w}) - f'(\boldsymbol{x}', \boldsymbol{w})|)$ for any $P$. Additionally, assume that $q_{\mathcal{A}}(\cdot)$ is a non-decreasing concave function satisfying $q_{\mathcal{A}}(0) = 0$. For any $\epsilon > 0$, from the definition of infimum, there exists a distribution $P' \in \mathcal{A}$ such that

$$
\rho(f'(\boldsymbol{x}', \cdot)) = \inf_{P \in \mathcal{A}} \rho_P(f'(\boldsymbol{x}', \cdot)) \geq \rho_{P'}(f'(\boldsymbol{x}', \cdot)) - \epsilon.
$$

Therefore, the following inequality holds:

$$
\begin{aligned}
\rho(f(\boldsymbol{x}, \cdot)) - \rho(f'(\boldsymbol{x}', \cdot)) &\leq \inf_{P \in \mathcal{A}} \rho_P(f(\boldsymbol{x}, \cdot)) - \rho_{P'}(f'(\boldsymbol{x}', \cdot)) + \epsilon \\
&\leq \rho_{P'}(f(\boldsymbol{x}, \cdot)) - \rho_{P'}(f'(\boldsymbol{x}', \cdot)) + \epsilon \\
&\leq q_{\mathcal{A}} \left( \sup_{\boldsymbol{w} \in \Omega} |f(\boldsymbol{x}, \boldsymbol{w}) - f'(\boldsymbol{x}', \boldsymbol{w})| \right) + \epsilon.
\end{aligned}
$$

Since $\epsilon > 0$ is arbitrary, we get

$$
\rho(f(\boldsymbol{x}, \cdot)) - \rho(f'(\boldsymbol{x}', \cdot)) \leq q_{\mathcal{A}} \left( \sup_{\boldsymbol{w} \in \Omega} |f(\boldsymbol{x}, \boldsymbol{w}) - f'(\boldsymbol{x}', \boldsymbol{w})| \right).
$$

This implies that $q(a) = q_{\mathcal{A}}(a)$.

**Monotone Lipschitz Map**  Let $\mathcal{M}$ be a $K$-Lipschitz function, and $\tilde{\rho}(\cdot)$ be a robustness measure. The target robustness measure is defined as $\rho(\cdot) = (\mathcal{M} \circ \tilde{\rho})(\cdot)$. Let $\tilde{q}(a)$ be a function satisfying

$$
|\tilde{\rho}(f(\boldsymbol{x}, \cdot)) - \tilde{\rho}(f'(\boldsymbol{x}', \cdot))| \leq \tilde{q} \left( \sup_{\boldsymbol{w} \in \Omega} |f(\boldsymbol{x}, \boldsymbol{w}) - f'(\boldsymbol{x}', \boldsymbol{w})| \right).
$$

Additionally, assume that $\tilde{q}(a)$ is a non-decreasing concave function satisfying $\tilde{q}(0) = 0$. Then, for the robustness measure $\rho(\cdot)$, the following holds:

$$
\begin{aligned}
\rho(f(\boldsymbol{x}, \cdot)) - \rho(f'(\boldsymbol{x}', \cdot)) = \mathcal{M}(\tilde{\rho}(f(\boldsymbol{x}, \cdot))) - \mathcal{M}(\tilde{\rho}(f'(\boldsymbol{x}', \cdot))) &\leq K|\tilde{\rho}(f(\boldsymbol{x}, \cdot)) - \tilde{\rho}(f'(\boldsymbol{x}', \cdot))| \\
&\leq K\tilde{q} \left( \sup_{\boldsymbol{w} \in \Omega} |f(\boldsymbol{x}, \boldsymbol{w}) - f'(\boldsymbol{x}', \boldsymbol{w})| \right).
\end{aligned}
$$

This implies that $q(a) = K\tilde{q}(a)$.

**Weighted Sum**  Let $\alpha_1, \alpha_2 \geq 0$, and let $\rho_i(\cdot)$ be a robustness measure. The target robustness measure is defined as $\rho(\cdot) = \alpha_1 \rho_1(\cdot) + \alpha_2 \rho_2(\cdot)$. Let $q_i(a)$ be a function satisfying

$$
|\rho_i(f(\boldsymbol{x}, \cdot)) - \rho_i(f'(\boldsymbol{x}', \cdot))| \leq q_i \left( \sup_{\boldsymbol{w} \in \Omega} |f(\boldsymbol{x}, \boldsymbol{w}) - f'(\boldsymbol{x}', \boldsymbol{w})| \right).
$$

Additionally, assume that $q_i(a)$ is a non-decreasing concave function satisfying $q_i(0) = 0$. Then, for the robustness measure $\rho(\cdot)$, the following holds:

$$
\begin{aligned}
\rho(f(\boldsymbol{x}, \cdot)) - \rho(f'(\boldsymbol{x}', \cdot)) &= \alpha_1 \rho_1(f(\boldsymbol{x}, \cdot)) + \alpha_2 \rho_2(f(\boldsymbol{x}, \cdot)) - (\alpha_1 \rho_1(f'(\boldsymbol{x}', \cdot)) + \alpha_2 \rho_2(f'(\boldsymbol{x}', \cdot))) \\
&= \alpha_1 (\rho_1(f(\boldsymbol{x}, \cdot)) - \rho_1(f'(\boldsymbol{x}', \cdot))) + \alpha_2 (\rho_2(f(\boldsymbol{x}, \cdot)) - \rho_2(f'(\boldsymbol{x}', \cdot))) \\
&\leq \alpha_1 q_1 \left( \sup_{\boldsymbol{w} \in \Omega} |f(\boldsymbol{x}, \boldsymbol{w}) - f'(\boldsymbol{x}', \boldsymbol{w})| \right) + \alpha_2 q_2 \left( \sup_{\boldsymbol{w} \in \Omega} |f(\boldsymbol{x}, \boldsymbol{w}) - f'(\boldsymbol{x}', \boldsymbol{w})| \right).
\end{aligned}
$$

This implies that $q(a) = \alpha_1 q_1(a) + \alpha_2 q_2(a)$.

$\square$

### C.6 Proof of Theorem A.1

*Proof.* For $t \geq 1$, let $\tau_t = \lceil b_1 d_1 r t^2 (\sqrt{\log(a_1 d_1 |\Omega|)} + \sqrt{\pi}/2) \rceil$. Suppose that $\mathcal{X}_t$ is a set of discretization for $\mathcal{X}$ with each coordinate equally divided into $\tau_t$. Note that $|\mathcal{X}_t| = \tau_t^{d_1}$. For each $\boldsymbol{x} \in \mathcal{X}$, let $[\boldsymbol{x}]_t$ be the element of $\mathcal{X}_t$ closest to $\boldsymbol{x}$. Then, $r_t$ can be expressed as follows:

$$
r_t = F(\boldsymbol{x}^*) - F(\hat{\boldsymbol{x}}_t) = F(\boldsymbol{x}^*) - F([\boldsymbol{x}^*]_t) + F([\boldsymbol{x}^*]_t) - F(\hat{\boldsymbol{x}}_t).
$$

Therefore, the following equality holds:

$$
\mathbb{E}[R_t] = \mathbb{E}\left[ \sum_{k=1}^t (F(\boldsymbol{x}^*) - F([\boldsymbol{x}^*]_k)) \right] + \mathbb{E}\left[ \sum_{k=1}^t (F([\boldsymbol{x}^*]_k) - F(\hat{\boldsymbol{x}}_k)) \right]. \tag{24}
$$

Let $\xi_t$ be a realization from the chi-squared distribution with two degrees of freedom, and let $\delta = \frac{1}{\exp(\xi_t/2)}$. Then, from the proof of Lemma 5.1 in Srinivas et al. (2010), with probability at least $1 - \delta$, the following inequality holds for any $(\boldsymbol{x}, \boldsymbol{w}) \in (\mathcal{X}_t \cup \{\hat{\boldsymbol{x}}_t\}) \times \Omega$:

$$
l_{t-1,\delta}(\boldsymbol{x}, \boldsymbol{w}) \equiv \mu_{t-1}(\boldsymbol{x}, \boldsymbol{w}) - \beta_\delta^{1/2} \sigma_{t-1}(\boldsymbol{x}, \boldsymbol{w}) \leq f(\boldsymbol{x}, \boldsymbol{w}) \leq \mu_{t-1}(\boldsymbol{x}, \boldsymbol{w}) + \beta_\delta^{1/2} \sigma_{t-1}(\boldsymbol{x}, \boldsymbol{w}) \equiv u_{t-1,\delta}(\boldsymbol{x}, \boldsymbol{w}),
$$

where $\beta_\delta = 2 \log(|(\mathcal{X}_t \cup \{\hat{\boldsymbol{x}}_t\}) \times \Omega|/\delta)$. From the definition of $\delta$, noting that $|(\mathcal{X}_t \cup \{\hat{\boldsymbol{x}}_t\}) \times \Omega| = (1 + \tau_t^{d_1})|\Omega| = \kappa_t^{(1)}$ and $\beta_\delta = 2 \log(\kappa_t^{(1)}) + \xi_t = \beta_t$, the following holds with probability at least $1 - \delta$:

$$
l_{t-1}(\boldsymbol{x}, \boldsymbol{w}) \leq f(\boldsymbol{x}, \boldsymbol{w}) \leq u_{t-1}(\boldsymbol{x}, \boldsymbol{w}).
$$

Hence, for the function $f(\boldsymbol{x}, \boldsymbol{w})$ with respect to $\boldsymbol{w}$, each element $\boldsymbol{x} \in \mathcal{X}_t \cup \{\hat{\boldsymbol{x}}_t\}$ satisfies $f(\boldsymbol{x}, \cdot) \in G_{t-1}(\boldsymbol{x})$. Moreover, from the theorem's assumption, since $\text{ucb}_{t-1}(\boldsymbol{x})$ and $\text{lcb}_{t-1}(\boldsymbol{x})$ satisfy equation 2, the following inequality holds:

$$
\text{lcb}_{t-1}(\boldsymbol{x}) \leq F(\boldsymbol{x}) \leq \text{ucb}_{t-1}(\boldsymbol{x}).
$$

Therefore, noting that $[\boldsymbol{x}^*]_t, \hat{\boldsymbol{x}}_t \in \mathcal{X}_t \cup \{\hat{\boldsymbol{x}}_t\}$, $F([\boldsymbol{x}^*]_t) - F(\hat{\boldsymbol{x}}_t)$ can be evaluated as follows:

$$
\begin{aligned}
F([\boldsymbol{x}^*]_t) - F(\hat{\boldsymbol{x}}_t) &= (F([\boldsymbol{x}^*]_t) - \max_{\boldsymbol{x} \in \mathcal{X}} \text{lcb}_{t-1}(\boldsymbol{x})) + (\max_{\boldsymbol{x} \in \mathcal{X}} \text{lcb}_{t-1}(\boldsymbol{x}) - F(\hat{\boldsymbol{x}}_t)) \\
&\leq (\text{ucb}_{t-1}([\boldsymbol{x}^*]_t) - \max_{\boldsymbol{x} \in \mathcal{X}} \text{lcb}_{t-1}(\boldsymbol{x})) + (\max_{\boldsymbol{x} \in \mathcal{X}} \text{lcb}_{t-1}(\boldsymbol{x}) - F(\hat{\boldsymbol{x}}_t)) \\
&\leq (\text{ucb}_{t-1}([\boldsymbol{x}^*]_t) - \max_{\boldsymbol{x} \in \mathcal{X}} \text{lcb}_{t-1}(\boldsymbol{x})) + (\rho(\mu_{t-1}(\hat{\boldsymbol{x}}_t, \cdot)) - F(\hat{\boldsymbol{x}}_t)) \\
&\leq (\text{ucb}_{t-1}(\tilde{\boldsymbol{x}}_t) - \max_{\boldsymbol{x} \in \mathcal{X}} \text{lcb}_{t-1}(\boldsymbol{x})) + (\text{ucb}_{t-1}(\hat{\boldsymbol{x}}_t) - \text{lcb}_{t-1}(\hat{\boldsymbol{x}}_t)) \\
&\leq (\text{ucb}_{t-1}(\tilde{\boldsymbol{x}}_t) - \text{lcb}_{t-1}(\tilde{\boldsymbol{x}}_t)) + (\text{ucb}_{t-1}(\hat{\boldsymbol{x}}_t) - \text{lcb}_{t-1}(\hat{\boldsymbol{x}}_t)) \\
&\leq (\text{ucb}_{t-1}(\boldsymbol{x}_t) - \text{lcb}_{t-1}(\boldsymbol{x}_t)) + (\text{ucb}_{t-1}(\boldsymbol{x}_t) - \text{lcb}_{t-1}(\boldsymbol{x}_t)) \\
&= 2(\text{ucb}_{t-1}(\boldsymbol{x}_t) - \text{lcb}_{t-1}(\boldsymbol{x}_t)),
\end{aligned}
$$

where the third inequality is derived by $\mu_{t-1}(\boldsymbol{x}, \cdot) \in G_{t-1}(\boldsymbol{x})$, the definition of $\hat{\boldsymbol{x}}_t$ and $\max_{\boldsymbol{x} \in \mathcal{X}} \mathrm{lcb}_{t-1}(\boldsymbol{x}) \leq \max_{\boldsymbol{x} \in \mathcal{X}} \rho(\mu_{t-1}(\boldsymbol{x}, \cdot)) = \rho(\mu_{t-1}(\hat{\boldsymbol{x}}_t, \cdot))$. Furthermore, from Assumption 4.1, there exists a function

$$q(a) = \sum_{i=1}^{n} \zeta_i h_i \left( \sum_{j=1}^{s_i} \lambda_{ij} a^{\nu_{ij}} \right)$$

such that $\mathrm{ucb}_{t-1}(\boldsymbol{x}_t) - \mathrm{lcb}_{t-1}(\boldsymbol{x}_t) \leq q(2\beta_t^{1/2}\sigma_{t-1}(\boldsymbol{x}_t, \boldsymbol{w}_t))$. This implies that

$$F([\boldsymbol{x}^*]_t) - F(\hat{\boldsymbol{x}}_t) \leq 2q(2\beta_t^{1/2}\sigma_{t-1}(\boldsymbol{x}_t, \boldsymbol{w}_t)).$$

Since the left-hand side does not depend on $\xi_t$, using the same argument as in the proof of Theorem 4.1 we get

$$\mathbb{E}[F([\boldsymbol{x}^*]_t) - F(\hat{\boldsymbol{x}}_t)] \leq \mathbb{E}[2q(2\beta_t^{1/2}\sigma_{t-1}(\boldsymbol{x}_t, \boldsymbol{w}_t))].$$

Thus, using the same argument as in the derivation of equation 16, equation 17, equation 21 and equation 22, we obtain

$$\mathbb{E}\left[ \sum_{k=1}^{t} (F([\boldsymbol{x}^*]_k) - F(\hat{\boldsymbol{x}}_k)) \right] \leq 2t \sum_{i=1}^{n} \zeta_i h_i \left( \frac{1}{t} \sum_{j=1}^{s_i} 2^{\nu_{ij}} \lambda_{ij} \left( t C_{2,\nu_{ij},t} \right)^{1-\nu'_{ij}/2} (C_1 \gamma_t)^{\nu'_{ij}/2} \right). \qquad (25)$$

Here, $\beta_1, \ldots, \beta_t$ do not have the same distribution, and equation 18 does not holds. Nevertheless, from the monotonicity of $\kappa_t^{(1)}$, the following holds:

$$\left( \mathbb{E}\left[ \sum_{k=1}^{t} \beta_k^{\nu_{ij}\eta/2} \right] \right)^{1/\eta} \leq \left( t\mathbb{E}\left[ \beta_t^{\nu_{ij}\eta/2} \right] \right)^{1/\eta} = \left( t\mathbb{E}\left[ \beta_t^{\nu_{ij}/(2-\nu_{ij})} \right] \right)^{1-\nu_{ij}/2} = \left( t\mathbb{E}\left[ \beta_t^{\nu_{ij}/(2-\nu'_{ij})} \right] \right)^{1-\nu'_{ij}/2}.$$

Hence, we have equation 25. On the other hand, from the definition, the following inequality holds:

$$F(\boldsymbol{x}^*) - F([\boldsymbol{x}^*]_t) = \rho(f(\boldsymbol{x}^*, \cdot)) - \rho(f([\boldsymbol{x}^*]_t, \cdot)) \leq q_1 \left( \max_{\boldsymbol{w} \in \Omega} |f(\boldsymbol{x}^*, \boldsymbol{w})) - f([\boldsymbol{x}^*]_t, \boldsymbol{w}))| \right).$$

Let $L_{max} = \sup_{\boldsymbol{w} \in \Omega} \sup_{1 \leq j \leq d_1} \sup_{\boldsymbol{x} \in \mathcal{X}} \left| \frac{\partial}{\partial x_j} f(\boldsymbol{x}, \boldsymbol{w}) \right|$. Then, the following holds:

$$^{\forall}\boldsymbol{x}, \boldsymbol{x}' \in \mathcal{X}, \boldsymbol{w} \in \Omega, |f(\boldsymbol{x}, \boldsymbol{w}) - f(\boldsymbol{x}', \boldsymbol{w})| \leq L_{max} \|\boldsymbol{x} - \boldsymbol{x}'\|_1.$$

Therefore, we get

$$|f(\boldsymbol{x}^*, \boldsymbol{w}) - f([\boldsymbol{x}^*]_t, \boldsymbol{w})| \leq L_{max} \|\boldsymbol{x}^* - [\boldsymbol{x}^*]_t\|_1.$$

In addition, noting that $\|\boldsymbol{x}^* - [\boldsymbol{x}^*]_t\|_1 \leq d_1 r/\tau_t$, we obtain

$$|f(\boldsymbol{x}^*, \boldsymbol{w}) - f([\boldsymbol{x}^*]_t, \boldsymbol{w})| \leq L_{max} d_1 r/\tau_t.$$

Thus, since

$$q_1 \left( \max_{\boldsymbol{w} \in \Omega} |f(\boldsymbol{x}^*, \boldsymbol{w})) - f([\boldsymbol{x}^*]_t, \boldsymbol{w}))| \right) \leq q_1 \left( L_{max} d_1 r/\tau_t \right),$$

using the concavity of $q_1(a)$ we get

$$\mathbb{E}[F(\boldsymbol{x}^*) - F([\boldsymbol{x}^*]_t)] \leq \mathbb{E}\left[ q_1 \left( L_{max} d_1 r/\tau_t \right) \right] \leq q_1 \left( \mathbb{E}\left[ L_{max} d_1 r/\tau_t \right] \right). \qquad (26)$$

Moreover, $L_{max}$ satisfies $\mathbb{E}[L_{max}] \leq b_1(\sqrt{\log(a_1 d_1 |\Omega|)} + \sqrt{\pi}/2)$ (see, Appendix C.7). This implies that $\mathbb{E}\left[ L_{max} d_1 r/\tau_t \right] \leq t^{-2}$. By substituting this into equation 26, we have

$$\mathbb{E}[F(\boldsymbol{x}^*) - F([\boldsymbol{x}^*]_t)] \leq q_1 \left( t^{-2} \right).$$

Therefore, the following holds:

$$\mathbb{E}\left[ \sum_{k=1}^{t} (F(\boldsymbol{x}^*) - F([\boldsymbol{x}^*]_k)) \right] \leq \sum_{k=1}^{t} q_1(k^{-2}) = t \sum_{k=1}^{t} t^{-1} q_1(k^{-2}) \leq t q_1 \left( t^{-1} \sum_{k=1}^{t} k^{-2} \right) \leq t q_1 \left( \frac{\pi^2}{6t} \right). \qquad (27)$$

Finally, by substituting equation 25 and equation 27 into equation 24, we obtain the desired result. $\qquad \square$

## C.7 Upper Bound of $\mathbb{E}[L_{max}]$

*Proof.* For any $j \in \{1, \ldots, d_1\}$ and $\boldsymbol{w} \in \Omega$, if $\sup_{\boldsymbol{x} \in \mathcal{X}} \left| \frac{\partial}{\partial x_j} f(\boldsymbol{x}, \boldsymbol{w}) \right| \leq L$, then $L_{max} \leq L$. Therefore, the following inequality holds:

$$\mathbb{P}(L_{max} > L) \leq \sum_{j=1}^{d_1} \sum_{\boldsymbol{w} \in \Omega} \mathbb{P}\left( \sup_{\boldsymbol{x} \in \mathcal{X}} \left| \frac{\partial f(\boldsymbol{x}, \boldsymbol{w})}{\partial x_j} \right| > L \right) \leq \sum_{j=1}^{d_1} \sum_{\boldsymbol{w} \in \Omega} a_1 \exp\left( -\left( \frac{L}{b_1} \right)^2 \right)$$
$$= a_1 d_1 |\Omega| \exp\left( -\left( \frac{L}{b_1} \right)^2 \right).$$

Let $a_1 d_1 |\Omega| = J$. Then, using the property of the expectation in non-negative random variables, $\mathbb{E}[L_{max}]$ can be evaluated as follows:

$$\begin{aligned}
\mathbb{E}[L_{max}] &= \int_0^\infty \mathbb{P}(L_{max} > L) \mathrm{d}L \\
&\leq \int_0^\infty \min\{1, Je^{-(L/b_1)^2}\} \mathrm{d}L \\
&= b_1 \sqrt{\log J} + \int_{b_1 \sqrt{\log J}}^\infty Je^{-(L/b_1)^2} \mathrm{d}L \\
&= b_1 \sqrt{\log J} + Jb_1\sqrt{\pi} \int_{b_1\sqrt{\log J}}^\infty \frac{1}{\sqrt{2\pi(b_1^2/2)}} e^{-(L/b_1)^2} \mathrm{d}L \\
&= b_1 \sqrt{\log J} + Jb_1\sqrt{\pi} \left( 1 - \Phi\left( \sqrt{2\log J} \right) \right) \\
&\leq b_1 \sqrt{\log J} + \frac{b_1\sqrt{\pi}}{2},
\end{aligned}$$

where $\Phi(\cdot)$ is the cumulative distribution function of the standard normal distribution, and the last inequality is derived by Lemma H.3 in Takeno et al. (2023). □

## C.8 Proof of Theorem A.3

*Proof.* For each $t \geq 1$, let $\tau_t^\dagger = \lceil b_2 d_2 rt^2(\sqrt{\log(a_2 d_2 |\mathcal{X}|)} + \sqrt{\pi}/2) \rceil$. Suppose that $\Omega_t$ is a set of discretization for $\Omega$ with each coordinate is equally divided into $\tau_t^\dagger$. Note that $|\Omega_t| = (\tau_t^\dagger)^{d_2}$. For each $\boldsymbol{w} \in \Omega$, let $[\boldsymbol{w}]_t$ be the element of $\Omega_t$ closest to $\boldsymbol{w}$. Then, the following equality holds:

$$r_t^\dagger = F(\boldsymbol{x}^*) - F(\hat{\boldsymbol{x}}_t^\dagger) = F(\boldsymbol{x}^*) - F_t^\dagger(\boldsymbol{x}^*) + F_t^\dagger(\boldsymbol{x}^*) - F_t^\dagger(\hat{\boldsymbol{x}}_t^\dagger) + F_t^\dagger(\hat{\boldsymbol{x}}_t^\dagger) - F(\hat{\boldsymbol{x}}_t^\dagger).$$

This implies that

$$\mathbb{E}[R_t^\dagger] = \mathbb{E}\left[ \sum_{k=1}^t (F(\boldsymbol{x}^*) - F_k^\dagger(\boldsymbol{x}^*)) \right] + \mathbb{E}\left[ \sum_{k=1}^t (F_k^\dagger(\boldsymbol{x}^*) - F_k^\dagger(\hat{\boldsymbol{x}}_k^\dagger)) \right] + \mathbb{E}\left[ \sum_{k=1}^t (F_k^\dagger(\hat{\boldsymbol{x}}_k^\dagger) - F(\hat{\boldsymbol{x}}_k^\dagger)) \right]. \quad (28)$$

Let $\xi_t$ be a realization from the chi-squared distribution with two degrees of freedom, and let $\delta = \frac{1}{\exp(\xi_t/2)}$. Then, from the proof of Lemma 5.1 in Srinivas et al. (2010), with probability at least $1 - \delta$, the following holds for any $(\boldsymbol{x}, \boldsymbol{w}) \in \mathcal{X} \times \Omega_t$:

$$l_{t-1,\delta}(\boldsymbol{x}, \boldsymbol{w}) \equiv \mu_{t-1}(\boldsymbol{x}, \boldsymbol{w}) - \beta_\delta^{1/2} \sigma_{t-1}(\boldsymbol{x}, \boldsymbol{w}) \leq f(\boldsymbol{x}, \boldsymbol{w}) \leq \mu_{t-1}(\boldsymbol{x}, \boldsymbol{w}) + \beta_\delta^{1/2} \sigma_{t-1}(\boldsymbol{x}, \boldsymbol{w}) \equiv u_{t-1,\delta}(\boldsymbol{x}, \boldsymbol{w}),$$

where $\beta_\delta = 2\log(|\mathcal{X} \times \Omega_t|/\delta)$. From the definition of $\delta$, since $|\mathcal{X} \times \Omega_t| = (\tau_t^\dagger)^{d_2}|\mathcal{X}| = \kappa_t^{(2)}$, we have $\beta_\delta = 2\log(\kappa_t^{(2)}) + \xi_t = \beta_t$. Hence, the following inequality holds with probability at least $1 - \delta$:

$$l_{t-1}(\boldsymbol{x}, \boldsymbol{w}) \leq f(\boldsymbol{x}, \boldsymbol{w}) \leq u_{t-1}(\boldsymbol{x}, \boldsymbol{w}).$$

Therefore, the following holds for any $\boldsymbol{w} \in \Omega$:

$$l_{t-1}(\boldsymbol{x}, [\boldsymbol{w}]_t) \leq f(\boldsymbol{x}, [\boldsymbol{w}]_t) \leq u_{t-1}(\boldsymbol{x}, [\boldsymbol{w}]_t).$$

Thus, for the function $f_t^\dagger(\boldsymbol{x}, \boldsymbol{w})$ with respect to $\boldsymbol{w}$, $f_t^\dagger(\boldsymbol{x}, \cdot) \in G_{t-1}^\dagger(\boldsymbol{x})$ holds. Here, from the theorem's assumption, since $\mathrm{ucb}_{t-1}^\dagger(\boldsymbol{x})$ and $\mathrm{lcb}_{t-1}^\dagger(\boldsymbol{x})$ satisfy equation 10, the following holds:

$$\mathrm{lcb}_{t-1}^\dagger(\boldsymbol{x}) \leq F_t^\dagger(\boldsymbol{x}) \leq \mathrm{ucb}_{t-1}^\dagger(\boldsymbol{x}).$$

Hence, we obtain

$$F_t^\dagger(\boldsymbol{x}^*) - F_t^\dagger(\hat{\boldsymbol{x}}_t^\dagger) \leq 2(\mathrm{ucb}_{t-1}^\dagger(\boldsymbol{x}_t) - \mathrm{lcb}_{t-1}^\dagger(\boldsymbol{x}_t)).$$

Furthermore, from Assumption A.5, there exists a function

$$q^\dagger(a) = \sum_{i=1}^{n} \zeta_i h_i^\dagger \left( \sum_{j=1}^{s_i} \lambda_{ij} a^{\nu_{ij}} \right)$$

such that $\mathrm{ucb}_{t-1}^\dagger(\boldsymbol{x}_t) - \mathrm{lcb}_{t-1}^\dagger(\boldsymbol{x}_t) \leq q^\dagger(2\beta_t^{1/2}\sigma_{t-1}(\boldsymbol{x}_t, \boldsymbol{w}_t))$. This implies that

$$F_t^\dagger(\boldsymbol{x}^*) - F_t^\dagger(\hat{\boldsymbol{x}}_t^\dagger) \leq 2q^\dagger(2\beta_t^{1/2}\sigma_{t-1}(\boldsymbol{x}_t, \boldsymbol{w}_t)).$$

Noting that the left-hand side does not depend on $\xi_t$, using the same argument as in the proof of Theorem 4.1 we get

$$\mathbb{E}[F_t^\dagger(\boldsymbol{x}^*) - F_t^\dagger(\hat{\boldsymbol{x}}_t^\dagger)] \leq \mathbb{E}[2q^\dagger(2\beta_t^{1/2}\sigma_{t-1}(\boldsymbol{x}_t, \boldsymbol{w}_t))].$$

Therefore, using the same argument as in the derivation of equation 16, equation 17, equation 21 and equation 22, we obtain

$$\mathbb{E}\left[ \sum_{k=1}^{t}(F_k^\dagger(\boldsymbol{x}^*) - F_k^\dagger(\hat{\boldsymbol{x}}_k^\dagger)) \right] \leq 2t \sum_{i=1}^{n} \zeta_i h_i^\dagger \left( \frac{1}{t} \sum_{j=1}^{s_i} 2^{\nu_{ij}} \lambda_{ij} \left( tC_{2,\nu_{ij},t} \right)^{1-\nu'_{ij}/2} (C_1\gamma_t)^{\nu'_{ij}/2} \right). \tag{29}$$

On the other hand, from the assumption, the following inequality holds:

$$F(\boldsymbol{x}^*) - F_t^\dagger(\boldsymbol{x}^*) = \rho(f(\boldsymbol{x}^*, \cdot)) - \rho(f_t^\dagger(\boldsymbol{x}^*, \cdot)) \leq q_2 \left( \max_{\boldsymbol{w} \in \Omega} |f(\boldsymbol{x}^*, \boldsymbol{w})) - f(\boldsymbol{x}^*, [\boldsymbol{w}]_t))| \right).$$

Let $L_{max}^\dagger = \sup_{\boldsymbol{x} \in \mathcal{X}} \sup_{1 \leq j \leq d_2} \sup_{\boldsymbol{w} \in \Omega} \left| \frac{\partial}{\partial w_j} f(\boldsymbol{x}, \boldsymbol{w}) \right|$. Then, the following holds:

$$\forall \boldsymbol{w}, \boldsymbol{w}' \in \Omega, \boldsymbol{x} \in \mathcal{X}, |f(\boldsymbol{x}, \boldsymbol{w}) - f(\boldsymbol{x}, \boldsymbol{w}')| \leq L_{max}^\dagger \|\boldsymbol{w} - \boldsymbol{w}'\|_1.$$

Hence, we have

$$|f(\boldsymbol{x}^*, \boldsymbol{w}) - f(\boldsymbol{x}^*, [\boldsymbol{w}]_t)| \leq L_{max}^\dagger \|\boldsymbol{w} - [\boldsymbol{w}]_t\|_1.$$

In addition, noting that $\|\boldsymbol{w} - [\boldsymbol{w}]_t\|_1 \leq d_2 r / \tau_t^\dagger$, we get

$$|f(\boldsymbol{x}^*, \boldsymbol{w}) - f(\boldsymbol{x}^*, [\boldsymbol{w}]_t)| \leq L_{max}^\dagger d_2 r / \tau_t^\dagger.$$

Thus, since

$$q_2 \left( \max_{\boldsymbol{w} \in \Omega} |f(\boldsymbol{x}^*, \boldsymbol{w})) - f(\boldsymbol{x}^*, [\boldsymbol{w}]_t))| \right) \leq q_2 \left( L_{max}^\dagger d_2 r / \tau_t^\dagger \right),$$

using the concavity of $q_2(a)$ we obtain

$$\mathbb{E}[F(\boldsymbol{x}^*) - F_t^\dagger(\boldsymbol{x}^*)] \leq \mathbb{E}\left[ q_2 \left( L_{max}^\dagger d_2 r / \tau_t^\dagger \right) \right] \leq q_2 \left( \mathbb{E}\left[ L_{max}^\dagger d_2 r / \tau_t^\dagger \right] \right). \tag{30}$$

Here, by using the same argument as in Appendix C.7, we get $\mathbb{E}[L_{max}^{\dagger}] \leq b_2(\sqrt{\log(a_2 d_2 |\mathcal{X}|)} + \sqrt{\pi}/2)$. This implies that $\mathbb{E}\left[L_{max}^{\dagger} d_2 r / \tau_t^{\dagger}\right] \leq t^{-2}$. Substituting this into equation 30, we have

$$\mathbb{E}[F(\boldsymbol{x}^*) - F_t^{\dagger}(\boldsymbol{x}^*)] \leq q_2\left(t^{-2}\right).$$

Therefore, the following inequality holds:

$$\mathbb{E}\left[\sum_{k=1}^{t}(F(\boldsymbol{x}^*) - F_k^{\dagger}(\boldsymbol{x}^*))\right] \leq \sum_{k=1}^{t} q_2(k^{-2}) = t\sum_{k=1}^{t} t^{-1} q_2(k^{-2}) \leq t q_2\left(t^{-1}\sum_{k=1}^{t} k^{-2}\right) \leq t q_2\left(\frac{\pi^2}{6t}\right). \quad (31)$$

By using the similar argument, the following inequality also holds:

$$\mathbb{E}\left[\sum_{k=1}^{t}(F_k^{\dagger}(\hat{\boldsymbol{x}}_k^{\dagger}) - F(\hat{\boldsymbol{x}}_k^{\dagger}))\right] \leq t q_2\left(\frac{\pi^2}{6t}\right). \quad (32)$$

Finally, substituting equation 29, equation 31 and equation 32 into equation 28, we obtain the desired result. $\qquad\square$

### C.9 Proof of Theorem A.5

*Proof.* For $r_t^{\dagger}$, the following holds:

$$\mathbb{E}[r_t^{\dagger}] = \mathbb{E}[F(\boldsymbol{x}^*) - F(\hat{\boldsymbol{x}}_t^{\dagger})] = \mathbb{E}[F(\boldsymbol{x}^*) - F_t^{\dagger}(\hat{\boldsymbol{x}}_t^{\dagger})] + \mathbb{E}[F_t^{\dagger}(\hat{\boldsymbol{x}}_t^{\dagger}) - F(\hat{\boldsymbol{x}}_t^{\dagger})] \leq \mathbb{E}[F(\boldsymbol{x}^*) - F_t^{\dagger}(\hat{\boldsymbol{x}}_t^{\dagger})] + \frac{1}{t^2}. \quad (33)$$

Here, we define $\tilde{t}$ as follows:

$$\tilde{t} = \arg\min_{1 \leq i \leq t} \mathbb{E}_{t-1}[F(\boldsymbol{x}^*) - F_t^{\dagger}(\hat{\boldsymbol{x}}_i^{\dagger})].$$

Then, noting that

$$\mathbb{E}_{t-1}[F(\boldsymbol{x}^*) - F_t^{\dagger}(\hat{\boldsymbol{x}}_{\tilde{t}}^{\dagger})] \leq \frac{1}{t}\sum_{i=1}^{t} \mathbb{E}_{t-1}[F(\boldsymbol{x}^*) - F_t^{\dagger}(\hat{\boldsymbol{x}}_i^{\dagger})],$$

the following inequality holds:

$$\mathbb{E}[F(\boldsymbol{x}^*) - F_t^{\dagger}(\hat{\boldsymbol{x}}_{\tilde{t}}^{\dagger})] \leq \frac{1}{t}\sum_{i=1}^{t} \mathbb{E}[F(\boldsymbol{x}^*) - F_t^{\dagger}(\hat{\boldsymbol{x}}_i^{\dagger})] = \frac{1}{t}\sum_{i=1}^{t} \mathbb{E}[F(\boldsymbol{x}^*) - F(\hat{\boldsymbol{x}}_i^{\dagger}) + F(\hat{\boldsymbol{x}}_i^{\dagger}) - F_t^{\dagger}(\hat{\boldsymbol{x}}_i^{\dagger})]$$

$$\leq \frac{\mathbb{E}[R_t^{\dagger}]}{t} + \frac{1}{t}\sum_{i=1}^{t}\frac{1}{t^2} = \frac{\mathbb{E}[R_t^{\dagger}]}{t} + \frac{1}{t^2}. \quad (34)$$

Next, we show $\tilde{t} = t$. From the definition, $\tilde{t}$ can be rewritten as follows:

$$\tilde{t} = \arg\max_{1 \leq i \leq t} \mathbb{E}_{t-1}[F_t^{\dagger}(\hat{\boldsymbol{x}}_i^{\dagger})].$$

Let $\Omega_t = \{\boldsymbol{w}^{(1)}, \ldots, \boldsymbol{w}^{(J)}\}$. Then, the following equality holds:

$$F_t^{\dagger}(\hat{\boldsymbol{x}}_i^{\dagger}) = \mathbb{E}_{\boldsymbol{w}}[f(\hat{\boldsymbol{x}}_i^{\dagger}, [\boldsymbol{w}]_t)] = \sum_{j=1}^{J} \mathbb{P}_{\boldsymbol{w}}(\boldsymbol{w} = \boldsymbol{w}^{(j)}) f(\hat{\boldsymbol{x}}_i^{\dagger}, \boldsymbol{w}^{(j)}).$$

Therefore, we get

$$\mathbb{E}_{t-1}[F_t^{\dagger}(\hat{\boldsymbol{x}}_i^{\dagger})] = \sum_{j=1}^{J} \mathbb{P}_{\boldsymbol{w}}(\boldsymbol{w} = \boldsymbol{w}^{(j)}) \mathbb{E}_{t-1}[f(\hat{\boldsymbol{x}}_i^{\dagger}, \boldsymbol{w}^{(j)})] = \sum_{j=1}^{J} \mathbb{P}_{\boldsymbol{w}}(\boldsymbol{w} = \boldsymbol{w}^{(j)}) \mu_{t-1}(\hat{\boldsymbol{x}}_i^{\dagger}, \boldsymbol{w}^{(j)})$$

$$= \mathbb{E}_{\boldsymbol{w}}[\mu_{t-1}(\hat{\boldsymbol{x}}_i^{\dagger}, [\boldsymbol{w}]_t)]$$

$$= \mathbb{E}_{\boldsymbol{w}}[\mu_{t-1}^{\dagger}(\hat{\boldsymbol{x}}_i^{\dagger}, \boldsymbol{w})] = \rho(\mu_{t-1}^{\dagger}(\hat{\boldsymbol{x}}_i^{\dagger}, \cdot)).$$

Hence, from the definition of $\hat{\boldsymbol{x}}_t^\dagger$, we have $\mathbb{E}_{t-1}[F_t^\dagger(\hat{\boldsymbol{x}}_i^\dagger)] \leq \mathbb{E}_{t-1}[F_t^\dagger(\hat{\boldsymbol{x}}_t^\dagger)]$. This implies that $\tilde{t} = t$. Hence, equation 34 can be expressed as follows:

$$\mathbb{E}[F(\boldsymbol{x}^*) - F_t^\dagger(\hat{\boldsymbol{x}}_t^\dagger)] \leq \frac{\mathbb{E}[R_t^\dagger]}{t} + \frac{1}{t^2}. \tag{35}$$

Thus, substituting equation 35 into equation 33, we obtain the desired result. $\qquad\square$

### C.10 Proof of Theorem A.6

*Proof.* For $r_t^\dagger$, the following equality holds:

$$r_t^\dagger = F(\boldsymbol{x}^*) - F(\hat{\boldsymbol{x}}_t^\dagger) = F(\boldsymbol{x}^*) - F_t^\dagger([\boldsymbol{x}^*]_t) + F_t^\dagger([\boldsymbol{x}^*]_t) - F_t^\dagger(\hat{\boldsymbol{x}}_t^\dagger) + F_t^\dagger(\hat{\boldsymbol{x}}_t^\dagger) - F(\hat{\boldsymbol{x}}_t^\dagger).$$

Let $\tilde{L}_{max} = \sup_{1 \leq j \leq d} \sup_{\boldsymbol{\theta} \in \Theta} \left| \frac{\partial f(\boldsymbol{\theta})}{\partial \theta_j} \right|$. From Assumption A.7, the following inequalities hold:

$$F(\boldsymbol{x}^*) - F_t^\dagger([\boldsymbol{x}^*]_t) \leq q_3 \left( \max_{\boldsymbol{w} \in \Omega} |f(\boldsymbol{x}^*, \boldsymbol{w}) - f([\boldsymbol{x}^*]_t, [\boldsymbol{w}]_t)| \right),$$

$$F_t^\dagger(\hat{\boldsymbol{x}}_t^\dagger) - F(\hat{\boldsymbol{x}}_t^\dagger) \leq q_3 \left( \max_{\boldsymbol{w} \in \Omega} |f(\hat{\boldsymbol{x}}_t^\dagger, \boldsymbol{w}) - f(\hat{\boldsymbol{x}}_t^\dagger, [\boldsymbol{w}]_t)| \right).$$

In addition, the following inequalities hold:

$$|f(\boldsymbol{x}^*, \boldsymbol{w}) - f([\boldsymbol{x}^*]_t, [\boldsymbol{w}]_t)| \leq \tilde{L}_{max} \|(\boldsymbol{x}^*, \boldsymbol{w}) - ([\boldsymbol{x}^*]_t, [\boldsymbol{w}]_t)\|_1 \leq \tilde{L}_{max} \frac{dr}{\tilde{\tau}_t},$$

$$|f(\hat{\boldsymbol{x}}_t^\dagger, \boldsymbol{w}) - f(\hat{\boldsymbol{x}}_t^\dagger, [\boldsymbol{w}]_t)| \leq \tilde{L}_{max} \|(\hat{\boldsymbol{x}}_t^\dagger, \boldsymbol{w}) - (\hat{\boldsymbol{x}}_t^\dagger, [\boldsymbol{w}]_t)\|_1 \leq \tilde{L}_{max} \frac{dr}{\tilde{\tau}_t}.$$

Here, under Assumption A.6, from Lemma H.1 in Takeno et al. (2023), we have $\mathbb{E}[\tilde{L}_{max}] \leq b_3(\sqrt{\log(a_3 d)} + \sqrt{\pi}/2)$. Hence, we get

$$\mathbb{E}[F(\boldsymbol{x}^*) - F_t^\dagger([\boldsymbol{x}^*]_t)] \leq q_3(t^{-2}), \ \mathbb{E}[F_t^\dagger(\hat{\boldsymbol{x}}_t^\dagger) - F(\hat{\boldsymbol{x}}_t^\dagger)] \leq q_3(t^{-2}).$$

On the other hand, noting that $[\boldsymbol{x}^*]_t, \hat{\boldsymbol{x}}_t^\dagger \in \mathcal{X}_t \cup \{\hat{\boldsymbol{x}}_t^\dagger\}$ and $|(\mathcal{X}_t \cup \{\hat{\boldsymbol{x}}_t^\dagger\}) \times \Omega_t| = (1 + \tilde{\tau}_t^{d_1})\tilde{\tau}_t^{d_2} = \kappa_t^{(3)}$, by using the same argument as in the proof of Theorem A.3, we obtain

$$\mathbb{E}[F_t^\dagger([\boldsymbol{x}^*]_t) - F_t^\dagger(\hat{\boldsymbol{x}}_t^\dagger)] \leq \mathbb{E}[2q^\dagger(2\beta_t^{1/2}\sigma_{t-1}(\boldsymbol{x}_t, \boldsymbol{w}_t))].$$

By combining these, and using the same argument as in the proof of Theorem A.3, we get the desired result. $\qquad\square$

### C.11 Proof of Theorems B.1–B.4

*Proof.* For each $t \geq 1$, let $D_{t-1} = \{(\boldsymbol{x}_1, \boldsymbol{w}_1, y_1, \beta_1), \ldots, (\boldsymbol{x}_{t-1}, \boldsymbol{w}_{t-1}, y_{t-1}, \beta_{t-1})\}$ and $D_0 = \emptyset$. Suppose that $\xi_t$ is a realization from the chi-squared distribution with two degrees of freedom. Then, by letting $\delta = \frac{1}{\exp(\xi_t/2)}$, using the same argument as in the proof of Theorem 4.1, the following holds with probability at least $1 - \delta$:

$$F(\boldsymbol{x}^*) - F(\hat{\boldsymbol{x}}_t) \leq 2(\text{ucb}_{t-1}(\boldsymbol{x}_t) - \text{lcb}_{t-1}(\boldsymbol{x}_t)).$$

Furthermore, from Assumption 4.1, there exists a function

$$q(a) = \sum_{i=1}^{n} \zeta_i h_i \left( \sum_{j=1}^{s_i} \lambda_{ij} a^{\nu_{ij}} \right)$$

such that $\mathrm{ucb}_{t-1}(\boldsymbol{x}_t) - \mathrm{lcb}_{t-1}(\boldsymbol{x}_t) \leq q(2\beta_t^{1/2}\sigma_{t-1}(\boldsymbol{x}_t, \boldsymbol{w}_t^{(max)}))$, where $\boldsymbol{w}_t^{(max)} = \arg\max_{\boldsymbol{w}\in\Omega} 2\beta_t^{1/2}\sigma_{t-1}(\boldsymbol{x}_t, \boldsymbol{w})$. Hence, the following inequality holds:

$$F(\boldsymbol{x}^*) - F(\hat{\boldsymbol{x}}_t) \leq 2q(2\beta_t^{1/2}\sigma_{t-1}(\boldsymbol{x}_t, \boldsymbol{w}_t^{(max)})).$$

Therefore, by using the same argument as in the proof of Theorem 4.1, we have

$$\mathbb{E}[r_t] = \mathbb{E}[F(\boldsymbol{x}^*) - F(\hat{\boldsymbol{x}}_t)] \leq \mathbb{E}[2q(2\beta_t^{1/2}\sigma_{t-1}(\boldsymbol{x}_t, \boldsymbol{w}_t^{(max)}))].$$

Note that under the uncontrollable setting, the expected value of the sum of the posterior variances in equation 19 and equation 20 is replaced by the following:

$$\mathbb{E}\left[\sum_{k=1}^t \sigma_{k-1}^2(\boldsymbol{x}_k, \boldsymbol{w}_k^{(max)})\right].$$

Here, for each $k$, under the given $D_{k-1}$ and $\beta_k$, $\sigma_{k-1}^2(\boldsymbol{x}_k, \boldsymbol{w}_k^{(max)})$ is a constant value. Then, the following holds:

$$\sigma_{k-1}^2(\boldsymbol{x}_k, \boldsymbol{w}_k^{(max)}) \leq \sum_{j=1}^J \sigma_{k-1}^2(\boldsymbol{x}_k, \boldsymbol{w}^{(j)}) \leq p_{min}^{-1}\sum_{j=1}^J p_j\sigma_{k-1}^2(\boldsymbol{x}_k, \boldsymbol{w}^{(j)}).$$

Since $\boldsymbol{w}_k$ does not depend on $D_{k-1}$ and $\beta_k$, the following equality holds:

$$\sum_{j=1}^J p_j\sigma_{k-1}^2(\boldsymbol{x}_k, \boldsymbol{w}^{(j)}) = \mathbb{E}[\sigma_{k-1}^2(\boldsymbol{x}_k, \boldsymbol{w}_k)|D_{k-1}, \beta_k].$$

Hence, we have

$$\mathbb{E}[\sigma_{k-1}^2(\boldsymbol{x}_k, \boldsymbol{w}_k^{(max)})] = p_{min}^{-1}\mathbb{E}[\sigma_{k-1}^2(\boldsymbol{x}_k, \boldsymbol{w}_k)].$$

Therefore, we get

$$\mathbb{E}\left[\sum_{k=1}^t \sigma_{k-1}^2(\boldsymbol{x}_k, \boldsymbol{w}_k^{(max)})\right] \leq p_{min}^{-1}C_1 = C_1'.$$

By combining these, and using the same argument as in the proof of Theorem 4.1, we obtain Theorem B.1. Theorems B.2–B.4 can also be obtained by using the same argument. □

## C.12 Proof of Theorems B.5–B.8

*Proof.* To show Theorems B.5–B.8, we evaluate $\mathbb{E}[\sigma_{k-1}^2(\boldsymbol{x}_k, \boldsymbol{w}_k^{(max)})]$ by using the same argument as in the proof of Theorems B.1–B.4. Under the given

$$\check{D}_{k-1} = \{(\boldsymbol{x}_1, \boldsymbol{w}_1, y_1, \beta_1), \ldots, (\boldsymbol{x}_{k-1}, \boldsymbol{w}_{k-1}, y_{k-1}, \beta_{k-1}), \boldsymbol{x}_k, \beta_k\},$$

$\sigma_{k-1}^2(\boldsymbol{x}_k, \boldsymbol{w}_k^{(max)})$ is a constant. Moreover, since $\check{D}_{k-1}$ and $\boldsymbol{w}_k$ are mutually independent, using the tower property, the partition $\mathcal{S}_k$ satisfies

$$p_{min,k}^{-1}\mathbb{E}_{\boldsymbol{w}_k}[\sigma_{k-1}^2(\boldsymbol{x}_k, \boldsymbol{w}_k)|\check{D}_{k-1}] = p_{min,k}^{-1}\sum_{\tilde{\Omega}\in\mathcal{S}_k}\mathbb{P}_{\boldsymbol{w}_k}(\boldsymbol{w}_k \in \tilde{\Omega})\mathbb{E}_{\boldsymbol{w}_k}[\sigma_{k-1}^2(\boldsymbol{x}_k, \boldsymbol{w}_k)|\check{D}_{k-1}, \boldsymbol{w}_k \in \tilde{\Omega}]$$

$$\geq \sum_{\tilde{\Omega}\in\mathcal{S}_k}\mathbb{E}_{\boldsymbol{w}_k}[\sigma_{k-1}^2(\boldsymbol{x}_k, \boldsymbol{w}_k)|\check{D}_{k-1}, \boldsymbol{w}_k \in \tilde{\Omega}],$$

where the last inequality is derived by using the inequality

$$p_{min,k}^{-1}\mathbb{P}_{\boldsymbol{w}_k}(\boldsymbol{w}_k \in \tilde{\Omega}) \geq 1.$$

Here, for $\tilde{\Omega}_j \in \mathcal{S}_k$, if $\boldsymbol{w}_k^{(max)} \in \tilde{\Omega}_j$, from Assumption B.3, the following holds for any $\boldsymbol{w} \in \tilde{\Omega}_j$:

$$\sigma_{k-1}^2(\boldsymbol{x}_k, \boldsymbol{w}_k^{(max)}) - \iota_k \leq \sigma_{k-1}^2(\boldsymbol{x}_k, \boldsymbol{w}).$$

Therefore, we get

$$\sigma_{k-1}^2(\boldsymbol{x}_k, \boldsymbol{w}_k^{(max)}) - \iota_k \leq \mathbb{E}_{\boldsymbol{w}_k}[\sigma_{k-1}^2(\boldsymbol{x}_k, \boldsymbol{w}_k) | \check{D}_{k-1}, \boldsymbol{w}_k \in \tilde{\Omega}_j].$$

Thus, we obtain

$$\sigma_{k-1}^2(\boldsymbol{x}_k, \boldsymbol{w}_k^{(max)}) - \iota_k \leq \sum_{\tilde{\Omega} \in \mathcal{S}_k} \mathbb{E}_{\boldsymbol{w}_k}[\sigma_{k-1}^2(\boldsymbol{x}_k, \boldsymbol{w}_k) | \check{D}_{k-1}, \boldsymbol{w}_k \in \tilde{\Omega}] \leq p_{min,k}^{-1} \mathbb{E}_{\boldsymbol{w}_k}[\sigma_{k-1}^2(\boldsymbol{x}_k, \boldsymbol{w}_k) | \check{D}_{k-1}].$$

This implies that

$$\mathbb{E}[\sigma_{k-1}^2(\boldsymbol{x}_k, \boldsymbol{w}_k^{(max)})] \leq \iota_k + p_{min,k}^{-1} \mathbb{E}[\sigma_{k-1}^2(\boldsymbol{x}_k, \boldsymbol{w}_k)].$$

Hence, the following inequality holds:

$$\begin{aligned}
\sum_{k=1}^t \mathbb{E}[\sigma_{k-1}^2(\boldsymbol{x}_k, \boldsymbol{w}_k^{(max)})] &\leq \sum_{k=1}^t \iota_k + \sum_{k=1}^t p_{min,k}^{-1} \mathbb{E}[\sigma_{k-1}^2(\boldsymbol{x}_k, \boldsymbol{w}_k)] \\
&= \varphi_t + \sum_{k=1}^t p_{min,k}^{-1} \mathbb{E}[\sigma_{k-1}^2(\boldsymbol{x}_k, \boldsymbol{w}_k)] \\
&\leq \varphi_t + p_{min,t}^{-1} \sum_{k=1}^t \mathbb{E}[\sigma_{k-1}^2(\boldsymbol{x}_k, \boldsymbol{w}_k)],
\end{aligned}$$

where the last inequality is derived by $p_{min,1} \geq \cdots \geq p_{min,t}$. Therefore, since

$$\varphi_t + p_{min,t}^{-1} \sum_{k=1}^t \mathbb{E}[\sigma_{k-1}^2(\boldsymbol{x}_k, \boldsymbol{w}_k)] \leq \varphi_t + p_{min,t}^{-1} C_1 \gamma_t,$$

we have the desired result. $\qquad\square$

## D  Experimental Details and Additional Experiments

In this section, we give details of the numerical experiments.

### D.1  Experimental Details

In the 2D synthetic function setting, the black-box function $f$ was sampled from $\mathcal{GP}(0, k)$, where the kernel function $k$ is given by

$$k(\boldsymbol{\theta}, \boldsymbol{\theta}') = \exp(-\|\boldsymbol{\theta} - \boldsymbol{\theta}'\|_2^2/2), \ \boldsymbol{\theta}, \boldsymbol{\theta}' \in \mathcal{X} \times \Omega.$$

In the 4D synthetic function setting, $f_{\mathrm{H}}(a, b)$ is defined as follows:

$$f_{\mathrm{H}}(a, b) = \frac{-\{(a^2 + b - 11)^2 + (a + b^2 - 7)^2\} + 104.8905}{\sqrt{3281.531}}.$$

In the 6D synthetic function setting, we first generated the sample paths $f_1, f_2, f_3, f_4$ independently from $\mathcal{GP}(0, k)$ defined on $\{-2, -4/3, -2/3, 0, 2/3, 4/3, 2\}^3$. Here, for $\boldsymbol{\theta}, \boldsymbol{\theta}' \in \mathbb{R}^3$, we used

$$k(\boldsymbol{\theta}, \boldsymbol{\theta}') = \exp(-\|\boldsymbol{\theta} - \boldsymbol{\theta}'\|_2^2/1.75).$$

Using this, we defined the black-box function $f$ as follows:

$$f(x_1, x_2, x_3, w_1, w_2, w_3) = f_1(x_1, x_2, x_3) + f_2(x_2, x_3, w_1) + f_3(x_3, w_1, w_2) + f_4(w_1, w_2, w_3).$$

For the setting in the 2D (6D) synthetic function setting, the black-box function was generated for each simulation based on the above, and a total of 100 different black-box functions were used as the true function.

For the probability mass function $p(\boldsymbol{w})$, we used $p(\boldsymbol{w}) = 1/50$ for the 2D synthetic function setting, and $p(\boldsymbol{w}) = \tilde{\phi}(w_1)\tilde{\phi}(w_2)$ for the 4D synthetic function setting. Here, for $a \in \{-2.5 + 2.5(i-1)/7 \mid i = 1, \dots, 15\} \equiv A$, $\tilde{\phi}(a)$ is defined as follows:

$$\tilde{\phi}(a) = \frac{0.25\phi(a-1) + 0.75\phi(a+5)}{\sum_{a' \in A}\{0.25\phi(a'-1) + 0.75\phi(a'+5)\}},$$

where $\phi(\cdot)$ is the probability density function of the standard normal distribution. In the 6D synthetic function setting, we used $p(\boldsymbol{w}) = \tilde{\phi}_1(w_1)\tilde{\phi}_2(w_2)\tilde{\phi}_3(w_3)$, where

$$\tilde{\phi}_1(b) = \frac{\phi(b-1)}{\sum_{b' \in B}\phi(b'-1)}, \ \ \tilde{\phi}_2(b) = \frac{\phi(b)}{\sum_{b' \in B}\phi(b')}, \ \ \tilde{\phi}_3(b) = \frac{\phi(b+1)}{\sum_{b' \in B}\phi(b'+1)}$$

and $b \in \{-2 + 2(i-1)/3 \mid i = 1, \dots, 7\} \equiv B$.

The details of each acquisition function are as follows:

**Random**  In Random, $\boldsymbol{x}_t$ was chosen uniformly at random from $\mathcal{X}$, and $\boldsymbol{w}_t$ was chosen randomly from $\Omega$ based on the probability mass function $p(\boldsymbol{w})$.

**US**  In US, $\boldsymbol{x}_t$ and $\boldsymbol{w}_t$ were chosen by $(\boldsymbol{x}_t, \boldsymbol{w}_t) = \arg\max_{(\boldsymbol{x}, \boldsymbol{w}) \in \mathcal{X} \times \Omega} \sigma_{t-1}^2(\boldsymbol{x}, \boldsymbol{w})$.

**BQ**  In BQ, we used the property that the integral of GP is again GP. Given a dataset $\{(\boldsymbol{x}_j, \boldsymbol{w}_j, y_j)\}_{j=1}^t$, the expectation with respect to $\boldsymbol{w}$ of $\mathcal{GP}(0, k)$ is again a GP, and its posterior distribution is given by $\mathcal{GP}(\tilde{\mu}_t(\boldsymbol{x}), \tilde{k}(\boldsymbol{x}, \boldsymbol{x}'))$, where $\tilde{\mu}_t(\boldsymbol{x})$ and $\tilde{k}(\boldsymbol{x}, \boldsymbol{x}')$ are given by

$$\tilde{\mu}_t(\boldsymbol{x}) = \left\{\sum_{\boldsymbol{w} \in \Omega} p(\boldsymbol{w})\boldsymbol{k}_t(\boldsymbol{x}, \boldsymbol{w})\right\}^{\top} (\boldsymbol{K}_t + \sigma_{\text{noise}}^2 \boldsymbol{I}_t)^{-1} \boldsymbol{y}_t,$$

$$\tilde{k}_t(\boldsymbol{x}, \boldsymbol{x}') = \left\{\sum_{\boldsymbol{w}, \boldsymbol{w}' \in \Omega} p(\boldsymbol{w})p(\boldsymbol{w}')k((\boldsymbol{x}, \boldsymbol{w}), (\boldsymbol{x}', \boldsymbol{w}'))\right\}$$

$$- \left\{\sum_{\boldsymbol{w} \in \Omega} p(\boldsymbol{w})\boldsymbol{k}_t(\boldsymbol{x}, \boldsymbol{w})\right\}^{\top} (\boldsymbol{K}_t + \sigma_{\text{noise}}^2 \boldsymbol{I}_t)^{-1} \left\{\sum_{\boldsymbol{w}' \in \Omega} p(\boldsymbol{w}')\boldsymbol{k}_t(\boldsymbol{x}', \boldsymbol{w}')\right\}.$$

Let $\hat{F}_t = \max_{\boldsymbol{x} \in \mathcal{X}} \tilde{\mu}_t(\boldsymbol{x})$ and $\tilde{\sigma}_t^2(\boldsymbol{x}) = \tilde{k}_t(\boldsymbol{x}, \boldsymbol{x})$. In BQ, $\boldsymbol{x}_t$ was selected based on the expected improvement (EI) maximization in $\mathcal{GP}(\tilde{\mu}_t(\boldsymbol{x}), \tilde{k}(\boldsymbol{x}, \boldsymbol{x}'))$ for $\hat{F}_t$. That is, for $z_{\boldsymbol{x},t} = \frac{\tilde{\mu}_t(\boldsymbol{x}) - \hat{F}_t}{\tilde{\sigma}_t(\boldsymbol{x})}$, the value of EI is calculated by $\tilde{\sigma}_t(\boldsymbol{x})\{z_{\boldsymbol{x},t}\Phi(z_{\boldsymbol{x},t}) + \phi(z_{\boldsymbol{x},t})\}$.

**BPT-UCB**  In BPT-UCB, we first defined $\eta = 0.5\min\{10^{-8}c/2, 10^{-16} \times 0.05 \times c/(8|\mathcal{X} \times \Omega|)\}$, where we used $c = 1$ in Section 5.1, and $c = 2$ in Section 5.3. Next, we defined $h_{\boldsymbol{x}, \boldsymbol{w}, t} = h + 2\eta$ if $|\mu_t(\boldsymbol{x}, \boldsymbol{w}) - h| < \eta$, and otherwise $h_{\boldsymbol{x}, \boldsymbol{w}, t} = h$. Using this, we defined $\hat{p}_t(\boldsymbol{x})$ and $\gamma_t^2(\boldsymbol{x})$ as follows:

$$\hat{p}_t(\boldsymbol{x}) = \sum_{\boldsymbol{w} \in \Omega} p(\boldsymbol{w})\Phi\left(\frac{\mu_t(\boldsymbol{x}, \boldsymbol{w}) - h_{\boldsymbol{x}, \boldsymbol{w}, t}}{\sigma_t(\boldsymbol{x}, \boldsymbol{w})}\right),$$

$$\gamma_t^2(\boldsymbol{x}) = \sum_{\boldsymbol{w} \in \Omega} p(\boldsymbol{w})\Phi\left(\frac{\mu_t(\boldsymbol{x}, \boldsymbol{w}) - h_{\boldsymbol{x}, \boldsymbol{w}, t}}{\sigma_t(\boldsymbol{x}, \boldsymbol{w})}\right)\left\{1 - \Phi\left(\frac{\mu_t(\boldsymbol{x}, \boldsymbol{w}) - h_{\boldsymbol{x}, \boldsymbol{w}, t}}{\sigma_t(\boldsymbol{x}, \boldsymbol{w})}\right)\right\}.$$

For $\beta_t = \frac{|\mathcal{X} \times \Omega| \pi^2 t^2}{3 \times 0.05}$, we defined $\text{BPTUCB}(\boldsymbol{x}) = \hat{p}_t(\boldsymbol{x}) + \beta_t^{1/10} \gamma_t^{2/10}(\boldsymbol{x})$. Then, $\boldsymbol{x}_{t+1}$ and $\boldsymbol{w}_{t+1}$ were selected by

$$\boldsymbol{x}_{t+1} = \arg\max_{\boldsymbol{x} \in \mathcal{X}} \text{BPTUCB}(\boldsymbol{x}),$$

$$\boldsymbol{w}_{t+1} = \arg\max_{\boldsymbol{w} \in \Omega} \Phi\left(\frac{\mu_t(\boldsymbol{x}_{t+1}, \boldsymbol{w}) - h_{\boldsymbol{x}_{t+1}, \boldsymbol{w}, t}}{\sigma_t(\boldsymbol{x}_{t+1}, \boldsymbol{w})}\right) \left\{1 - \Phi\left(\frac{\mu_t(\boldsymbol{x}_{t+1}, \boldsymbol{w}) - h_{\boldsymbol{x}_{t+1}, \boldsymbol{w}, t}}{\sigma_t(\boldsymbol{x}_{t+1}, \boldsymbol{w})}\right)\right\}.$$

**BPT-UCB (fixed)**  In BPT-UCB (fixed), the definition of $\text{BPTUCB}(\boldsymbol{x})$ in BPT-UCB was changed to $\text{BPTUCB}(\boldsymbol{x}) = \hat{p}_t(\boldsymbol{x}) + 3\gamma_t(\boldsymbol{x})$, and $\boldsymbol{x}_{t+1}$ and $\boldsymbol{w}_{t+1}$ were selected using the same procedure as BPT-UCB.

**BBBMOBO**  In BBBMOBO, we used $\beta_t = 2\log(|\mathcal{X} \times \Omega| \pi^2 t^2/(6 \times 0.05))$ and $\boldsymbol{x}_t = \tilde{\boldsymbol{x}}_t$, where $\tilde{\boldsymbol{x}}_t$ is given in Definition 3.1.

**BBBMOBO (fixed)**  In BBBMOBO (fixed), we used $\beta_t = 9$ and $\boldsymbol{x}_t = \tilde{\boldsymbol{x}}_t$, where $\tilde{\boldsymbol{x}}_t$ is given in Definition 3.1.

**Proposed**  In Proposed, $\boldsymbol{x}_t$ was selected by Definition 3.1.

**Proposed (fixed)**  In Proposed (fixed), $\boldsymbol{x}_t$ was selected by Definition 3.1, where we used $\beta_t = 9$.

Values of $\text{ucb}_{t-1}(\boldsymbol{x})$ and $\text{lcb}_{t-1}(\boldsymbol{x})$ in EXP, PTR and EXP-MAE were calculated by using results in Table 3 in Inatsu et al. (2024a). Here, $\text{ucb}_{t-1}(\boldsymbol{x})$ and $\text{lcb}_{t-1}(\boldsymbol{x})$ for $-\alpha\mathbb{E}[|f(\boldsymbol{x}, \boldsymbol{w}) - F_{\exp}(\boldsymbol{x})|]$ were calculated by using the results of the mean absolute deviation and monotonic Lipschitz map in the table. Combining these and the results for the weighted sum, we calculated $\text{ucb}_{t-1}(\boldsymbol{x})$ and $\text{lcb}_{t-1}(\boldsymbol{x})$ for EXP-MAE.

## D.2  Identity of BBBMOBO (fixed) and Proposed (fixed) in the Expectation Measure

In the expectation measure, $\text{ucb}_{t-1}(\boldsymbol{x})$ and $\text{lcb}_{t-1}(\boldsymbol{x})$ are calculated as follows:

$$\text{ucb}_{t-1}(\boldsymbol{x}) = \mathbb{E}_{\boldsymbol{w}}[\mu_{t-1}(\boldsymbol{x}, \boldsymbol{w}) + \beta_t^{1/2}\sigma_{t-1}(\boldsymbol{x}, \boldsymbol{w})], \ \text{lcb}_{t-1}(\boldsymbol{x}) = \mathbb{E}_{\boldsymbol{w}}[\mu_{t-1}(\boldsymbol{x}, \boldsymbol{w}) - \beta_t^{1/2}\sigma_{t-1}(\boldsymbol{x}, \boldsymbol{w})].$$

Therefore, equation 4 in Definition 3.1 is given by $\boldsymbol{x}_t = \arg\max_{\boldsymbol{x} \in \tilde{\boldsymbol{x}}_t, \hat{\boldsymbol{x}}_t} \mathbb{E}_{\boldsymbol{w}}[\sigma_{t-1}(\boldsymbol{x}, \boldsymbol{w})]$. Also, the definition of $\tilde{\boldsymbol{x}}_t$ can be rewritten as $\tilde{\boldsymbol{x}}_t = \arg\max_{\boldsymbol{x} \in \mathcal{X}} \text{ucb}_{t-1}(\boldsymbol{x})$. Here, the difference between BBBMOBO (fixed) and Proposed (fixed) is only whether to use $\boldsymbol{x}_t = \tilde{\boldsymbol{x}}_t$ or equation 4 in Definition 3.1. In Proposed (fixed), when $\boldsymbol{x}_t = \arg\max_{\boldsymbol{x} \in \tilde{\boldsymbol{x}}_t, \hat{\boldsymbol{x}}_t} \mathbb{E}_{\boldsymbol{w}}[\sigma_{t-1}(\boldsymbol{x}, \boldsymbol{w})] = \tilde{\boldsymbol{x}}_t$, it is consistent with BBBMOBO (fixed). Also, when $\boldsymbol{x}_t = \arg\max_{\boldsymbol{x} \in \tilde{\boldsymbol{x}}_t, \hat{\boldsymbol{x}}_t} \mathbb{E}_{\boldsymbol{w}}[\sigma_{t-1}(\boldsymbol{x}, \boldsymbol{w})] = \hat{\boldsymbol{x}}_t$,

$$\begin{aligned}
\text{ucb}_{t-1}(\tilde{\boldsymbol{x}}_t) &= \mathbb{E}_{\boldsymbol{w}}[\mu_{t-1}(\tilde{\boldsymbol{x}}_t, \boldsymbol{w}) + \beta_t^{1/2}\sigma_{t-1}(\tilde{\boldsymbol{x}}_t, \boldsymbol{w})] = \mathbb{E}_{\boldsymbol{w}}[\mu_{t-1}(\tilde{\boldsymbol{x}}_t, \boldsymbol{w})] + \beta_t^{1/2}\mathbb{E}_{\boldsymbol{w}}[\sigma_{t-1}(\tilde{\boldsymbol{x}}_t, \boldsymbol{w})] \\
&\leq \mathbb{E}_{\boldsymbol{w}}[\mu_{t-1}(\hat{\boldsymbol{x}}_t, \boldsymbol{w})] + \beta_t^{1/2}\mathbb{E}_{\boldsymbol{w}}[\sigma_{t-1}(\tilde{\boldsymbol{x}}_t, \boldsymbol{w})] \\
&\leq \mathbb{E}_{\boldsymbol{w}}[\mu_{t-1}(\hat{\boldsymbol{x}}_t, \boldsymbol{w})] + \beta_t^{1/2}\mathbb{E}_{\boldsymbol{w}}[\sigma_{t-1}(\hat{\boldsymbol{x}}_t, \boldsymbol{w})] = \text{ucb}_{t-1}(\hat{\boldsymbol{x}}_t).
\end{aligned}$$

Here, the first inequality follows from the definition of $\hat{\boldsymbol{x}}_t$, $\hat{\boldsymbol{x}}_t = \arg\max_{\boldsymbol{x} \in \mathcal{X}} \mathbb{E}_{\boldsymbol{w}}[\mu_{t-1}(\boldsymbol{x}, \boldsymbol{w})]$, and the second inequality is derived by the assumption, $\boldsymbol{x}_t = \arg\max_{\boldsymbol{x} \in \tilde{\boldsymbol{x}}_t, \hat{\boldsymbol{x}}_t} \mathbb{E}_{\boldsymbol{w}}[\sigma_{t-1}(\boldsymbol{x}, \boldsymbol{w})] = \hat{\boldsymbol{x}}_t$. Therefore, $\text{ucb}_{t-1}(\tilde{\boldsymbol{x}}_t) \leq \text{ucb}_{t-1}(\hat{\boldsymbol{x}}_t)$, but on the other hand, from the definition of $\tilde{\boldsymbol{x}}_t$, $\text{ucb}_{t-1}(\tilde{\boldsymbol{x}}_t) \geq \text{ucb}_{t-1}(\hat{\boldsymbol{x}}_t)$. Hence, $\text{ucb}_{t-1}(\tilde{\boldsymbol{x}}_t) = \text{ucb}_{t-1}(\hat{\boldsymbol{x}}_t)$. In the EXP experiments in Section 5, there were no cases where multiple $\boldsymbol{x}$ values maximized $\text{ucb}_{t-1}(\boldsymbol{x})$. As a result, $\tilde{\boldsymbol{x}}_t = \hat{\boldsymbol{x}}_t$, which is consistent with BBBMOBO (fixed).

