# OpenReview forum: "Bayesian Optimization of Robustness Measures under Input Uncertainty: A Randomized Gaussian Process Upper Confidence Bound Approach"
_TMLR — Accepted by TMLR_

### Review · Reviewer_Nk3H · 2025-05-17

**Summary Of Contributions:**

This paper extends the existing randomized Gaussian Process Upper Confidence Bound (GP-UCB) framework to optimize black-box functions under input uncertainty. By sampling the trade-off parameter $\beta$ from a chi-squared-based distribution, it overcomes the need for manual specification of $\beta$ required in conventional GP-UCB methods. Moreover, the proposed approach provides tight bounds on the expected regret and numerical experiments.

**Audience:**

Yes

**Broader Impact Concerns:**

I have no concerns regarding the ethical implications of this work.

**Claims And Evidence:**

Yes

**Requested Changes:**

While experiments with continuous inputs would strengthen the work, it is more critical for the paper to improve the support for (or revise) the claims discussed in the "Weaknesses" section (excluding the experiments) and to improve clarity on the points mentioned above.

**Strengths And Weaknesses:**

A major strength of this paper lies in its introduction of randomized $\beta$ into the context of optimizing robustness measures. This eliminates the need for manual specification of $\beta$, making the approach more practical and broadly applicable, which is an aspect other researchers can benefit from.

Additionally, the proposed method is applicable to a wide range of robustness measures, an advantage developed from the work of Inatsu et al. (2024a). The paper also establishes a bound on the expected regret and identifies the optimal solution for the expectation measure.

---

However, the paper would benefit from improved clarity in several points and more robust support for some of its key claims.

+ Claims:

  + The paper claims that "To the best of our knowledge, no research has been conducted on BO methods based on GP-UCB for general robustness measures that achieves a tighter regret bound without requiring the growth of $\beta_t$." However, the high-probability bound in this work is also not tigher than GP-UCB. Could the authors provide support for the above claim?

  + Is Assumption 4.1 same as the one in Inatsu et al. (2024a), as claimed? If I am not mistaken, Inatsu et al. (2024a) only require the function $q$ to be strictly increasing. If the assumption here (defined by the set $Q$) differs, probably the author needs to provide justification.

  + While the paper claims to support continuous inputs (theoretically justified in Appendix A), the experiments are limited to discrete input spaces. Including empirical validation for continuous inputs would strengthen the experiments.

  + The claim "To the best of our knowledge, this study is the first to establish regret bounds for various robustness measures." may not be entirely accurate as Inatsu et al. (2024a) also provide such bounds.

+ Clarity:

  + What is the symbol $|^{\forall}$ in the definition of $G_{t-1}(x)$?

  + $\rho(f(x,w)) = F(x)$: $F(x)$ does not depend on $w$, but $\rho(f(x,w))$ depend on $w$?

  + $G_{t-1}(x)$ is a set of functions over $w$ but it is defined as $\\{g(x,w) \| \dots\\}$ (elements are scalars?)

  + Instead of referring readers to Inatsu et al. (2024a), the paper should explicitly define the bounds $lcb_{t-1}(x)$, $ucb_{t-1}(x)$, and $q(a)$ for all applicable robustness measures. This would enhance clarity and make the paper more self-contained.

Additionally, the paper mentions several limitations that may reduce its appeal relative to existing methods.

  + Computing $\hat{t}$ (the index of the optimal solution) is intractable. Does this imply that the guarantee on the expected value of the instantaneous regret is not practically meaningful?

  + The high-probability bound appears to be loose due to the presence of $\delta^{-1}$ compared to other GP-UCB-based bounds.

---

> ### Author Response · Authors · 2025-05-25
> **Answer to Review Comments (1/2)**
>
> We thank the reviewer for the constructive suggestions and comments.
>
> **Answer to Weakness #1**
> ```
> The paper claims that "To the best of our knowledge, no research has been conducted on BO methods based on GP-UCB for general robustness measures that achieves a tighter regret bound without requiring the growth of \beta_t." However, the high-probability bound in this work is also not tighter than GP-UCB. Could the authors provide support for the above claim?
> ```
> The intention of the last part in Section 1.1 is that, to the best of our knowledge, there is no existing method that is i) applicable to general robustness measures, ii) does not require an increase in the parameter $\beta_t$, and iii) provides a tight regret bound based on GP-UCB. At the end of Section 4, when comparing only the $\log (\delta ^{-1})$ term appearing in the upper bound of the theoretical analysis results for the standard GP-UCB with the $\delta^{-1}$ term in the proposed method, the proposed method is less tight than GP-UCB. However, the standard GP-UCB is not designed for robustness measures and additionally requires an increase in $\beta_t$. Additionally, the method proposed by Inatsu et al. (2024a) also requires an increase in $\beta_t$, while other methods are either specialized for specific robustness measures or do not provide theoretical analysis.
>
> **Answer to Weakness #2**
> ```
> Is Assumption 4.1 same as the one in Inatsu et al. (2024a), as claimed? If I am not mistaken, Inatsu et al. (2024a) only require the function q to be strictly increasing. If the assumption here (defined by the set Q) differs, probably the author needs to provide justification.
> ```
> As you pointed out, $q(a)$ used in Inatsu et al. (2024a) assumes only monotonic increase, whereas $q(a)$ used in this study additionally assumes that it is a sum of concave functions. Therefore, q(a) in this study requires stronger assumptions than q(a) in Inatsu et al. (2024a), and they are different. The reason for making such assumptions in this study is due to differences in the method of evaluating regret. Inatsu et al. (2024a) deals with high-probability bounds, where there is no need to consider the expectation, whereas in this paper, it is necessary to evaluate the expectation for the regret. As a result, there are parts where Jensen's inequality must be used (e.g., Equation 16), and therefore, additional assumptions regarding concave functions are imposed.
>
> **Answer to Weakness #3**
> ```
> While the paper claims to support continuous inputs (theoretically justified in Appendix A), the experiments are limited to discrete input spaces. Including empirical validation for continuous inputs would strengthen the experiments.
> ```
> As you pointed out, we agree that conducting experiments in continuous space can enhance our understanding of practical performance in continuous space.
> However, the main claim of this paper is to propose a method with theoretical guarantees that does not require an increase in $\beta_t$ or hyperparameter tuning. In this sense, we recognize that the continuous extension is merely supplementary.
> In fact, for example, in Appendix A.1, we perform a continuous extension, but this requires additional assumptions and changes to the definition of $\beta_t$.
> As a result, according to Theorem A.1, $\beta_t$ must be increased with respect to $t$, and additionally, the parameters $a_i$ and $b_i$ based on Assumption A.1 are required.
> These $a_i$ and $b_i$ are constants determined by the Gaussian process that $f$ follows, but they are generally difficult to calculate analytically.
> Ultimately, in the continuous setting, it is difficult to compare the method under theoretical guarantees with other methods, as was done in the numerical experiments in the main text.
>
> **Answer to Weakness #4**
> ```
> The claim "To the best of our knowledge, this study is the first to establish regret bounds for various robustness measures." may not be entirely accurate as Inatsu et al. (2024a) also provide such bounds.
> ```
> This description referred to expected regret and expected cumulative regret. Therefore, in the revised version, we will change the description to be more accurate.
>
> **Answer to Weakness #5**
> ```
> What is the symbol |^\foral in the definition of G_{t-1} (x)?
> ```
> The symbol $^\forall$ means “for all.”
> The meaning of $g(x,w)$ in $G_{t-1}(x)$ is a function with respect to $w$ when $x$ is fixed, i.e., $g(x,\cdot)$.
> Therefore, the current notation
>
> $G_{t-1} (x) = ${$g(x,w)|^\forall w \in \Omega, g(x,w) \in Q_t (x,w)$}
>
> should be written as
>
>  $G_{t-1} (x) = ${$g(x,\cdot)|^\forall w \in \Omega, g(x,w) \in Q_t (x,w)$}.
>
> This means the set of all functions $g(x,\cdot)$ such that $g(x,w) \in Q_t (x,w)$ for all $w \in \Omega$.

---

> ### Author Response · Authors · 2025-05-25
> **Answer to Review Comments (2/2)**
>
> **Answer to Weakness #6**
> ```
> \rho(f(x,w)) = F(x): F(x) does not depend on w, but \rho(f(x,w)) depend on w?
> ```
> In this definition, $f(x,w)$ is a function of $w$ with fixed $x$, i.e., $f(x,\cdot)$.
> Therefore, it should have been written as $F(x) = \rho (f(x,\cdot))$.
>
> **Answer to Weakness #7**
> ```
> G_{t-1}(x) is a set of functions over w, but it is defined as {g(x,w)|...} (elements are scalars?)
> ```
> See Answer to Weakness #5.
>
> **Answer to Weakness #8**
> ```
> Instead of referring readers to Inatsu et al. (2024a), the paper should explicitly define the bounds lcb_{t-1} (x),ucb_{t-1}(x),and q(a) for all applicable robustness measures. This would enhance clarity and make the paper more self-contained.
> ```
> In the revised manuscript, we will describe $lcb_{t-1} (x)$, $ucb_{t-1} (x)$ and $q(a)$ for each robustness measure using tables such as Table 3 and 4 of Inatsu et al. (2024a).
>
> **Answer to Weakness #9**
> ```
> Computing \hat{t} (the index of the optimal solution) is intractable. Does this imply that the guarantee on the expected value of the instantaneous regret is not practically meaningful?
> ```
> We believe that it is meaningful to evaluate the expected value of instantaneous regret.
> First, it was not obvious whether there was a way to choose an index such that $E[r_{t^\prime} ]$ converged to 0, but in this paper, we showed that using $\hat{t}$ guarantees that $E[r_{\hat{t}} ]$ converges to 0, which is significant.
> Furthermore, just before Theorem 4.3, we stated that it is possible to estimate $\hat{t}$ by sampling from the posterior distribution of $f$. If the number of samples is sufficiently large, we can expect that the deviation between the approximated $\hat{t}$ and the true $\hat{t}$ will be small, so it is also meaningful to have a means of obtaining an approximate $\hat{t}$ is available.
> Finally, we must emphasize that in the numerical experiments, we evaluate $r_t$, not $r_{\hat{t}}$.
> From Figure 2, we can see that even for PTR and EXP-MAE, where $\hat{t}=t$ does not necessarily hold, $r_t$ converges to 0.
> From these results, we can conclude that while theoretically $E[r_{\hat{t}}]$ is required, in practice $E[r_t]$ is sufficient.
> Clarifying whether $\hat{t}=t$ is acceptable for general robustness measures is an important direction for future research.
>
> **Answer to Weakness #10**
> ```
> The high-probability bound appears to be loose due to the presence of \delta^{-1} compared to other GP-UCB-based bounds.
> ```
> When comparing high-probability bounds based on existing GP-UCB-based methods and the proposed method, focusing only on $\delta$, the proposed method is indeed loose.
> On the other hand, there are differences between the high-probability bounds of each method and those of the proposed method, in addition to $\delta$. Therefore, depending on what is considered a variable, the high-probability bound of the proposed method can still be considered tight.
> Here, let $R_1$ denote the cumulative regret based on the standard GP-UCB under the assumption that $\mathcal{X} \times \Omega \subset \mathbb{R}^d$ and $w \in \Omega$ is not an environmental variable. Let $R_2$ denote the cumulative regret under Inatsu et al. (2024a) when $w$ is an environmental variable and the expectation measure is considered as the robustness measure. Furthermore, let $R_3$ denote the cumulative regret in this study. When comparing these, the following holds with probability at least $1-\delta$:
>
> $R_1 \leq O(\sqrt{T \beta_T \gamma_T}),$
>
> $R_2 \leq O(\sqrt{T \beta_T \gamma_T}),$
>
> $R_3 \leq O (\delta^{-1} \sqrt{T \gamma_T}).$
>
> When using the Gaussian kernel as the kernel function, according to Theorem 5 in Srinivas et al. (2010), $\gamma_t = O((\log t)^{d+1})$.
> Furthermore, according to Theorem 1 in Srinivas et al. (2010) and Theorem 4.1 in Inatsu et al. (2024a), $\beta_t$ is $O(\log t+ \log(\delta^{-1}))$ and $O( (\log t)^{d+1} + \log (\delta^{-1}))$, respectively.
> Therefore, the orders of $R_1$, $R_2$, and $R_3$ are as follows:
>
> $R_1 = O(\sqrt{T ( \log T + \log (\delta^{-1}) ) (\log T)^{d+1} }),$
>
> $R_2 = O(\sqrt{T ( (\log T)^{d+1} + \log (\delta^{-1}) ) (\log T)^{d+1} }),$
>
> $R_3 = O (\delta^{-1} \sqrt{T (\log T)^{d+1}}).$
>
> Since $\delta$ is a user-specified pre-determined probability, if $\delta$ is considered a constant, the order of $R_3$ in the proposed method is tighter than the other two.
>
> **Answer to Requested Changes**
> ```
> While experiments with continuous inputs would strengthen the work, it is more critical for the paper to improve the support for (or revise) the claims discussed in the "Weaknesses" section (excluding the experiments) and to improve clarity on the points mentioned above.
> ```
> In the revised version, we will correct unclear notations and explanations based on the answers to reviewer comments.

---

> > ### Comment · Reviewer_Nk3H · 2025-05-27
> >
> > Thank you for providing the detailed responses. They address my concerns. While the high-probability bound dependence on  remains a limitation (which is acknowledged in the paper), this drawback is outweighed by the other contributions presented in this work.

---

### Review · Reviewer_HJS8 · 2025-05-20

**Summary Of Contributions:**

The paper focuses on the robust Bayesian optimzation setting, where an additional environment variable $w$ is present and can be selected. The authors propose a Gaussian process upper confidence bound-based method, where the trade-off parameter $\beta_t$ is sampled from a two-parameter exponential distribution, following IRGP-ICB, rather than increasing by $\log t$ every iteration as in GP-UCB. They show that this leads to tight bounds on expected cumulative regret bounds across various robustness measures. Experimental results on both synthetic and real-world datasets were also shown.

**Audience:**

Yes

**Claims And Evidence:**

Yes

**Requested Changes:**

Can the authors clarify the following:

* $\tilde{x}_t$ and $x_t$ are defined in both Lemma 3.1 and Definition 3.1 in slightly different ways. Why are there multiple definitions? Lemma 3.1 does not seem to be used as well.

* On the information gain defined at the end of page 7. Why is it defined with $(\tilde{x}_t, \tilde{w}_t)$, instead of say $(x_t, w_t)$? Also, where is $\tilde{w}_t$ defined?

* In Equation 8, why define $\hat{t}$ using $F(x^*)$, which is unknown? Theorem 4.2 and 4.4 are also based on Equation 8. What about only using the alternate definition of $\hat{t}$ that does not depend on $x^*$, already defined in the paper, for theoretical analysis?

Minor:

* Correct capitalization in bibliography for "gaussian", "bayesian", etc

**Strengths And Weaknesses:**

**Strengths:**

* Generally applicable to common robustness measures
* Tighter expected theoretical bounds across common robustness measures
* Detailed appendix extending to various settings, including continuous domains and uncontrollable $w$

**Weaknesses:**
* Experimental results do not indicate that the proposed method is significantly better than existing methods empirically, which is acknowledged by the authors

---

> ### Author Response · Authors · 2025-05-24
> **Answer to Review Comments**
>
> We thank the reviewer for the constructive suggestions and comments.
>
> **Answer to Weakness:**
> ```
> Experimental results do not indicate that the proposed method is significantly better than existing methods empirically, which is acknowledged by the authors.
> ```
> As you pointed out, the proposed method does not dramatically improve the performance compared with existing methods, and we acknowledge this. However, as stated in the second paragraph in Section 6, the proposed method is not specialized for specific robustness measures and is superior in that it can be used in more general settings. Furthermore, compared to Inatsu et al. (2024a), which provides similar theoretical guarantees, our method is more user-friendly as it does not require any hyperparameters in acquisition functions that need to be adjusted. In summary, while the empirical performance does not show a dramatic improvement, we believe that our method is significant in that it achieves performance equivalent to or better than existing methods under a wider range of settings and improves usability.
>
> **Answer to Requested Change #1**
> ```
> $\tilde{x}_t$ and $x_t$ are defined in both Lemma 3.1 and Definition 3.1 in slightly different ways. Why are there multiple definitions? Lemma 3.1 does not seem to be used as well.
> ```
> We apologize for the confusion, but $\tilde{x} _t$ and $x_t$ in Lemma 3.1 are different from $\tilde{x}_t$ and $x_t$ in Definition 3.1.
> In Lemma 3.1, we claim that in the standard Bayesian optimization framework without environmental variables, the standard GP-UCB maximization point $x^{(u)}_t$ can be expressed using $\tilde{x}_t$ and $x_t$.
> By extracting this idea, we can apply the same reasoning to $x_t$ in the setting of this paper, where environmental variables exist, which is the intention of Definition 3.1.
> Therefore, to avoid confusion, in the revised manuscript, we change $\tilde{x}_t, \check{x}_t$, and $x_t$ in Lemma 3.1 to $\tilde{x}^{(f)}_t, \check{x}^{(f)}_t$, and $x^{(f)}_t$, respectively, and emphasize that they are different from $\tilde{x}_t$ and $x_t$ in Definition 3.1.
>
> **Answer to Requested Change #2**
> ```
> On the information gain defined at the end of page 7. Why is it defined with $(\tilde{x}_t,\tilde{w}_t)$, instead of say $(x_t,w_t)$?  Also, where is $\tilde{w}_t$ defined?
> ```
> For each $t$, the maximum information gain $\gamma_t$ is expressed as follows:
> $$
> \gamma_t = 1/2 \sup_{A \subset \mathcal{X} \times \Omega} \log {\rm det} (I_t + \sigma^{-2}_{\rm noise} K_A),
> $$
> where $A =$ {$a₁, \ldots, a_t $}  is an arbitrary finite subset of $\mathcal{X} \times \Omega$, and the the $(i, j)$-th component of $K_A$  is $k(a_i, a_j)$.
> Therefore, $\tilde{w}_t$ denotes an arbitrary element. The reason for not using $(x_t,w_t)$ and $K_t$ is that $x_t$ and $w_t$ are meaningful elements selected by the acquisition function, not arbitrary elements, and $K_t$ is calculated using $(x_t,w_t)$, so the current notation is used for differentiation.
> However, in the current manuscript, $\gamma_t$ is defined using $\tilde{x}_t$, but this $\tilde{x}_t$ is different from the one used in Definition 3.1. To avoid confusion, a different symbol will be used in the revised manuscript.
>
> **Answer to Requested Change #3**
> ```
> In Equation 8, why define $\hat{t}$ using $F(x^)$, which is unknown? Theorem 4.2 and 4.4 are also based on Equation 8. What about only using the alternate definition of $\hat{t}$ that does not depend on $x^$, already defined in the paper, for theoretical analysis?
> ```
> As stated five lines above Theorem 4.3, $\hat{t}$ can also be expressed as
> $$
> \hat{t} = {\rm argmax} _ {1 \leq i \leq t} E_{t-1} [F(\hat{x}_i )].
> $$
> Therefore, it is possible to define $\hat{t}$ without using the unknown $x^\ast$. The reason for defining it in the form of Equation 8 is that it simplifies the inequality evaluations when performing theoretical analysis. In fact, in Appendix C.3, we have to show the following inequality:
> $$
> E _{t-1} [F(x^\ast) - F(x _ {\hat{t}} )] \leq
> E _{t-1} [F(x^\ast) - F(x_i )].
> $$
> By defining $\hat{t}$ as in Equation 8, this is clearly true. On the other hand, even in Equation 8, the unknown function $F$ is used, and ultimately, it is generally difficult to analytically determine $\hat{t}$. However, Theorem 4.3 shows that $\hat{t}=t$ is acceptable in the case of the expectation measure. We will add these explanations in the revised version.
>
> **Answer to Minor Comments**
> ```
> Correct capitalization in bibliography for "gaussian", "bayesian", etc
> ```
> In the revised manuscript, we will appropriately correct the incorrect descriptions in the references.

---

### Review · Reviewer_DyNJ · 2025-06-09

**Summary Of Contributions:**

This paper proposed a new Bayesian optimization (BO) algorithm, called RRGP-UCB, for optimizing functions with input uncertainty. Instead of using deterministic trade-off parameter $\beta_t$ , RRGP-UCB randomly samples $\beta_t$ from a chi-squared distribution such that $\beta_t$ will not keep increasing with $t$ as in the conventional GP-UCB. In contrast to some existing methods designed for handling only specific robustness measures, RRGP-UCB is applicable to any robustness measure (e.g., expectation, worst-case, value-at-risk, etc.) that satisfies certain conditions. Both theoretical and empirical results are provided to show the performance of the proposed method.

**Audience:**

Yes

**Broader Impact Concerns:**

There is no ethical concern.

**Claims And Evidence:**

No

**Requested Changes:**

1. The author(s) discussed only the setting with discrete input space in the main paper and put all the other results in the appendix. However, since the robustness of the proposed method is one contribution as claimed in the introduction, I suggest the authors to consider put some important results (both theoretical and empirical) of the other settings (e.g., continuous input spaces) in the main paper such that the contributions of the proposed method can be identified more easily.

2. The notations in this paper are complex, especially those for various $x$. It's better to provide a summary of the notation such that the paper is easier to follow. Also, there exist some confusing notations or typos. For example,

* Is $\tilde{x}$ in Lemma 3.1 and that in the definition of $\gamma_t$ the same? If yes, what is $\tilde{w}$ in $\gamma_t$? Otherwise, it's better to use other notations in the definition of $\gamma_t$

* In the equation below (4), should it be ucb_t or ucb_{t-1}?

* In (2), how are the lower bound lcb() and upper bound ucb() computed? Inatsu et al. (2024a) has provided several methods for approximating them. Which is used in this work?

* Lemma 3.1 is proved using specific definitions of $u_{t-1}$ and $l_{t-1}$. Does it still hold when replacing $u_{t-1}$ and $l_{t-1}$ with ucb(x) and lcb(x), respectively? If yes, why not use ucb(x) and lcb(x) directly in Lemma 3.1? Otherwise, what's the relationship between Lemma 3.1 and Definition 3.1?

3. In Figs. 2&3, why are the results of BBBMOBO (fixed) missing in all the EXP graphs?

4. In the experiment discussion, it is claimed that "small trade-off parameters limit exploration, often resulting in convergence to local optima". How is it demonstrated in the empirical results?

**Strengths And Weaknesses:**

**Strengths:**

* The paper has provided clear motivations for introducing the randomized trade-off parameter into GP-UCB with robustness measures. The related works are carefully discussed such that the contributions of the proposed method are clear.

* The proposed RRGP-UCB is general enough to handle various robustness optimization settings such as controllable environmental settings, uncontrollable settings, finite input spaces, and continuous input spaces. The algorithm design and theoretical results are discussed for each setting.

**Weakness:**

* Even though strong theoretical results are provided for various robustness settings, the experiments didn't demonstrate significant empirical results. In many cases (as shown in Figs. 2&3), fixed $\beta_t$ can achieve lower or comparable regret than using the designed randomized policy even without tuning in various settings and different methods ($\beta_t=9$ is used for both BBBMOBO (fix) and Proposed (fix) in all the experiments). Since the major contribution of this paper is to extend the randomized trade-off parameter to BO with input uncertainty, such empirical results decreased the significance of the proposed method. In addition, the algorithms designed for continuous input spaces are not tested in the experiments.

* The writing of this paper needs to be improved. The complex notations and some confusing claims make it hard to follow. See detailed comments below.

---

> ### Author Response · Authors · 2025-06-18
> **Answer to Review Comments (1/3)**
>
> We thank the reviewer for the constructive suggestions and comments.
>
> **Answer to Weakness #1**
> ```
> Even though strong theoretical results are provided for various robustness settings, the experiments didn't demonstrate significant empirical results. In many cases (as shown in Figs. 2&3), fixed \beta_t can achieve lower or comparable regret than using the designed randomized policy even without tuning in various settings and different methods (\beta_t = 9 is used for both BBBMOBO (fix) and Proposed (fix) in all the experiments). Since the major contribution of this paper is to extend the randomized trade-off parameter to BO with input uncertainty, such empirical results decreased the significance of the proposed method. In addition, the algorithms designed for continuous input spaces are not tested in the experiments.
> ```
>
> The main contribution of this paper is that by introducing randomization, it is no longer necessary to adjust the trade-off parameter $\beta_t$ or to increase it with time $t$, and additionally, theoretically valid bounds can be derived. Although the proposed method has not been confirmed to significantly outperform other methods in experiments, it is meaningful that the proposed method, which has theoretical validity, achieves performance comparable to that of practically useful ad hoc methods (BBBMOBO (fixed) and Proposed (fixed)) that lack theoretical validity. Furthermore, we believe that the fact that parameter adjustment is not required is also highly significant. In addition, in the EXP experiment with 4D synthetic functions, both BBBMOBO (fixed) and Proposed (fixed) were stuck in local solutions (BBBMOBO (fixed) and Proposed (fixed) are the same in this setting, and see also the answer to Requested Changes #3). This is thought to be caused by the small value of $\beta_t$, but in the proposed method with randomization, large $\beta_t$ values can appear, and as a result, Proposed avoids getting stuck in local optima. We believe that this behavior demonstrates the practical usefulness of introducing randomization.
>
> Furthermore, as mentioned in the answer to the Requested Changes #1, we recognize that extension to continuous spaces is possible but supplementary. The reason is that when extending to continuous spaces, it is ultimately necessary to increase $\beta_t$ according to time, and parameter estimation is also required, which negates the advantages of the proposed method. Therefore, in the continuous setting, it is difficult to compare the proposed method, which incorporates theoretical guarantees through randomization, with the ad hoc approach. For these reasons, the extension to continuous spaces is described in the appendix, and experiments could not carried out.
>
> **Answer to Requested Changes #1**
> ```
> The author(s) discussed only the setting with discrete input space in the main paper and put all the other results in the appendix. However, since the robustness of the proposed method is one contribution as claimed in the introduction, I suggest the authors to consider put some important results (both theoretical and empirical) of the other settings (e.g., continuous input spaces) in the main paper such that the contributions of the proposed method can be identified more easily.
> ```
> Thank you for your important comment. We also agree that the extensibility to other settings is one of the contributions of the proposed method. However, we believe that the main contribution of this paper is that it is not necessary to adjust the trade-off parameter $\beta_t$ in the acquisition function, that it is not necessary to increase it according to time $t$, and that we were able to derive a theoretically valid bound. In this sense, we consider that the extension to continuous space is possible as a result, but its contribution is supplementary. In fact, in Theorem A.1, $\beta_t$ must be increased as $t$ increases. Furthermore, the theoretical parameters in this case require constants $a_1$ and $b_1$ based on the continuity assumption Assumption A.1. Although these values are constants determined based on the kernel, it is difficult to know their actual values, and ultimately, estimation or ad hoc adjustments are required. Therefore, the main contribution of this method, which was derived in the finite setting, namely that $\beta_t$ does not need to be adjusted and does not need to be increased over time, is lost. On the other hand, in the extension to the uncontrollable setting where both $\mathcal{X}$ and $\Omega$ are finite, the fact that $\beta_t$ does not need to be adjusted and does not need to be increased over time is still preserved. We would like to move the results of this setting from the appendix to the main text. Regarding the extension to continuous spaces, we will state in Section 4 that it is possible in principle, but also explain that there is an issue where $\beta_t$ requires adjustment and needs to be scaled with time.

---

> ### Author Response · Authors · 2025-06-18
> **Answer to Review Comments (2/3)**
>
> **Answer to Requested Changes #2**
> ```
> The notations in this paper are complex, especially those for various $x$. It's better to provide a summary of the notation such that the paper is easier to follow. Also, there exist some confusing notations or typos. For example,
> ```
> In the revised manuscript, we plan to add a table summarizing particularly important notations used in the main text in Section 2.
>
> ```
> Is $\tilde{x}$ in Lemma 3.1 and that in the definition of $\gamma_t$ the same? If yes, what is $\tilde{w}$ in $\gamma_t$? Otherwise, it's better to use other notations in the definition of $\gamma_t$.
> ```
> We apologize for the confusing typo. In the revised version, we will change it to the correct notations. The $\tilde{x}_t$ in Lemma 3.1 and $\tilde{x}_t$ used in the definition of $\gamma_t$ are different, and $\tilde{x}_t$ and $\tilde{w}_t$ in the definition of $\gamma_t$ are just variables, so we will change them to different notations to avoid confusion.
>
> ```
> In the equation below (4), should it be ucb_t or ucb_{t-1}?
> ```
> As you pointed out, $ucb_t (x)$ and $lcb_t (x)$ in the definition of $\tilde{x}_ t$ in Definition 3.1 should be $ucb_{t-1} (x)$ and $lcb_{t-1} (x)$, respectively.
>
> ```
> In (2), how are the lower bound lcb() and upper bound ucb() computed? Inatsu et al. (2024a) has provided several methods for approximating them. Which is used in this work?
> ```
> Regarding the robustness measures listed in Table 3 of Inatsu et al. (2024a), we recommend calculating the lower bound $lcb_{t-1} (x)$ and upper bound $ucb_{t-1} (x)$ using the calculation method listed in Table 3. For other measures, we recommend generating $S$ sample paths $f^{(1)} (x,w), \ldots, f^{(S)} (x,w)$ from the posterior distribution of the function $f(x,w)$ and estimating $lcb_{t-1} (x)$ and $ucb_{t-1} (x)$ as follows:
> $$
> lcb_{t-1} (x) = \min_{1 \leq s \leq S} \rho (f^{(s)} (x,\cdot)),
> $$
> $$
> ucb_{t-1} (x) = \max_{1 \leq s \leq S} \rho (f^{(s)} (x,\cdot)),
> $$
> where $\rho(f(x,\cdot))$ denotes the robustness measure at point $x$ (we will change the notation $\rho(f(x,w))$ to $\rho(f(x,\cdot))$ for avoiding confusion). In the revised manuscript, we plan to add these explanations.
>
> ```
> Lemma 3.1 is proved using specific definitions of u_{t-1} and l_{t-1}. Does it still hold when replacing u_{t-1} and l_{t-1} with ucb(x) and lcb(x), respectively? If yes, why not use ucb(x) and lcb(x) directly in Lemma 3.1? Otherwise, what's the relationship between Lemma 3.1 and Definition 3.1?
> ```
> As explained in the answer to Requested Changes #3, in the case of EXP, $ucb_{t-1} ( \tilde{x}_ t)=ucb_{t-1}(x_t)$ holds, and the same claim as in Lemma 3.1 holds. On the other hand, in general robustness measures, $ucb_{t-1}( \tilde{x}_ t)=ucb_{t-1}(x_t)$ does not necessarily hold. In fact, this is the reason why BBBMOBO (fixed) and Proposed (fixed) exhibit different behaviors except for EXP. The reason for introducing Lemma 3.1 is to provide a different perspective on Definition 3.1. As a result of carefully investigating the theoretical validity in the robust Bayesian optimization, the acquisition function in Definition 3.1 was derived. This definition does not seem to be related to the usual GP-UCB at first glance. However, the usual GP-UCB can be expressed in a different form as in Lemma 3.1, and since this equivalent expression is similar to Definition 3.1, the aim was to explain that the proposed acquisition function can also be interpreted as a GP-UCB-based acquisition function. Nevertheless, as mentioned earlier, it is not necessarily the case that $ucb_{t-1}( \tilde{x}_ t) = ucb_{t-1}(x_t)$ for all robustness measures. Therefore, the acquisition function in Definition 3.1 is not a complete equivalent representation of UCB in robust Bayesian optimization. These explanations will be added in the revised manuscript.

---

> ### Author Response · Authors · 2025-06-18
> **Answer to Review Comments (3/3)**
>
> **Answer to Requested Changes #3**
> ```
> In Figs. 2&3, why are the results of BBBMOBO (fixed) missing in all the EXP graphs?
> ```
> We apologize for the confusion regarding the experimental results. In Figures 2 and 3, when using the expectation measure (EXP), BBBMOBO (fixed) and Proposed (fixed) are the same, so they completely overlap. In EXP, $ucb_{t-1} (x)$ and $lcb_{t-1} (x)$ were calculated as follows:
> $$
> ucb_{t-1} (x) = E_w [\mu_{t-1}(x,w)+3\sigma_{t-1}(x,w)],
> $$
> $$
> lcb_{t-1} (x) = E_w [\mu_{t-1}(x,w)-3\sigma_{t-1}(x,w)].
> $$
>
> Therefore, (4) in Definition 3.1 is given by $x_t =  argmax_ {x \in \tilde{x}_ t, \hat{x}_ t} E_w [\sigma_ {t-1} (x,w)]$.
>  Also, the definition of $\tilde{x}_ t$ can be rewritten as $\tilde{x}_ t = argmax_ {x \in \mathcal{X} } ucb_ {t-1} (x)$ (the current manuscript has $ucb_ t (x)$, but $ucb_ {t-1} (x)$ is correct). Here, the difference between BBBMOBO (fixed) and Proposed (fixed) is only whether to use $x_ t = \tilde{x}_ t$ or (4) in Definition 3.1. In Proposed (fixed), when $x_ t = argmax_ {x \in \tilde{x}_ t,\hat{x}_ t} E_w [\sigma_ {t-1} (x,w)] = \tilde{x}_ t$, it is consistent with BBBMOBO (fixed). Also, when $x_ t = argmax_ {x \in \tilde{x}_ t,\hat{x}_ t} E_w [\sigma_ {t-1}(x,w)] = \hat{x}_ t$,
> $$
> ucb_ {t-1} (\tilde{x}_ t) = E_w [\mu_{t-1} (\tilde{x}_ t,w) + 3 \sigma_ {t-1} (\tilde{x}_ t,w)]
> =E_w [\mu_{t-1} (\tilde{x}_ t,w)] + 3 E_w [ \sigma_{t-1} (\tilde{x}_ t,w) ]
> $$
> $$
> \leq E_w [\mu_{t-1} (\hat{x}_ t,w)] +3 E_w [ \sigma_{t-1} (\tilde{x}_ t,w)] \leq E_w [\mu_{t-1} (\hat{x}_ t,w)] +3 E_w[ \sigma_{t-1} (\hat{x}_ t,w)] = ucb_{t-1} (\hat{x}_ t).
> $$
> Here, the first inequality follows from the definition of $\hat{x}_ t$, $\hat{x}_ t = argmax_ {x \in \mathcal{X} } E_w [\mu_{t-1} (x,w)]$, and the second inequality is derived by the assumption, $x_ t = argmax_ {x \in \tilde{x}_ t,\hat{x}_ t} E_w [\sigma_ {t-1} (x,w)] = \hat{x}_ t$. Therefore, $ucb_ {t-1} (\tilde{x}_ t) \leq ucb_ {t-1} (\hat{x}_ t)$, but on the other hand, from the definition of $\tilde{x}_ t$, $ucb_ {t-1} (\tilde{x}_ t) \geq ucb_ {t-1} (\hat{x}_ t)$. Therefore, $ucb_ {t-1} (\tilde{x}_ t) = ucb_ {t-1} (\hat{x}_ t)$. In the experiment, there were no cases where multiple $x$ values maximized $ucb_ {t-1} (x)$. As a result, $\tilde{x}_ t = \hat{x}_ t$, which is consistent with BBBMOBO (fixed). In the revised version, we will add the explanation in Section 5 that BBBMOBO (fixed) and Proposed (fixed) are consistent in EXP, and include its proof in Appendix D.
>
> **Answer to Requested Changes #4**
> ```
> In the experiment discussion, it is claimed that "small trade-off parameters limit exploration, often resulting in convergence to local optima". How is it demonstrated in the empirical results?
> ```
> For example, in the context of standard BO, consider a black-box function optimization with no input uncertainty and only one-dimensional input $x$. Let the black-box function $f(x)$ be given by
> $$
> f(x) = 1.5 \exp(-(x+2.5)^2) + 4 \exp(-(x-2.5)^2).
> $$
> That is, $f(x)$ has local maxima near $x = -2.5$ and $2.5$, and the global maximum is near $x = 2.5$. Suppose that the function value at the initial value $x = -1$ is given, and the GP regression is performed using the kernel function $k(x, x^\prime ) = \exp(-(x - x^\prime )^2)$, followed by BO based on UCB maximization. That is, $x_t = argmax_ {x \in \mathcal{X} } (\mu_{t-1} (x) + \beta^{1/2} \sigma_ {t-1}(x))$. Here, when $\beta^{1/2}$ is small (e.g., $\beta^ {1/2}=1$), repeating UCB maximization results in the UCB in the negative region of $x$ being at most about 1.5, while the maximum UCB in the positive region of $x$ is at most about 1, so the positive region is not explored, resulting in getting stuck in a local optimum. On the other hand, when $\beta^{1/2} = 2$, the UCB in the positive region of $x$ may exceed the maximum UCB in the negative region of $x$, resulting in exploration of the positive region of $x$ and the discovery of the global maximum solution. We uploaded the zip file as the Supplementary Material, and it contains the folder (BO_demo) with figures showing the behavior up to 10 iterations for $\beta^{1/2} =1$ and 2.
>
> In Srinivas et al. (2010), they also compared BO using the maximization of the posterior mean, i.e., $\beta_t =0$, and as shown in Srinivas et al. (2010), Figure 5, (a)-(c) Mean Only, getting stuck in a local optimal is confirmed.
> Furthermore, in BO with input uncertainty, Inatsu et al. (2024a) investigated the impact of a fixed $\beta$ when using EXP as the robustness measure. In the lower three panels of Figure 5 in Inatsu et al. (2024a), it is shown that using a small $\beta$ results in behavior where the optimal solution cannot be fully discovered.
> In the revised manuscript, we will add an explanation of how small trade-off parameters can cause convergence to local optima, referencing the experimental results from the existing paper.

---

### Author Response · Authors · 2025-06-22
**Revised Manuscript**

We submitted the revised manuscript that appropriately incorporated reviewers' comments. Revisions in the manuscript are written in red except for the references. The changes are as follows:

[1] Notation for the function of $w$ when $x$ is fixed (Reviewer Nk3H). On page 4, lines 27-28, we added the explanation that for any function $\varphi(x,w)$ on $\mathcal{X} \times \Omega$, when $x$ is fixed, it becomes a function with respect to $w$, and is subsequently denoted as $\varphi(x,\cdot)$. In the revised manuscript, functions defined as functions of $w$ are explicitly denoted, for example, as $f(x,\cdot), \mu_{t-1}(x,\cdot)$.

[2] Clarification of the domain of robustness measures. On page 4, line 27, we clarified that the robustness measure $\rho (\cdot)$ is a mapping from functions of $w$ to real numbers.

[3] Summary of notation (Reviewer DyNJ). On page 6, we summarized particularly important notations to Table 3.

[4] Revision of the definition of $G_{t-1} (x)$ (Reviewer Nk3H). On page 6, last line, we changed the notation $^\forall$ in the definition of $G_{t-1}(x)$ to “for any” without using symbols.

[5] Explicit notation of ucb$ _  {t-1} (x)$, lcb$ _ {t-1} (x)$ and $q(a)$ (Reviewer DyNJ, Nk3H). On pages 7, lines 9–16, and 9, lines 29–39, we explained that ucb$ _  {t-1} (x)$, lcb$ _ {t-1} (x)$ and $q(a)$ shown in Inatsu et al. (2024a) are also applicable in this study, and summarize their results in Tables 4 and 5.

[6] Mention of cases where ucb$ _ {t-1} (x)$ and lcb$ _ {t-1} (x)$ cannot be calculated (Reviewer DyNJ). On page 7, lines 16–20, we  added the explanation of the approximation method for cases where ucb$ _ {t-1} (x)$ and lcb$ _ {t-1} (x)$ cannot be explicitly calculated.

[7] Change in notation in Lemma 3.1 (Reviewer DyNJ, HJS8). On page 8, we added superscript $(f)$ to the notation in Lemma 3.1.

[8] Explanation of the significance of Lemma 3.1 (Reviewer DyNJ, HJS8). On page 9, lines 1-4, we explained the purpose of introducing Lemma 3.1.

[9] Change in the definition of $\gamma_t$ (Reviewer DyNJ, HJS8). On page 9, lines 17-19, we added subscripts () to the symbols used in the definition of $\gamma_t$.

[10] Explanation of stronger assumptions regarding $q(a)$ (Reviewer Nk3H). On page 9, lines 29-34, we explained that $q(a)$ in this setting is stronger than that in Inatsu et al. (2024a).

[11] Regarding the definition of $\hat{t}$ (Reviewer HJS8). On page 11, lines 28-29, we added the explanation that $\hat{t}$ can also be defined without using the unknown $x^\ast$, and why it is defined in the form (9).

[12] Interpretation of $\mathbb{E} [ r_ {\hat{t}} ]$ (Reviewer Nk3H). On page 12, lines 1–9, we added the explanation of the evaluation of $\mathbb{E} [ r_ {\hat{t}} ]$.

[13] Non-tightness with respect to $\delta$ in high-probability bounds (Reviewer Nk3H). On page 12, lines 23, 24, and 26, and on page 19, line 5, we added the explanation that the high-probability bounds are not tight with respect to $\delta$ compared to existing ones.

[14] Tightness wit respect to $t$ in high-probability bounds (Reviewer Nk3H). On page 12, lines 27-38, we explained that when $\delta$ is considered a constant, the proposed high-probability bounds are tighter than existing ones with respect to $t$.

[15] Extension to other settings (Reviewer DyNJ, Nk3H). On page 12, in Section 4.1, we added the proposed method for uncontrollable settings and the results. We also explained that extension to continuous spaces is possible but some issues remain.

[16] BBBMOBO (fixed) and Proposed (fixed) (Reviewer DyNJ). On page 15, lines 24-25, we explained that BBBMOBO (fixed) and Proposed (fixed) are the same in the expectation measure. The proof was added to Appendix D.2, and we added explanations to Figure 2-4 that the results overlap.

[17] Explanation of getting stuck in local solutions (Reviewer DyNJ). On page 15, lines 37-40, we cited the results of existing BO and explained that a small $\beta_t$ causes getting stuck in local solutions.

[18] Addition of experiments under uncontrollable settings (Reviewer DyNJ). On page 15, we moved the experimental results under uncontrollable settings from Appendix to Section 5.2.

[19] Regarding expected bounds for robustness measures (Reviewer Nk3H). On page 18, last line, we emphasized that our approach provides theoretically valid bounds on expected regret regarding various robustness measures.

[20] Regarding the drawbacks of the proposed method (Reviewer DyNJ). On page 19, lines 5-7, we added the explanation that although extension to continuous spaces is possible, the advantages in finite spaces are lost.

[21] References (Reviewer HJS8). On page 19, in References, we corrected "gaussian" and "bayesian" to "Gaussian" and "Bayesian," respectively.

[22] Typo corrections (Reviewer DyNJ). On page 8, we corrected ucb$ _t (x)$ and  lcb$ _t (x)$ in Definition 3.1 to ucb$ _{t-1} (x)$ and  lcb$ _{t-1} (x)$, respectively. We also corrected (fix) in the experiments to (fixed).

---

### Decision · Action_Editor_E5mD · 2025-07-11

**Recommendation:** Accept as is

**Additional Comments:**

The authors can consider the following missing references for distributionally robust Bayesian optimization algorithms after that of Kirschner et al. (2020) and describe the relation to this work (including a unifying framework):

Efficient Distributionally Robust Bayesian Optimization with Worst-case Sensitivity. ICML 2022.
Distributionally Robust Bayesian Optimization with φ-divergences. NeurIPS 2023.
A Unified Framework for Bayesian Optimization under Contextual Uncertainty. ICLR 2024.

**Audience:**

Yes

**Audience Explanation:**

Yes, there would be researchers interested to know the findings of this paper due to the main contribution specified above and also that this work caters to several robustness measures.

**Claims And Evidence:**

Yes

**Claims Explanation:**

Initially, there is a concern by the reviewers that the main contribution of this paper (i.e., not needing to adjust the exploration parameter $beta_t$) cannot be extended to the case of continuous space. However, as clarified by the author in the rebuttal, the case of continuous space is supplementary and should not be part of the main contribution.

Another initial concern is that in the experimental results, the proposed method is not outperforming existing ones. As clarified by the author in the rebuttal, there is no claim that the proposed method would outperform them. Instead, the author clarified that one should view the main contribution as doing away with the need to adjust the exploration parameter through randomization, with theoretical performance guarantees. The author did not claim better empirical performance for the proposed method.

Finally, initially, the technical clarity is poor with several notations needing clarifications, as pointed out carefully by the reviewers. The author has clarified them in the rebuttal and revised the paper accordingly.

So yes, the claims made in the submission are supported by accurate, convincing and clear evidence after the revision.

---

> ### Author Response · Authors · 2025-07-23
> **Camera-Ready Submission**
>
> Dear Action Editor,
>
> We sincerely thank you for your decision. We have submitted a camera-ready version of the manuscript, which includes the following changes:
>
> * In page 2, line 48, we have changed the sentence "distributionally robust expectation (Kirschner et al., 2020)." to "distributionally robust measure (Kirschner et al., 2020; Tay et al., 2022; 2024)."
>
> * In pge 2, line 51, we have changed the sentence "Thompson sampling-based methods (Iwazaki et al., 2021a)." to "Thompson sampling-based methods (Iwazaki et al., 2021a; Tay et al., 2024)."
>
> * In page 3, line 1, we have added the sentence ", and approximation-based methods (Tay et al., 2022)."
>
> * In page 3, lines 9-16, we have added the sentence "On the other hand,... theoretical bounds they derived."
>
> * In References, we have added two references Tay et al. (2022; 2024).
>
> * In page 29, line 7, we have changed the sentence "Define $\beta_t = 2 \log (|\mathcal{X} \times \Omega|) + \xi_t$." to
> "Let $\kappa^{(4)}_t = |\mathcal{X} \times \Omega|$, and define $\beta_t = 2 \log \kappa^{(4)}_t + \xi_t$."
>
> * In Theorem B.3 and B.4, we have revised the typos "$\kappa^{(1)}_t$" to "$\kappa^{(4)}_t$."